# ORDER MATTERS: AGENT-BY-AGENT POLICY OPTIMIZATION

**Xihuai Wang**[1,2]*, **Zheng Tian**[3]†, **Ziyu Wan**[1,2], **Ying Wen**[1], **Jun Wang**[2,4], **Weinan Zhang**[1]†
[1] Shanghai Jiao Tong University, [2] Digital Brain Lab,
[3] ShanghaiTech University, [4] University College London

## ABSTRACT

While multi-agent trust region algorithms have achieved great success empirically in solving coordination tasks, most of them, however, suffer from a non-stationarity problem since agents update their policies simultaneously. In contrast, a sequential scheme that updates policies agent-by-agent provides another perspective and shows strong performance. However, sample inefficiency and lack of monotonic improvement guarantees for each agent are still the two significant challenges for the sequential scheme. In this paper, we propose the **A**gent-by-**a**gent **P**olicy **O**ptimization (A2PO) algorithm to improve the sample efficiency and retain the guarantees of monotonic improvement for each agent during training. We justify the tightness of the monotonic improvement bound compared with other trust region algorithms. From the perspective of sequentially updating agents, we further consider the effect of agent updating order and extend the theory of non-stationarity into the sequential update scheme. To evaluate A2PO, we conduct a comprehensive empirical study on four benchmarks: StarCraftII, Multi-agent MuJoCo, Multi-agent Particle Environment, and Google Research Football full game scenarios. A2PO consistently outperforms strong baselines.

## 1 INTRODUCTION

Trust region learning methods in reinforcement learning (RL) (Kakade & Langford, 2002) have achieved great success in solving complex tasks, from single-agent control tasks (Andrychowicz et al., 2020) to multi-agent applications (Albrecht & Stone, 2018; Ye et al., 2020). The methods deliver superior and stable performances because of their theoretical guarantees of monotonic policy improvement. Recently, several works that adopt trust region learning in multi-agent reinforcement learning (MARL) have been proposed, including algorithms in which agents independently update their policies using trust region methods (de Witt et al., 2020; Yu et al., 2022) and algorithms that coordinate agents' policies during the update process (Wu et al., 2021; Kuba et al., 2022). Most algorithms update the agents simultaneously, that is, all agents perform policy improvement at the same time and cannot observe the change of other agents, as shown in Fig. 1c. The simultaneous update scheme brings about the non-stationarity problem, i.e., the environment dynamic changes from one agent's perspective as other agents also change their policies (Hernandez-Leal et al., 2017).

In contrast to the simultaneous update scheme, algorithms that sequentially execute agent-by-agent updates allow agents to perceive changes made by preceding agents, presenting another perspective for analyzing inter-agent interaction (Gemp et al., 2022). Bertsekas (2021) proposed a sequential update framework, named Rollout and Policy Iteration for a Single Agent (RPISA) in this paper, which performs a rollout every time an agent updates its policy (Fig. 1a). RPISA effectively turns non-stationary MARL problems into stationary single agent reinforcement learning (SARL) ones. It retains the theo-

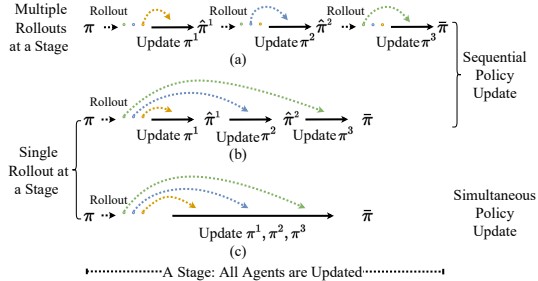

Figure 1: The taxonomy on the rollout scheme and the policy update scheme.

---

*Work done at Digital Brain Lab. †Correspondence to Zheng Tian <tianzheng@shanghaitech.edu.cn> and Weinan Zhang <wnzhang@sjtu.edu.cn>.

retical properties of the chosen SARL base algorithm, such as the monotonic improvement (Kakade & Langford, 2002). However, it is sample-inefficient since it only utilizes $1/n$ of the collected samples to update $n$ agents' policies. On the other hand, heterogeneous Proximal Policy Optimization (HAPPO) (Kuba et al., 2022) sequentially updates agents based on their local advantages estimated from the same rollout samples (Fig. 1b). Although it avoids the waste of collected samples and has a monotonic improvement on the joint policy, the policy improvement of a single agent is not theoretically guaranteed. Consequently, one agent's policy update may offset previous agents' policy improvement, reducing the overall joint policy improvement.

In this paper, we aim to combine the merits of the existing single rollout and sequential policy update schemes. Firstly, we show that naive sequential update algorithms with a single rollout can lose the monotonic improvement guarantee of PPO for a single agent's policy. To tackle this problem, we propose a surrogate objective with a novel off-policy correction method, *preceding-agent off-policy correction* (PreOPC), which retains the monotonic improvement guarantee on both the joint policy and each agent's policy. Then we further show that the joint monotonic bound built on the single agent bound is tighter than those of other simultaneous update algorithms and is tightened during updating the agents at a stage[1]. This leads to *Agent-by-agent Policy Optimization* (A2PO), a novel sequential update algorithm with single rollout scheme (Fig. 1b). Further, we study the significance of the agent update order and extend the theory of non-stationarity to the sequential update scheme. We test A2PO on four popular cooperative multi-agent benchmarks: StarCraftII, multi-agent MuJoCo, multi-agent particle environment, and Google Research Football full game scenarios. On all benchmark tasks, A2PO consistently outperforms strong baselines with a large margin in both performance and sample efficiency and shows an advantage in encouraging inter-agent coordination. To sum up, the main contributions of this work are as follows:

1. **Monotonic improvement bound**. We prove that the guarantees of monotonic improvement on each agent's policy could be retained under the single rollout scheme with the off-policy correction method PreOPC we proposed. We further prove that the monotonic bound on the joint policy achieved given theoretical guarantees of each agent is the tightest among single rollout algorithms, yielding effective policy optimization.

2. **A2PO algorithm**. We propose A2PO, the first agent-by-agent sequential update algorithm that retains the monotonic policy improvement on both each agent's policy and the joint policy and does not require multiple rollouts when performing policy improvement.

3. **Agent update order**. We further investigate the connections between the sequential policy update scheme, the agent update order, and the non-stationarity problem, which motivates two novel methods: a semi-greedy agent selection rule for optimization acceleration and an adaptive clipping parameter method for alleviating the non-stationarity problem.

## 2 RELATED WORKS

Trust Region Policy Optimization (TRPO) (Schulman et al., 2015) and Proximal Policy Optimization (PPO) (Schulman et al., 2017) are popular trust region algorithms with strong performances, benefiting from the guarantee of monotonic policy improvement (Kakade & Langford, 2002). Several recent works delve deeper into understanding these methods (Wang et al., 2019; Liu et al., 2019; Wang et al., 2020). In the multi-agent scenarios, de Witt et al. (2020) and Papoudakis et al. (2020) empirically studied the performance of Independent PPO in multi-agent tasks. Yu et al. (2022) conducted a comprehensive benchmark and analyzed the factor influential to the performance of Multi-agent PPO (MAPPO), a variant of PPO with centralized critics. Coordinate PPO (CoPPO) (Wu et al., 2021) integrates the value decomposition (Sunehag et al., 2017) and approximately performs a joint policy improvement with monotonic improvement. Several further trials to implement trust region methods are discussed in Wen et al. (2021); Li & He (2020); Sun et al. (2022); Ye et al. (2022). However, these MARL algorithms suffer from the non-stationarity problem as they update agents simultaneously. The environment dynamic changes from one agent's perspective as others also change their policies. Consequently, agents suffer from the high variance of gradients and require more samples for convergence (Hernandez-Leal et al., 2017). To alleviate the non-stationarity problem, Multi-Agent Mirror descent policy algorithm with Trust region decomposition (MAMT) (Li et al., 2022b) factorizes the trust regions of the joint policy and constructs the connections among the factorized trust regions, approximately constraining the diversity of joint policy.

---

[1]We define a *stage* as a period during which all the agents have been updated once (Fig. 1).

Rollout and Policy Iteration for a Single Agent (RPISA) (Bertsekas, 2021) and Heterogeneous PPO (HAPPO) (Kuba et al., 2022) consider the sequential update scheme. RPISA suffers from sample inefficiency as it requires $n$ times of rollout for $n$ agents to complete their policies update. Additionally, their work lacks a practical algorithm for complex tasks. In contrast, we propose a practical algorithm A2PO that updates all agents using the same samples from a single rollout. HAPPO is derived from the advantage decomposition lemma, proposed as Lemma 1 in Kuba et al. (2022). It does not consider the distribution shift caused by preceding agents, and has no monotonic policy improvement guarantee for each agent's policy. While A2PO is derived without decomposing the advantage, and has a guarantee of monotonic improvement for each agent's policy. We further discuss other MARL methods in Appx. C.

## 3 TRUST REGION METHOD IN SEQUENTIAL POLICY UPDATE SCHEME

### 3.1 MARL PROBLEM FORMULATION

We consider formulating the sequential decision-making problem in multi-agent scenarios as a decentralized Markov decision process (DEC-MDP) (Bernstein et al., 2002). An $n$-agent DEC-MDP can be formalized as a tuple $(\mathcal{S}, \{\mathcal{A}^i\}_{i \in \mathcal{N}}, r, \mathcal{T}, \gamma)$, where $\mathcal{N} = \{1, \ldots, n\}$ is the set of agents, $\mathcal{S}$ is the state space. $\mathcal{A}^i$ is the action space of agent $i$, and $\mathcal{A} = \mathcal{A}^1 \times \cdots \times \mathcal{A}^n$ is the joint action space. $r : \mathcal{S} \times \mathcal{A} \mapsto \mathbb{R}$ is the reward function, and $\mathcal{T} : \mathcal{S} \times \mathcal{A} \times \mathcal{S} \mapsto [0, 1]$ is the dynamics function denoting the transition probability. $\gamma \in [0, 1)$ is a reward discount factor. At time step $t$, each agent $i$ takes action $a_t^i$ from its policy $\pi^i(\cdot|s_t)$, simultaneously according to the state $s_t$, forming the joint action $\boldsymbol{a}_t = \{a_t^1, \ldots, a_t^n\}$ and the joint policy $\boldsymbol{\pi}(\cdot|s_t) = \pi^1 \times \ldots \times \pi^n$. The joint policy $\boldsymbol{\pi}$ of these $n$ agents induces a normalized discounted state visitation distribution $d^{\boldsymbol{\pi}}$, where $d^{\boldsymbol{\pi}}(s) = (1-\gamma)\sum_{t=0}^{\infty}\gamma^t Pr(s_t = s|\boldsymbol{\pi})$ and $Pr(\cdot|\boldsymbol{\pi}) : \mathcal{S} \mapsto [0, 1]$ is the probability function under $\boldsymbol{\pi}$. We then define the value function $V^{\boldsymbol{\pi}}(s) = \mathbb{E}_{\tau \sim (\mathcal{T}, \boldsymbol{\pi})}[\sum_{t=0}^{\infty}\gamma^t r(s_t, \boldsymbol{a}_t)|s_0 = s]$ and the advantage function $A^{\boldsymbol{\pi}}(s, \boldsymbol{a}) = r(s, \boldsymbol{a}) + \gamma\mathbb{E}_{s' \sim \mathcal{T}(\cdot|s, \boldsymbol{a})}[V^{\boldsymbol{\pi}}(s')] - V^{\boldsymbol{\pi}}(s)$, where $\tau = \{(s_0, \boldsymbol{a}_0), (s_1, \boldsymbol{a}_1), \ldots\}$ denotes one sampled trajectory. The agents maximize their expected return, denoted as: $\boldsymbol{\pi}^* = \text{argmax}_{\boldsymbol{\pi}} \mathcal{J}(\boldsymbol{\pi}) = \text{argmax}_{\boldsymbol{\pi}} \mathbb{E}_{\tau \sim (\mathcal{T}, \boldsymbol{\pi})}[\sum_{t=0}^{\infty}\gamma^t r(s_t, \boldsymbol{a}_t)]$ ,.

### 3.2 MONOTONIC IMPROVEMENT IN SEQUENTIAL POLICY UPDATE SCHEME

We assume agents are updated in the order $1, 2, \ldots, n$, without loss of generality. We define $\boldsymbol{\pi}$ as the joint base policy from which the agents are updated at a stage, $e^i = \{1, \ldots, i-1\}$ as the set of preceding agents updated before agent $i$, and $\bar{\pi}^i$ as the updated policy of agent $i$. We denote the joint policy composed of updated policies of agents in the set $e^i$, the updated policy of agent $i$ and base policies of other agents as $\hat{\boldsymbol{\pi}}^i = \bar{\pi}^1 \times \ldots \times \bar{\pi}^i \times \pi^{i+1} \times \ldots \times \pi^n$, and define $\hat{\boldsymbol{\pi}}^0 = \boldsymbol{\pi}$ and $\hat{\boldsymbol{\pi}}^n = \bar{\boldsymbol{\pi}}$. A general sequential update scheme is shown as follows, where $\mathcal{L}_{\hat{\boldsymbol{\pi}}^{i-1}}(\hat{\boldsymbol{\pi}}^i)$ is the surrogate objective for agent $i$:

$$\boldsymbol{\pi} = \hat{\boldsymbol{\pi}}^0 \xrightarrow[\text{Update } \pi^1]{\max_{\pi^1} \mathcal{L}_{\boldsymbol{\pi}}(\hat{\boldsymbol{\pi}}^1)} \hat{\boldsymbol{\pi}}^1 \rightarrow \cdots \rightarrow \hat{\boldsymbol{\pi}}^{n-1} \xrightarrow[\text{Update } \pi^n]{\max_{\pi^n} \mathcal{L}_{\hat{\boldsymbol{\pi}}^{n-1}}(\hat{\boldsymbol{\pi}}^n)} \hat{\boldsymbol{\pi}}^n = \bar{\boldsymbol{\pi}}.$$

We wish our sequential update scheme retains the desired monotonic improvement guarantee while improving the sample efficiency. Before going to our method, we first discuss **_why naively updating agents sequentially with the same rollout samples will fail in monotonic improvement for each agent._** Since agent $i$ updates its policy from $\hat{\boldsymbol{\pi}}^{i-1}$, an intuitive surrogate objective (Schulman et al., 2015) used by agent $i$ could be formulated as $\mathcal{L}_{\hat{\boldsymbol{\pi}}^{i-1}}^I(\hat{\boldsymbol{\pi}}^i) = \mathcal{J}(\hat{\boldsymbol{\pi}}^{i-1}) + \mathcal{O}_{\boldsymbol{\pi}}(\hat{\boldsymbol{\pi}}^i)$, where $\mathcal{O}_{\boldsymbol{\pi}}(\hat{\boldsymbol{\pi}}^i) = \frac{1}{1-\gamma}\mathbb{E}_{(s, \boldsymbol{a}) \sim (d^{\boldsymbol{\pi}}, \hat{\boldsymbol{\pi}}^i)}[A^{\boldsymbol{\pi}}(s, \boldsymbol{a})]$ and the superscript $I$ means 'Intuitive'. The expected return, however, is not guaranteed to improve with such a surrogate objective, as elaborated in the following proposition.

**Proposition 1** *For agent $i$, let $\epsilon = \max_{s, \boldsymbol{a}}|A^{\boldsymbol{\pi}}(s, \boldsymbol{a})|$, $\alpha^j = D_{TV}^{\max}(\pi^j\|\bar{\pi}^j) \ \forall j \in (e^i \cup \{i\})$, where $D_{TV}(p\|q)$ is the total variation distance between distributions $p$ and $q$ and we define $D_{TV}^{\max}(\pi\|\bar{\pi}) = \max_s D_{TV}(\pi(\cdot|s)\|\bar{\pi}(\cdot|s))$, then we have:*

$$\left|\mathcal{J}(\hat{\boldsymbol{\pi}}^i) - \mathcal{L}_{\hat{\boldsymbol{\pi}}^{i-1}}^I(\hat{\boldsymbol{\pi}}^i)\right| \leq 2\epsilon\alpha^i\left(\frac{3}{1-\gamma} - \frac{2}{1-\gamma(1-\sum_{j \in (e^i \cup \{i\})}\alpha^j)}\right) + \overbrace{\frac{2\epsilon\sum_{j \in e^i}\alpha^j}{1-\gamma}}^{Uncontrollable} = \beta_i^I \ . \ (1)$$

The proof can be found in Appx. A.3.

**Remark**. From Eq. (1) and the definition of $\mathcal{L}^I_{\hat{\pi}^{i-1}}$, we know $\mathcal{J}(\hat{\pi}^i) - \mathcal{J}(\hat{\pi}^{i-1}) > \mathcal{O}_{\pi}(\hat{\pi}^i) - \beta^I_i$. Thus $\mathcal{J}(\hat{\pi}^i) > \mathcal{J}(\hat{\pi}^{i-1})$ when $\mathcal{O}_{\pi}(\hat{\pi}^i) > \beta^I_i$, which can be satisfied by constraining $\beta^I_i$ and optimizing $\mathcal{O}_{\pi}(\hat{\pi}^i)$. However, in $\beta^I_i$, the term $2\epsilon \sum_{j \in e^i} \alpha^j / (1 - \gamma)$, is uncontrollable by agent $i$. Consequently, the upper bound $\beta^I_i$ may be large and the expected performance $\mathcal{J}(\hat{\pi}^i)$ may not be improved after optimizing $\mathcal{O}_{\pi}(\hat{\pi}^i)$ when $\mathcal{O}_{\pi}(\hat{\pi}^i) < \beta^I_i$ even if $\alpha^i$ is well constrained. Although one can still prove a monotonic guarantee for the joint policy by summing Eq. (1) for all the agents, we will show that the monotonic improvement on every single agent, if guaranteed, brings a tighter monotonic bound on the joint policy and incrementally tightens the monotonic bound on the joint policy when updating agents during a stage. Uncontrollable terms also appear when similarly analyzing HAPPO and cause the loss of monotonic improvement for a single agent[2].

### 3.3 PRECEDING-AGENT OFF-POLICY CORRECTION

The uncontrollable term in Prop. 1 is caused by one ignoring how the updating of its preceding agents' policies influences its advantage function. We investigate reducing the uncontrollable term in policy evaluation. Since agent $i$ is updated from $\hat{\pi}^{i-1}$, the advantage function $A^{\hat{\pi}^{i-1}}$ should be used in agent $i$'s surrogate objective rather than $A^{\pi}$. However, $A^{\hat{\pi}^{i-1}}$ is impractical to estimate using samples collected under $\pi$ due to the off-policyness (Munos et al., 2016) of these samples. Nevertheless, we can approximate $A^{\hat{\pi}^{i-1}}$ by correcting the discrepancy between $\hat{\pi}^{i-1}$ and $\pi$ at each time step (Harutyunyan et al., 2016). To retain the monotonic improvement properties, we propose *preceding-agent off-policy correction* (PreOPC), which approximates $A^{\hat{\pi}^{i-1}}$ using samples collected under $\pi$ by correcting the state probability at each step with truncated product weights:

$$A^{\pi, \hat{\pi}^{i-1}}(s_t, \boldsymbol{a}_t) = \delta_t + \sum_{k \geq 1} \gamma^k \Big( \prod_{j=1}^k \lambda \min \big(1.0, \frac{\hat{\pi}^{i-1}(\boldsymbol{a}_{t+j}|s_{t+j})}{\pi(\boldsymbol{a}_{t+j}|s_{t+j})} \big) \Big) \delta_{t+k} , \qquad (2)$$

where $\delta_t = r(s_t, \boldsymbol{a}_t) + \gamma V(s_{t+1}) - V(s_t)$ is the temporal difference for $V(s_t)$, $\lambda$ is a parameter controlling the bias and variance, as used in Schulman et al. (2016). $\min(1.0, \frac{\hat{\pi}^{i-1}(\boldsymbol{a}_{t+j}|s_{t+j})}{\pi(\boldsymbol{a}_{t+j}|s_{t+j})}) \ \forall j \in \{1, \ldots, k\}$ are truncated importance sampling weights, approximating the probability of $s_{t+k}$ at time step $t + k$ under $\hat{\pi}^{i-1}$. The derivation of Eq. (2) can be found in Appx. A.8. With PreOPC, the surrogate objective of agent $i$ becomes $\mathcal{L}_{\hat{\pi}^{i-1}}(\hat{\pi}^i) = \mathcal{J}(\hat{\pi}^{i-1}) + \frac{1}{1-\gamma}\mathbb{E}_{(s,\boldsymbol{a}) \sim (d^{\pi}, \hat{\pi}^i)}[A^{\pi, \hat{\pi}^{i-1}}(s, \boldsymbol{a})]$ , and we summarize the surrogate objective of updating all agents as follows:

$$\mathcal{G}_{\pi}(\bar{\pi}) = \mathcal{J}(\pi) + \frac{1}{1 - \gamma} \sum_{i=1}^n \mathbb{E}_{(s,\boldsymbol{a}) \sim (d^{\pi}, \hat{\pi}^i)}[A^{\pi, \hat{\pi}^{i-1}}(s, \boldsymbol{a})] . \qquad (3)$$

Note that Eq. (3) takes the sum of expectations of the global advantage function approximated under different joint policies, different from the advantage decomposition lemma in Kuba et al. (2022) which decomposes the global advantage function into local ones.

We can now prove that the monotonic policy improvement guarantee of both updating one agent's policy and updating the joint policy is retained by using Eq. (3) as the surrogate objective. The detailed proofs can be found in Appx. A.4.

**Theorem 1 (Single Agent Monotonic Bound)** *For agent $i$, let $\epsilon^i = \max_{s,\boldsymbol{a}} |A^{\hat{\pi}^{i-1}}(s, \boldsymbol{a})|$, $\xi^i = \max_{s,\boldsymbol{a}} |A^{\pi, \hat{\pi}^{i-1}}(s, \boldsymbol{a}) - A^{\hat{\pi}^{i-1}}(s, \boldsymbol{a})|$, $\alpha^j = D^{\max}_{TV}(\pi^j \| \bar{\pi}^j) \ \forall j \in (e^i \cup \{i\})$, then we have:*

$$\Big| \mathcal{J}(\hat{\pi}^i) - \mathcal{L}_{\hat{\pi}^{i-1}}(\hat{\pi}^i) \Big| \leq 4\epsilon^i \alpha^i \Big( \frac{1}{1 - \gamma} - \frac{1}{1 - \gamma(1 - \sum_{j \in (e^i \cup \{i\})} \alpha^j)} \Big) + \frac{\xi^i}{1 - \gamma}$$

$$\leq \frac{4\gamma\epsilon^i}{(1 - \gamma)^2} \Big( \alpha^i \sum_{j \in (e^i \cup \{i\})} \alpha^j \Big) + \frac{\xi^i}{1 - \gamma} . \qquad (4)$$

The single agent monotonic bound depends on $\epsilon^i$, $\xi^i$, and $\alpha^i$ and the total variation distances of preceding agents. Unlike Eq. (1), we can effectively constrain the monotonic bound by controlling $\alpha^i$ since $\xi^i$ decreases as agent $i$ updating its value function (Munos et al., 2016) and does not

---

[2]More discussions about why HAPPO fails to guarantee monotonic improvement for a single agent's policy can be found in Appx. A.6.

Table 1: Comparisons of trust region MARL algorithms. The proofs of the monotonic bounds can be found in Appx. A. Note that we also provide the monotonic bound of RPISA-PPO, which implements RPISA with PPO as the base algorithm. We separate RPISA-PPO from other methods as it has low sample efficiency and thus does not constitute a fair comparison.

| Algorithm | Rollout | Update | Sample Efficiency | Monotonic Bound |
|-----------|---------|--------|-------------------|-----------------|
| RPISA-PPO | Multiple | Sequential | Low | $4\epsilon \sum_{i=1}^n \alpha^i \left( \frac{1}{1-\gamma} - \frac{1}{1-\gamma(1-\alpha^i)} \right)$ 
 Single Agent: $4\epsilon\alpha^i \left( \frac{1}{1-\gamma} - \frac{1}{1-\gamma(1-\alpha^i)} \right)$ |
| MAPPO | Single | Simultaneous | High | $4\epsilon \sum_{i=1}^n \frac{\alpha^i}{1-\gamma}$ |
| CoPPO | Single | Simultaneous | High | $4\epsilon \sum_{i=1}^n \alpha^i \left( \frac{1}{1-\gamma} - \frac{1}{1-\gamma(1-\sum_{j=1}^n \alpha^j)} \right)$ |
| HAPPO | Single | Sequential | High | $4\epsilon \sum_{i=1}^n \alpha^i \left( \frac{1}{1-\gamma} - \frac{1}{1-\gamma(1-\sum_{j=1}^n \alpha^j)} \right)$ 
 Single Agent: No Guarantee |
| A2PO (ours) | Single | Sequential | High | $4\epsilon \sum_{i=1}^n \alpha^i \left( \frac{1}{1-\gamma} - \frac{1}{1-\gamma(1-\sum_{j\in(e^i \cup \{i\})} \alpha^j)} \right) + \frac{\sum_{i=1}^n \xi^i}{1-\gamma}$ 
 Single Agent: $4\epsilon^i\alpha^i \left( \frac{1}{1-\gamma} - \frac{1}{1-\gamma(1-\sum_{j\in(e^i \cup \{i\})} \alpha^j)} \right) + \frac{\xi^i}{1-\gamma}$ |

lead to an unsatisfiable bound when $\alpha^i$ is well constrained, providing the guarantee for monotonic improvement when updating a single agent. Given the above bound, we can prove the monotonic improvement of the joint policy.

**Theorem 2 (Joint Monotonic Bound)** *For each agent $i \in \mathcal{N}$, let $\epsilon^i = \max_{s,\boldsymbol{a}} |A^{\hat{\boldsymbol{\pi}}^{i-1}}(s,\boldsymbol{a})|$, $\alpha^i = D_{TV}^{\max}(\pi^i \| \bar{\pi}^i)$, $\xi^i = \max_{s,\boldsymbol{a}} |A^{\boldsymbol{\pi},\hat{\boldsymbol{\pi}}^{i-1}}(s,\boldsymbol{a}) - A^{\hat{\boldsymbol{\pi}}^{i-1}}(s,\boldsymbol{a})|$, and $\epsilon = \max_i \epsilon^i$, then we have:*

$$|\mathcal{J}(\bar{\boldsymbol{\pi}}) - \mathcal{G}_{\boldsymbol{\pi}}(\bar{\boldsymbol{\pi}})| \leq 4\epsilon \sum_{i=1}^n \alpha^i \left( \frac{1}{1-\gamma} - \frac{1}{1-\gamma(1-\sum_{j\in(e^i \cup \{i\})} \alpha^j)} \right) + \frac{\sum_{i=1}^n \xi^i}{1-\gamma}$$

$$\leq \frac{4\gamma\epsilon}{(1-\gamma)^2} \sum_{i=1}^n \left( \alpha^i \sum_{j\in(e^i \cup \{i\})} \alpha^j \right) + \frac{\sum_{i=1}^n \xi^i}{1-\gamma} . \tag{5}$$

Eq. (5) suggests a condition for monotonic improvement of the joint policy, similar to that in the remark under Prop. 1. We further prove that the joint monotonic bound is incrementally tightened when performing the policy optimization agent-by-agent during a stage due to the single agent monotonic bound, i.e., the condition for improving $\mathcal{J}(\bar{\boldsymbol{\pi}})$ is relaxed and more likely to be satisfied. The details can be found in Appx. A.5. We present the monotonic bounds of other algorithms in Tab. 1. Since $-\frac{1}{1-\gamma(1-\sum_{j\in(e^i \cup \{i\})} \alpha^j)} < -\frac{1}{1-\gamma(1-\sum_{j=1}^n \alpha^j)}$, Eq. (5) achieves the tightest bound compared to other single rollout algorithms, with $\xi^i \ \forall i \in \mathcal{N}$ small enough. The assumption about $\xi^i$ is valid since preceding-agent off-policy correction is a contraction operator, which is a corollary of Theorem 1 in Munos et al. (2016). A tighter bound improves expected performance by optimizing the surrogate objective more effectively (Li et al., 2022a).

## 4 AGENT-BY-AGENT POLICY OPTIMIZATION

We first give a practical implementation for optimizing the surrogate objective $\mathcal{G}_{\boldsymbol{\pi}}(\bar{\boldsymbol{\pi}})$. When updating agent $i$, the monotonic bound in Eq. (4) consists of the total variation distances related to the preceding agents and agent $i$, i.e., $\alpha^i \sum_{j\in(e^i \cup \{i\})} \alpha^j$. It suggests that we can control the monotonic bound by controlling total variation distances $\alpha^j \ \forall j \in (e^i \cup \{i\})$, to effectively improve the expected performance. We consider applying the clipping mechanism to control the total variation distances $\alpha^j \ \forall j \in (e^i \cup \{i\})$ (Queeney et al., 2021; Sun et al., 2022). In the surrogate objective of agent $i$, i.e., $\mathcal{J}(\hat{\boldsymbol{\pi}}^{i-1}) + \frac{1}{1-\gamma} \mathbb{E}_{(s,\boldsymbol{a})\sim(d^{\boldsymbol{\pi}},\boldsymbol{\pi})} \left[ \frac{\bar{\pi}^i \prod_{j\in e^i} \bar{\pi}^j}{\pi^i \prod_{j\in e^i} \pi^j} A^{\boldsymbol{\pi},\hat{\boldsymbol{\pi}}^{i-1}}(s,\boldsymbol{a}) \right]$, $\mathcal{J}(\hat{\boldsymbol{\pi}}^{i-1})$ has no dependence to agent $i$, while the joint policy ratio $\frac{\bar{\pi}^i \prod_{j\in e^i} \bar{\pi}^j}{\pi^i \prod_{j\in e^i} \pi^j}$ in the advantage estimation is appropriate for applying the clipping mechanism. We further consider reducing the instability in estimating agent $i$'s policy gradient by clipping the joint policy ratio of preceding agents first, with a narrower clipping range (Wu et al., 2021). Thus we apply the clipping mechanism on the joint policy ratio twice: once on the joint policy ratio of preceding agents and once on the policy ratio of agent $i$. Finally, the practical objective for updating agent $i$ becomes:

$$\tilde{\mathcal{L}}_{\hat{\boldsymbol{\pi}}^{i-1}}(\hat{\boldsymbol{\pi}}^i) = \mathbb{E}_{(s,\boldsymbol{a})\sim(d^{\boldsymbol{\pi}},\boldsymbol{\pi})} \left[ \min \left( l(s,\boldsymbol{a}) A^{\boldsymbol{\pi},\hat{\boldsymbol{\pi}}^{i-1}}, \text{clip} \left( l(s,\boldsymbol{a}), 1 \pm \epsilon^i \right) A^{\boldsymbol{\pi},\hat{\boldsymbol{\pi}}^{i-1}} \right) \right] , \tag{6}$$

where $l(s, \boldsymbol{a}) = \frac{\bar{\pi}^i(a^i|s)}{\pi^i(a^i|s)} g(s, \boldsymbol{a})$, and $g(s, \boldsymbol{a}) = \text{clip}(\frac{\prod_{j \in e^i} \bar{\pi}^j(a^j|s)}{\prod_{j \in e^i} \pi^j(a^j|s)}, 1 \pm \frac{\epsilon^i}{2})$. The clipping parameter $\epsilon^i$ is selected as $\epsilon^i = \mathcal{C}(\epsilon, i)$, where $\epsilon$ is the base clipping parameter and $\mathcal{C}(\cdot, \cdot)$ is the clipping parameter adapting function. We summarize our proposed *Agent-by-agent Policy Optimization* (A2PO) in Alg. 1. Note that in Line 6, the agent for the next update iteration is selected according to the agent selection rule $\mathcal{R}(\cdot)$.

---

**Algorithm 1:** Agent-by-agent Policy Optimization (A2PO)

---

1   Initialize the joint policy $\boldsymbol{\pi}_0 = \{\pi_0^1, \ldots, \pi_0^n\}$, and the global value function $V$.
2   **for** *iteration* $m = 1, 2, \ldots$ **do**
3     Collect data using $\boldsymbol{\pi}_{m-1} = \{\pi_{m-1}^1, \ldots, \pi_{m-1}^n\}$.
4     **for** *Order* $k = 1, \ldots, n$ **do**
5       Select an agent according to the selection rule as $i = \mathcal{R}(k)$.
6       Policy $\pi_m^i = \pi_{m-1}^i$, preceding agents $e^i = \{\mathcal{R}(1), \ldots, \mathcal{R}(k-1)\}$.
7       Joint policy $\hat{\boldsymbol{\pi}}^i = \{\pi_m^i, \pi_m^{j \in e^k}, \pi_{m-1}^{j \in \mathcal{N}-e^k}\}$.
8       Compute the advantage approximation as $A^{\boldsymbol{\pi}, \hat{\boldsymbol{\pi}}^{i-1}}(s, \boldsymbol{a})$ via Eq. (2).
9       Compute the value target $v(s_t) = A^{\boldsymbol{\pi}, \hat{\boldsymbol{\pi}}^{i-1}}(s, \boldsymbol{a}) + V(s)$.
10      **for** *P epochs* **do**
11        $\pi_m^i = \arg\max_{\pi_m^i} \tilde{\mathcal{L}}_{\hat{\boldsymbol{\pi}}^{i-1}}(\hat{\boldsymbol{\pi}}^i)$ as in Eq. (6).
12        $V = \arg\min_V \mathbb{E}_{s \sim d^{\boldsymbol{\pi}}} \|v(s) - V(s)\|^2$.

---

Eq. (6) approximates the surrogate objective of a single agent. We remark that the monotonic improvement guarantee of a single agent reveals how the update of a single agent affects the overall objective. We will further discuss $\mathcal{R}(\cdot)$ and $\mathcal{C}(\cdot, \cdot)$ from the perspective of how to benefit the optimization of the overall surrogate objective by coordinating the policy updates of each agent.

**Semi-greedy Agent Selection Rule**. With the monotonic policy improvement guarantee on the joint policy, as shown in Thm. 2, we can effectively improve the expected performance $\mathcal{J}(\bar{\boldsymbol{\pi}})$ by optimizing the surrogate objective of all agents $\mathcal{G}_{\boldsymbol{\pi}}(\bar{\boldsymbol{\pi}}) = \mathcal{J}(\boldsymbol{\pi}) + \sum_{i=1}^n \mathcal{L}_{\hat{\boldsymbol{\pi}}^{i-1}}(\hat{\boldsymbol{\pi}}^i)$. Since the policies except $\pi^i$ are fixed when maximizing $\mathcal{L}_{\hat{\boldsymbol{\pi}}^{i-1}}(\hat{\boldsymbol{\pi}}^i)$, we recognize maximizing $\sum_{i=1}^n \tilde{\mathcal{L}}_{\hat{\boldsymbol{\pi}}^{i-1}}(\hat{\boldsymbol{\pi}}^i)$ as performing a block coordinate ascent, i.e., iteratively seeking to update a block of chosen coordinates (agents) while other blocks (agents) are fixed. As a special case of the coordinate selection rule, the agent selection rule becomes crucial for convergence. On the one hand, intuitively, updating agent with a bigger absolute value of the advantage function contributes more to optimizing $\mathcal{G}_{\boldsymbol{\pi}}(\bar{\boldsymbol{\pi}})$. Inspired by the Gauss-Southwell rule (Gordon & Tibshirani, 2015), we propose the greedy agent selection rule, under which an agent with a bigger absolute value of the expected advantage function is updated with a higher priority. We will verify that the agents with small absolute values of the advantage function also benefit from the greedy selelction rule in Appx. B.2.5. On the other hand, purely greedy selection may lead to early convergence which harms the performance. Therefore, we introduce randomness into the agent selection rule to avoid converging too early (Lu et al., 2018). Combining the merits, we propose the semi-greedy agent selection rule as

$$\begin{cases} \mathcal{R}(k) = \arg\max_{i \in (\mathcal{N}-e)} \mathbb{E}_{s, a^i} [|A^{\boldsymbol{\pi}, \hat{\boldsymbol{\pi}}^{\mathcal{R}(k-1)}}|], & k2 = 0 \\ \mathcal{R}(k) \sim \mathcal{U}(\mathcal{N}-e), & k2 = 1 \end{cases}$$, where $e = \{\mathcal{R}(1), \ldots, \mathcal{R}(k-1)\}$ and $\mathcal{U}$ is a uniform distribution. We verify that the semi-greedy agent selection rule contributes to the performance of A2PO in Sec. 5.2.

**Adaptive Clipping Parameter**. We improve the sample efficiency by updating all agents using the samples collected under the base joint policy $\boldsymbol{\pi}$. However, when updating agent $i$ by optimizing $\frac{1}{1-\gamma} \mathbb{E}_{(s, \boldsymbol{a}) \sim (d^{\boldsymbol{\pi}}, \hat{\boldsymbol{\pi}}^i)} [A^{\boldsymbol{\pi}, \hat{\boldsymbol{\pi}}^{i-1}}(s, \boldsymbol{a})]$, the expectation of advantage function is estimated using the states sampled under $\boldsymbol{\pi}$ instead of $\hat{\boldsymbol{\pi}}^{i-1}$, which reintroduces the non-stationarity since agent $i$ can not perceive the change of the preceding agents. With the non-stationarity modeled by the state transition shift (Sun et al., 2022), we define the state transition shift encountered when updating agent $i$ as $\Delta_{\pi^1, \ldots, \pi^n}^{\bar{\pi}^1, \ldots, \bar{\pi}^{i-1}, \pi^i, \ldots, \pi^n}(s'|s) = \sum_{\boldsymbol{a}} [\mathcal{T}(s'|s, \boldsymbol{a})(\hat{\boldsymbol{\pi}}^{i-1}(\boldsymbol{a}|s) - \boldsymbol{\pi}(\boldsymbol{a}|s))]$. The state transition shift has the following property.

**Proposition 2** *The state transition shift* $\Delta_{\pi^1,...,\pi^n}^{\bar{\pi}^1,...,\bar{\pi}^{i-1},\pi^i,...,\pi^n}(s'|s)$ *can be decomposed as follows.*

$$\Delta_{\pi^1,...,\pi^n}^{\bar{\pi}^1,...,\bar{\pi}^{i-1},\pi^i,...,\pi^n} = \Delta_{\pi^1,...,\pi^n}^{\bar{\pi}^1,\pi^2,...,\pi^n} + \Delta_{\bar{\pi}^1,\pi^2,...,\pi^n}^{\bar{\pi}^1,\bar{\pi}^2,\pi^3,...,\pi^n} + \cdots + \Delta_{\bar{\pi}^1,...,\bar{\pi}^{i-2},\pi^{i-1},...,\pi^n}^{\bar{\pi}^1,...,\bar{\pi}^{i-1},\pi^i,...,\pi^n}$$

Prop. 2 shows that the total state transition shift encountered by agent $i$ can be decomposed into the sum of state transition shift caused by each agent whose policy has been updated. Shifts caused by agents with higher priorities will be encountered by more following agents and thus contribute more to the non-stationarity problem. Recall that the state transition shift effectively measures the total variation distance between policies. Therefore, in order to reduce the non-stationarity brought by the agents' policy updates, we can adaptively clip each agent's surrogate objective according to their update priorities. We propose a simple yet effective method, named adaptive clipping parameter, to adjust the clipping parameters according to the updating order: $\mathcal{C}(\epsilon, k) = \epsilon \cdot c_\epsilon + \epsilon \cdot (1 - c_\epsilon) \cdot k/n$, where $c_\epsilon$ is a hyper-parameter. We demonstrate how the agents with higher priorities affect the following agents in Fig. 2. Under the clipping mechanism, the influence of the agents with higher priority could be reflected in the clipping ranges of the joint policy ratio. The policy changes of the preceding agents may constrain the following agents to optimize the surrogate objective within insufficient clipping ranges, as shown on the left side of Fig. 2. The right side of Fig. 2 demonstrates that the adaptive clipping parameter method leads to balanced and sufficient clipping ranges.

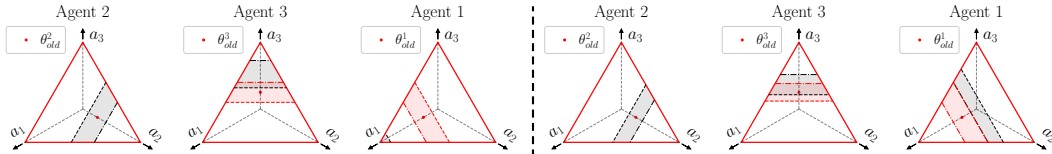

Figure 2: The clipping ranges of three agents. The surface $a_1 + a_2 + a_3 = 1$ demonstrates the policy space of three discrete actions. The agents are updated in the order of $2, 3, 1$. The areas in gray/pink are the clipping ranges with/without considering the joint policy ratio of preceding agents. **Left**: The agents have the same clipping parameters. The clipping range of agent 1 is insufficient due to the large variation in the policies of agent 2 and agent 3. **Right**: The clipping ranges are more balanced and sufficient with the adaptive clipping parameter method.

## 5 EXPERIMENTS

In this section, we empirically evaluate and analyze A2PO in the widely adopted cooperative multi-agent benchmarks, including the StarCraftII Multi-agent Challenge (SMAC) (Samvelyan et al., 2019), Multi-agent MuJoCo (MA-MuJoCo) (de Witt et al., 2020), Multi-agent Particle Environment (MPE) (Lowe et al., 2017)[3], and more challenging Google Research Football (GRF) full-game scenarios (Kurach et al., 2020). Experimental results demonstrate that **1)** *A2PO achieves performance and efficiency superior to those of state-of-the-art MARL Trust Region methods*, **2)** *A2PO has strength in encouraging coordination behaviors to complete complex cooperative tasks*, and **3)** *the PreOPC, the semi-greedy agent selection rule, and the adaptive clipping parameter methods significantly contribute to the performance improvement.* [4]

We compare A2PO with advanced MARL trust-region methods: MAPPO (Yu et al., 2022), CoPPO (Wu et al., 2021) and HAPPO (Kuba et al., 2022). We implement all the algorithms as parameter sharing in SMAC and MPE, and as parameter-independent in MA-MuJoCo and GRF, according to the homogeneity and heterogeneity of agents. We divide the agents into blocks for tasks with numerous agents to control the training time of A2PO comparable to other algorithms. Full experimental details can be found in Appx. B.

### 5.1 PERFORMANCE AND EFFICIENCY

We evaluate the algorithms in 9 maps of SMAC with various difficulties, 14 tasks of 6 scenarios in MA-MuJoCo, and the 5-vs-5 and 11-vs-11 full game scenarios in GRF. Results in Tab. 2, Fig. 3, and Fig. 4 show that A2PO consistently outperforms the baselines and achieves higher sample efficiency in all benchmarks. More results and the experimental setups can be found in Appx. B.2.

---

[3]We evaluate A2PO in fully cooperative and general-sum MPE tasks respectively, showing the potential of extending A2PO to general-sum games, see Appx. B.2.3 for full results.

[4]Code is available at `https://anonymous.4open.science/r/A2PO`.

**StarCraftII Multi-agent Challenge (SMAC)**. As shown in Tab. 2, A2PO achieves (nearly) 100% win rates in 6 out of 9 maps and significantly outperforms other baselines in most maps. In Tab. 2, we additionally compare the performance with that of Qmix (Rashid et al., 2018), a well known baseline in SMAC. We also observe that CoPPO and A2PO have better stability as they consider clipping joint policy ratios.

Table 2: Median win rates and standard deviations on SMAC tasks.

| Map | Difficulty | MAPPO w/ PS | CoPPO w/ PS | HAPPO w/ PS | A2PO w/ PS | Qmix w/ PS |
|---|---|---|---|---|---|---|
| MMM | Easy | 96.9$_{(0.988)}$ | 96.9$_{(1.25)}$ | 95.3$_{(2.48)}$ | **100$_{(1.07)}$** | 95.3$_{(5.2)}$ |
| 3s5z | Hard | 84.4$_{(4.39)}$ | 92.2$_{(2.35)}$ | 92.2$_{(1.74)}$ | **98.4$_{(1.04)}$** | 88.3$_{(2.9)}$ |
| 5m_vs_6m | Hard | 84.4$_{(2.77)}$ | 84.4$_{(2.12)}$ | 87.5$_{(2.51)}$ | **90.6$_{(3.06)}$** | 75.8$_{(3.7)}$ |
| 8m_vs_9m | Hard | 84.4$_{(2.39)}$ | 84.4$_{(2.04)}$ | 96.9$_{(3.78)}$ | **100$_{(1.04)}$** | 92.2$_{(2.0)}$ |
| 10m_vs_11m | Hard | 93.8$_{(18.7)}$ | 96.9$_{(2.6)}$ | 98.4$_{(2.99)}$ | **100$_{(0.521)}$** | 95.3$_{(1.0)}$ |
| 6h_vs_8z | Super Hard | 87.5$_{(1.53)}$ | **90.6$_{(0.765)}$** | 87.5$_{(1.49)}$ | **90.6$_{(1.32)}$** | 9.4$_{(2.0)}$ |
| 3s5z_vs_3s6z | Super Hard | 82.8$_{(19.2)}$ | 84.4$_{(2.9)}$ | 37.5$_{(13.2)}$ | **93.8$_{(19.8)}$** | 82.8$_{(5.3)}$ |
| MMM2 | Super Hard | 90.6$_{(8.89)}$ | 90.6$_{(6.93)}$ | 51.6$_{(9.01)}$ | **98.4$_{(1.25)}$** | 87.5$_{(2.6)}$ |
| 27m_vs_30m | Super Hard | 93.8$_{(3.75)}$ | 93.8$_{(2.2)}$ | 90.6$_{(4.77)}$ | **100$_{(1.55)}$** | 39.1$_{(9.8)}$ |
| Overall | / | 88.7$_{(6.96)}$ | 90.5$_{(2.57)}$ | 81.9$_{(4.67)}$ | **96.9$_{(3.41)}$** | 74.0$_{(3.83)}$ |

**Multi-agent MuJoCo environment (MA-MuJoCo)**. We investigate whether A2PO can scale to more complex continuous control multi-agent tasks in MA-MuJoCo. We calculate the normalized score $\frac{\text{return} - \text{minimum return}}{\text{maximum return} - \text{minimum return}}$ over all the 14 tasks in the left of Fig. 3. We also present part of results in the right of Fig. 3, where the control complexity and observation dimension, depending on the number of the robot's joints, increases from left to right. We observe that A2PO generally shows an increasing advantage over the baselines with increasing task complexity.

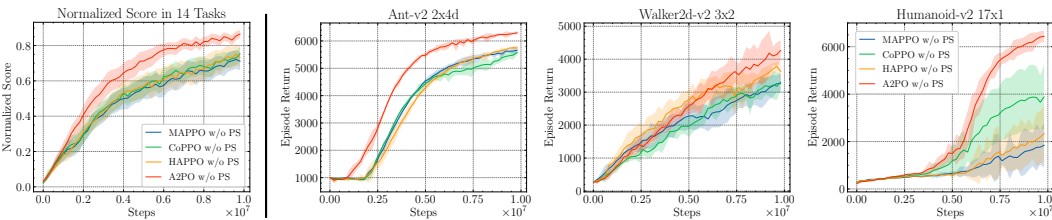

Figure 3: Experiments in MA-MuJoCo. **Left**: Normalized scores on all the 14 tasks. **Right**: Comparisons of averaged return on selected tasks. The number of robot joints increases from left to right.

**Google Research Football (GRF)**. We evaluate A2PO in GRF full-game scenarios, where agents have difficulty discovering complex coordination behaviors. A2PO obtains nearly 100% win rate in the 5-vs-5 scenario. In both scenarios, we attribute the performance gain of A2PO to the learned coordination behavior. We analyze the experiments in GRF to verify that A2PO encourages agents to learn coordination behaviors in complex tasks. In Tab. 3, an 'Assist' is attributed to the player who passes the ball to the teammate that makes a score, a 'Pass' is counted when the passing-and-receiving process is finished, 'Pass Rate' is the proportion of success passes over the pass attempts. A2PO have an advantage in passing-and-receiving coordination, leading to more assists and scores.

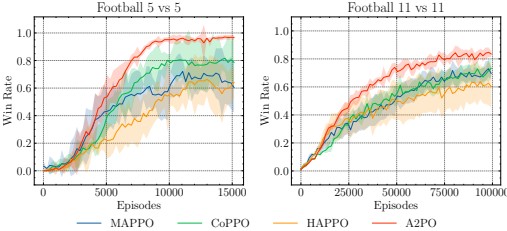

Figure 4: Averaged win rate on the Google Research Football full-game scenarios.

Table 3: Learned behaviors on the Google Research Football 5-vs-5 scenario. Bigger values are better except fot the 'Lost' metric.

| Metric | MAPPO | CoPPO | HAPPO | A2PO |
|---|---|---|---|---|
| Assist | 0.04$_{(0.02)}$ | 0.19$_{(0.08)}$ | 0.07$_{(0.05)}$ | **0.56$_{(0.20)}$** |
| Goal | 1.95$_{(1.17)}$ | 4.42$_{(2.08)}$ | 2.68$_{(0.86)}$ | **9.01$_{(0.95)}$** |
| Lost | **0.49$_{(0.11)}$** | 0.74$_{(0.33)}$ | 1.04$_{(0.12)}$ | 0.78$_{(0.15)}$ |
| Pass | 1.52$_{(0.13)}$ | 3.44$_{(1.04)}$ | 4.03$_{(1.97)}$ | **6.42$_{(2.23)}$** |
| Pass Rate | 19.3$_{(10.0)}$ | 35.0$_{(10.3)}$ | 48.9$_{(25.7)}$ | **67.1$_{(11.7)}$** |

## 5.2 ABLATION STUDY

This section studies how PreOPC, the semi-greedy agent selection rule, and the adaptive clipping parameter affect the performance. Full ablation details can be found in Appx. B.2.5

**PreOPC**. Fig. 5 shows the effects of utilizing off-policy correction in two cases: 1) Correction on all agents' policies for simultaneous update algorithms, i.e., MAPPO w/ V-trace (Espeholt et al., 2018) and CoPPO w/ V-trace, and 2) Correction on the preceding agents' policies for sequential update algorithms, i.e., HAPPO w/ PreOC and A2PO. V-trace brings no general improvement to MAPPO and CoPPO, while PreOPC significantly improves the sequential update cases. PreOPC improves the performance of HAPPO significantly, while A2PO still outperforms HAPPO w/ PreOPC. The per-

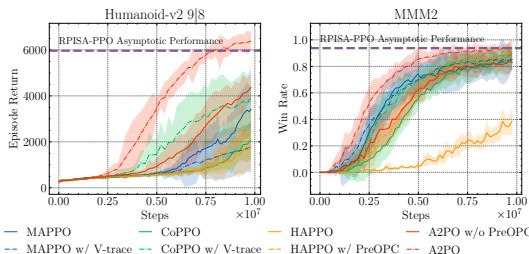

Figure 5: Ablation experiments on preceding-agent off-policy correction.

formance gap lies in that A2PO clips the joint policy ratios, which matches the monotonic bound in Thm. 1. The results verify that A2PO reaches or outperforms the asymptotic performance of RPISA-PPO using an approximated advantage function and updating all the agents with the same rollout samples. Additionally, preceding-agent off-policy correction does not increase the sensitivity of the hyper-parameter $\lambda$, as shown in Appx. B.2.5.

**Agent Selection Rule**. We provide comparisons of different agent selection rules in Fig. 6. The 'Cyclic' rule means select agents in the order $1, \ldots, n$, and other rules have been introduced in sec. 4. The semi-greedy rule considers the optimization acceleration and the performance balance among agents and thus performs the best in all tasks.

**Adaptive Clipping Parameter**. We propose the adaptive clipping parameter method for balanced and sufficient clipping ranges of agents. As shown in Fig. 7, the adaptive clipping parameter contributes to the performance gain of A2PO.

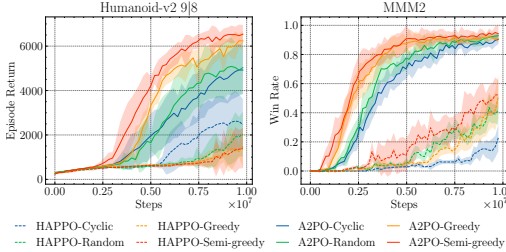

Figure 6: Ablation experiments on the agent selection rules.

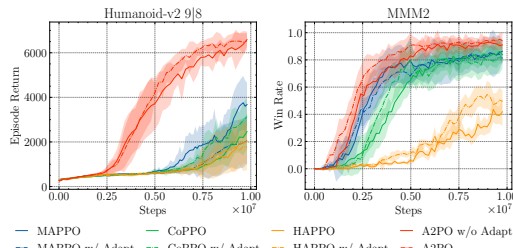

Figure 7: Ablation experiments on the adaptive clipping parameter method.

## 6 CONCLUSION

In this paper, we investigate the potential of the sequential update scheme in coordination tasks. We introduce A2PO, a sequential algorithm using a single rollout at a stage, which guarantees monotonic improvement on both the joint policy and each agent's policy. We also justify that the monotonic bound achieved by A2PO is the tightest among existing trust region MARL algorithms under single rollout scheme. Furthermore, A2PO integrates the proposed semi-greedy agent selection rule and adaptive clipping parameter method. Experiments in various benchmarks demonstrate that A2PO consistently outperforms state-of-the-art methods in performance and sample efficiency and encourages coordination behaviors for completing complex tasks. For future work, we plan to analyze the theoretical underpinnings of the agent selection rules and study the learnable methods to select agents and clipping parameters.

**Acknowledgements.** The SJTU team is partially supported by "New Generation of AI 2030" Major Project (2018AAA0100900), the Shanghai Municipal Science and Technology Major Project (2021SHZDZX0102), the Shanghai Sailing Program (21YF1421900), the National Natural Science Foundation of China (62076161, 62106141). Xihuai Wang and Ziyu Wan are supported by Wu Wen Jun Honorary Scholarship, AI Institute, Shanghai Jiao Tong University. We thank Yan Song and He Jiang for their help in the football experiments.

**Ethics Statement.** Our method and algorithm do not involve any adversarial attack, and will not endanger human security. All our experiments are performed in the simulation environment, which does not involve ethical and fair issues.

**Reproducibility Statement.** The source code of this paper is available at `https://anonymous.4open.science/r/A2PO`. We provide proofs in appx. A, including the proofs of intuitive sequential update, monotonic policy improvement of A2PO, incrementally tightened bound of A2PO and monotonic policy improvement of MAPPO, CoPPO and HAPPO. We specify all the experiments implementation details, the experiments setup, and the additional results in the appx. B. The related works of coordinate descent are shown in appx. D.

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

# Supplementary Material

## Table of Contents

# A PROOFS

## A.1 NOTATIONS

We list the main notations used in Tab. 4.

Table 4: The notations and symbols used in this paper.

| Notation | Definition |
|---|---|
| $\mathcal{S}$ | The state space |
| $\mathcal{N}$ | The set of agents |
| $n$ | The number of agents |
| $i$ | The agent index |
| $\mathcal{A}^i$ | The action space of agent $i$ |
| $r$ | The reward function |
| $\mathcal{T}$ | The transition function |
| $\gamma$ | The discount factor |
| $t$ | The time-step |
| $s_t$ | The state at time-step $t$ |
| $a_t^i$ | The action of agent $i$ at time-step $t$ |
| $\boldsymbol{a}_t$ | The joint action at time-step $t$ |
| $d^{\boldsymbol{\pi}}$ | The discounted state visitation distribution |
| $Pr$ | The state probability function |
| $V$ | The value function |
| $A$ | The advantage function |
| $\tau$ | The trajectory of an episode |
| $e$ | A set of preceding agents |
| $e^i$ | The set of preceding agents updated before agent $i$ |
| $\pi^i$ | The policy of agent $i$ |
| $\bar{\pi}^i$ | The updated policy of agent $i$ |
| $\boldsymbol{\pi}$ | The joint policy |
| $\lambda$ | The bias and variance balance parameter |
| $\bar{\boldsymbol{\pi}}$ | The joint target policy |
| $\hat{\boldsymbol{\pi}}^i$ | The joint policy after updating agent $i$ |
| $\mathcal{J}(\boldsymbol{\pi})$ | The expected return / performance of the joint policy $\boldsymbol{\pi}$ |
| $\mathcal{L}_{\hat{\boldsymbol{\pi}}^{i-1}}(\hat{\boldsymbol{\pi}}^i)$ | The surrogate objective of agent $i$ |
| $\mathcal{L}_{\hat{\boldsymbol{\pi}}^{i-1}}^I(\hat{\boldsymbol{\pi}}^i)$ | An intuitive surrogate objective of agent $i$ |
| $\mathcal{G}_{\boldsymbol{\pi}}(\bar{\boldsymbol{\pi}})$ | The surrogate objective of all agents |
| $\epsilon$ | The upper bound of an advantage function |
| $D_{TV}$ | The total variation distance function |
| $\alpha$ | The total variation distance between 2 policies |
| $\xi^i$ | The off policy correction error of $\hat{\boldsymbol{\pi}}^{i-1}$ |
| $\mathcal{C}$ | The clipping parameter adaptation function |
| $\mathcal{R}$ | The agent selection function |

## A.2 USEFUL LEMMAS

**Lemma 1** *(Multi-agent Policy Performance Difference Lemma). Given any joint policies $\bar{\boldsymbol{\pi}}$ and $\boldsymbol{\pi}$, the difference between the performance of two joint policies can be expressed as:*

$$\mathcal{J}(\bar{\boldsymbol{\pi}}) - \mathcal{J}(\boldsymbol{\pi}) = \frac{1}{1-\gamma}\mathbb{E}_{(s,\boldsymbol{a})\sim(d^{\bar{\boldsymbol{\pi}}},\bar{\boldsymbol{\pi}})}\left[A^{\boldsymbol{\pi}}(s,\boldsymbol{a})\right] \ ,$$

*where $d^{\pi} = (1-\gamma)\sum_{t=0}^{\infty}\gamma^t Pr(s_t = s|\pi)$ is the normalized discounted state visitation distribution.*

*Proof.* A corollary of the Policy Performance Difference Lemma, see Lemma 1.16 in Alekh et al. (2022). $\square$

For convenience, we give some properties and definitions of coupling[5] and the definition of $\alpha$-coupled policy pair (Schulman et al., 2015) here.

---

[5]The definition of coupling and the properties can be found in any textbook containing Markov Chains.

**Definition 1 (*Coupling*)** *A coupling of two probability distributions $\mu$ and $\nu$ is a pair of random variables $(X, Y)$ such that the marginal distribution of $X$ is $\mu$ and the marginal distribution of $Y$ is $\nu$. A coupling $(X, Y)$ satisfies the following constraints: $Pr(X = x) = \mu(x)$ and $Pr(Y = y) = \nu(y)$.*

**Proposition 3** *For any coupling $(X, Y)$ that $D_{TV}(\mu\|\nu) \leq Pr(X \neq Y)$.*

**Proposition 4** *There exists a coupling $(X, Y)$ that $D_{TV}(\mu\|\nu) = Pr(X \neq Y)$.*

**Corollary 1** *For all $s$, there exists a coupling $(\boldsymbol{\pi}(\cdot|s), \bar{\boldsymbol{\pi}}(\cdot|s))$, that $Pr(\boldsymbol{a} = \bar{\boldsymbol{a}}) \geq 1 - D_{TV}^{max}(\boldsymbol{\pi}\|\bar{\boldsymbol{\pi}})$, for $\boldsymbol{a} \sim \boldsymbol{\pi}(\cdot|s)$, $\bar{\boldsymbol{a}} \sim \bar{\boldsymbol{\pi}}(\cdot|s)$.*

*Proof.* By prop. 4 there exists a coupling $(\boldsymbol{\pi}(\cdot|s), \bar{\boldsymbol{\pi}}(\cdot|s))$, s.t.

$$1 - Pr(\boldsymbol{a} = \bar{\boldsymbol{a}}) = Pr(\boldsymbol{a} \neq \bar{\boldsymbol{a}}) = D_{TV}(\boldsymbol{\pi}, \bar{\boldsymbol{\pi}}) \leq D_{TV}^{max}(\boldsymbol{\pi}\|\bar{\boldsymbol{\pi}})$$

$\square$

**Corollary 2** *For all $s$, $D_{TV}(\boldsymbol{\pi}(\cdot|s)\|\bar{\boldsymbol{\pi}}(\cdot|s)) \leq \sum_{i=1}^{n} D_{TV}(\pi^i(\cdot|s)\|\bar{\pi}^i(\cdot|s))$.*

*Proof.* We denote $\pi(\cdot|s)$ as $\pi(\cdot)$ for brevity.

$$D_{TV}(\boldsymbol{\pi}(\cdot|s)\|\bar{\boldsymbol{\pi}}(\cdot|s))$$

$$= \frac{1}{2} \sum_{a^1, a^2, \ldots, a^n} \left| \prod_{i=1}^{n} \pi^i(a^i) - \prod_{i=1}^{n} \bar{\pi}^i(a^i) \right|$$

$$= \frac{1}{2} \sum_{a^1, a^2, \ldots, a^n} \left| \prod_{i=1}^{n} \pi^i(a^i) - \pi^1(a^1) \prod_{i=2}^{n} \bar{\pi}^i(a^i) + \pi^1(a^1) \prod_{i=2}^{n} \bar{\pi}^i(a^i) - \prod_{i=1}^{n} \bar{\pi}^i(a^i) \right|$$

$$\leq \frac{1}{2} \sum_{a^1} |\pi^1(a^1)| \sum_{a^2, \ldots, a^n} \left| \prod_{i=2}^{n} \pi^i(a^i) - \prod_{i=2}^{n} \bar{\pi}^i(a^i) \right| + \frac{1}{2} \sum_{a^1} |\pi^1(a^1) - \bar{\pi}^1(a^1)| \sum_{a^2, \ldots, a^n} \left| \prod_{i=2}^{n} \bar{\pi}^i(a^i) \right|$$

$$= \frac{1}{2} \sum_{a^2, \ldots, a^n} \left| \prod_{i=2}^{n} \pi^i(a^i) - \prod_{i=2}^{n} \bar{\pi}^i(a^i) \right| + \frac{1}{2} \sum_{a^1} |\pi^1(a^1) - \bar{\pi}^1(a^1)|$$

$$\ldots$$

$$\leq \frac{1}{2} \sum_{i=1}^{n} \sum_{a^i} |\pi^i(a^i) - \bar{\pi}^i(a^i)|$$

$$= \sum_{i=1}^{n} D_{TV}(\pi^i(\cdot|s)\|\bar{\pi}^i(\cdot|s))$$

$\square$

**Definition 2 ($\alpha$-*coupled policy pair*)** *If $(\boldsymbol{\pi}, \bar{\boldsymbol{\pi}})$ is an $\alpha$-coupled policy pair, then $(\boldsymbol{a}, \bar{\boldsymbol{a}}|s)$ satisfies $Pr(\boldsymbol{a} \neq \bar{\boldsymbol{a}}|s) \leq \alpha$ for all $s$, and $\boldsymbol{a} \sim \boldsymbol{\pi}(\cdot|s)$, $\bar{\boldsymbol{a}} \sim \bar{\boldsymbol{\pi}}(\cdot|s)$.*

From Corollaries 1 and 2, we know that given any joint policy pair $\boldsymbol{\pi}$ and $\bar{\boldsymbol{\pi}}$, select $\alpha = D_{TV}^{max}(\boldsymbol{\pi}(\cdot|s)\|\bar{\boldsymbol{\pi}}(\cdot|s))$, then $(\boldsymbol{\pi}, \bar{\boldsymbol{\pi}})$ is an $\alpha$-coupled policy pair that for all $s$, $Pr(\boldsymbol{a} \neq \bar{\boldsymbol{a}}|s) \leq D_{TV}^{max}(\boldsymbol{\pi}(\cdot|s)\|\bar{\boldsymbol{\pi}}(\cdot|s)) \leq \sum_{i=1}^{n} \alpha^i$, where $\alpha^i = D_{TV}^{max}(\pi^i\|\bar{\pi}^i)$.

**Lemma 2** *Given any joint policies $\bar{\boldsymbol{\pi}}$ and $\boldsymbol{\pi}$, if $(\bar{\boldsymbol{\pi}}, \boldsymbol{\pi})$ is a coupled policy pair, the following inequality holds:*

$$\left| \mathbb{E}_{\boldsymbol{a} \sim \bar{\boldsymbol{\pi}}} \left[ A^{\boldsymbol{\pi}}(s, \boldsymbol{a}) \right] \right| \leq 2\epsilon \sum_{i=1}^{n} \alpha^i \,,$$

*where $\alpha^i = D_{TV}^{max}(\bar{\pi}^i\|\pi^i)$ and $\epsilon = \max_{s,\boldsymbol{a}} |A^{\boldsymbol{\pi}}(s, \boldsymbol{a})|$.*

*Proof.* Note that $\mathbb{E}_{\boldsymbol{a} \sim \boldsymbol{\pi}}[A^{\boldsymbol{\pi}}(s, \boldsymbol{a})] = 0$. We have

$$
\begin{aligned}
|\mathbb{E}_{\boldsymbol{a} \sim \bar{\boldsymbol{\pi}}}\left[A^{\boldsymbol{\pi}}(s, \boldsymbol{a})\right]| &= |\mathbb{E}_{\bar{\boldsymbol{a}} \sim \bar{\boldsymbol{\pi}}}\left[A^{\boldsymbol{\pi}}(s, \bar{\boldsymbol{a}})\right] - \mathbb{E}_{\boldsymbol{a} \sim \boldsymbol{\pi}}\left[A^{\boldsymbol{\pi}}(s, \boldsymbol{a})\right]| \\
&= \left|\mathbb{E}_{(\bar{\boldsymbol{a}}, \boldsymbol{a}) \sim (\bar{\boldsymbol{\pi}}, \boldsymbol{\pi})}\left[A^{\boldsymbol{\pi}}(s, \bar{\boldsymbol{a}}) - A^{\boldsymbol{\pi}}(s, \boldsymbol{a})\right]\right| \\
&= \left|Pr(\bar{\boldsymbol{a}} \neq \boldsymbol{a}|s)\mathbb{E}_{(\bar{\boldsymbol{a}}, \boldsymbol{a}) \sim (\bar{\boldsymbol{\pi}}, \boldsymbol{\pi})}\left[A^{\boldsymbol{\pi}}(s, \bar{\boldsymbol{a}}) - A^{\boldsymbol{\pi}}(s, \boldsymbol{a})\right]\right| \\
&\leq \sum_{i=1}^{n} \alpha^i \mathbb{E}_{(\bar{\boldsymbol{a}}, \boldsymbol{a}) \sim (\bar{\boldsymbol{\pi}}, \boldsymbol{\pi})}\left[|A^{\boldsymbol{\pi}}(s, \bar{\boldsymbol{a}}) - A^{\boldsymbol{\pi}}(s, \boldsymbol{a})|\right] \\
&\leq \sum_{i=1}^{n} \alpha^i \cdot 2 \max_{s, \boldsymbol{a}} |A^{\boldsymbol{\pi}}(s, \boldsymbol{a})|
\end{aligned}
$$

$\square$

**Lemma 3** *(Multi-agent Advantage Discrepancy Lemma). Given any joint policies $\boldsymbol{\pi}^1$, $\boldsymbol{\pi}^2$ and $\boldsymbol{\pi}^3$, if $(\boldsymbol{\pi}^1, \boldsymbol{\pi}^2)$ and $(\boldsymbol{\pi}^2, \boldsymbol{\pi}^3)$ are coupled policy pairs, the following inequality holds:*

$$
\begin{aligned}
&\left|\mathbb{E}_{(s_t, \boldsymbol{a}_t) \sim (Pr^{\boldsymbol{\pi}^2}, \boldsymbol{\pi}^2)}\left[A^{\boldsymbol{\pi}^1}\right] - \mathbb{E}_{(s_t, \bar{\boldsymbol{a}}_t) \sim (Pr^{\boldsymbol{\pi}^3}, \boldsymbol{\pi}^2)}\left[A^{\boldsymbol{\pi}^1}\right]\right| \\
&\leq 4\epsilon^{\boldsymbol{\pi}^1} \cdot D_{TV}^{max}(\boldsymbol{\pi}^1 \| \boldsymbol{\pi}^2) \cdot (1 - (1 - D_{TV}^{max}(\boldsymbol{\pi}^2 \| \boldsymbol{\pi}^3))^t),
\end{aligned}
$$

*where $\epsilon^{\boldsymbol{\pi}^1} = \max_{s, \boldsymbol{a}} \|A^{\boldsymbol{\pi}^1}(s, \boldsymbol{a})\|$ and we denote $A(s, \boldsymbol{a})$ as $A$ for brevity.*

*Proof.* Let $n_t$ represent the times $\boldsymbol{a} \neq \bar{\boldsymbol{a}}$ ($\boldsymbol{\pi}^1$ disagrees with $\boldsymbol{\pi}^3$) before timestamp $t$.

$$
\begin{aligned}
&\left|\mathbb{E}_{(s_t, \boldsymbol{a}_t) \sim (Pr^{\boldsymbol{\pi}^2}, \boldsymbol{\pi}^2)}\left[A^{\boldsymbol{\pi}^1}\right] - \mathbb{E}_{(s_t, \bar{\boldsymbol{a}}_t) \sim (Pr^{\boldsymbol{\pi}^3}, \boldsymbol{\pi}^2)}\left[A^{\boldsymbol{\pi}^1}\right]\right| \\
&= Pr(n_t > 0) \cdot \left|\mathbb{E}_{(s_t, \boldsymbol{a}_t) \sim (Pr^{\boldsymbol{\pi}^2}, \boldsymbol{\pi}^2)|n_t > 0}\left[A^{\boldsymbol{\pi}^1}\right] - \mathbb{E}_{(s_t, \bar{\boldsymbol{a}}_t) \sim (Pr^{\boldsymbol{\pi}^3}, \boldsymbol{\pi}^2)|n_t > 0}\left[A^{\boldsymbol{\pi}^1}\right]\right| \\
&\overset{(a)}{=} (1 - Pr(n_t = 0)) \cdot E \\
&\leq (1 - \prod_{k=1}^{t} Pr(\boldsymbol{a}_k = \bar{\boldsymbol{a}}_k | \boldsymbol{a}_k \sim \boldsymbol{\pi}^2(\cdot|s_k), \bar{\boldsymbol{a}}_k \sim \boldsymbol{\pi}^3(\cdot|s_k))) \cdot E \\
&\overset{(b)}{\leq} (1 - \prod_{k=1}^{t} (1 - D_{TV}^{max}(\boldsymbol{\pi}^2 \| \boldsymbol{\pi}^3))) \cdot E \\
&= (1 - (1 - D_{TV}^{max}(\boldsymbol{\pi}^2 \| \boldsymbol{\pi}^3))^t) \cdot E \\
&\leq (1 - (1 - D_{TV}^{max}(\boldsymbol{\pi}^2 \| \boldsymbol{\pi}^3))^t) \cdot 2 \cdot 2 \cdot D_{TV}^{max}(\boldsymbol{\pi}^1 \| \boldsymbol{\pi}^2) \cdot \epsilon^{\boldsymbol{\pi}^1} \\
&= 4\epsilon^{\boldsymbol{\pi}^1} \cdot D_{TV}^{max}(\boldsymbol{\pi}^1 \| \boldsymbol{\pi}^2) \cdot (1 - (1 - D_{TV}^{max}(\boldsymbol{\pi}^2 \| \boldsymbol{\pi}^3))^t)
\end{aligned}
$$

In (a), we denote $|\mathbb{E}_{(s_t, \boldsymbol{a}_t) \sim (Pr^{\boldsymbol{\pi}^2}, \boldsymbol{\pi}^2)|n_t > 0}[A^{\boldsymbol{\pi}^1}] - \mathbb{E}_{(s_t, \bar{\boldsymbol{a}}_t) \sim (Pr^{\boldsymbol{\pi}^3}, \boldsymbol{\pi}^2)|n_t > 0}[A^{\boldsymbol{\pi}^1}]|$ as $E$ for brevity. (b) follows the definition of $\alpha$-coupled policy pair. $\square$

We provide a useful equation of the normalized discounted state visitation distribution here.

**Proposition 5**

$$
\begin{aligned}
\mathbb{E}_{(s, \boldsymbol{a}) \sim (d^{\boldsymbol{\pi}^1}, \boldsymbol{\pi}^2)}[f(s, \boldsymbol{a})] &= (1 - \gamma) \sum_{s} \sum_{t=0}^{\infty} \gamma^t Pr(s_t = s|\boldsymbol{\pi}^1) \sum_{\boldsymbol{a}} \boldsymbol{\pi}^2(\boldsymbol{a}|s) f(s, \boldsymbol{a}) \\
&= (1 - \gamma) \sum_{t=0}^{\infty} \gamma^t \sum_{s} Pr(s_t = s|\boldsymbol{\pi}^1) \sum_{\boldsymbol{a}} \boldsymbol{\pi}^2(\boldsymbol{a}|s) f(s, \boldsymbol{a}) \\
&= (1 - \gamma) \sum_{t=0}^{\infty} \gamma^t \mathbb{E}_{(s_t, \boldsymbol{a}_t) \sim (Pr^{\boldsymbol{\pi}^1}, \boldsymbol{\pi}^2)}[f(s_t, \boldsymbol{a}_t)]
\end{aligned}
$$

### A.3 PROOFS OF INTUITIVE SEQUENTIAL UPDATE

$$
\left| \mathcal{J}(\hat{\boldsymbol{\pi}}^i) - \mathcal{J}(\hat{\boldsymbol{\pi}}^{i-1}) - \frac{1}{1-\gamma} \mathbb{E}_{(s,\boldsymbol{a}) \sim (d^{\boldsymbol{\pi}}, \hat{\boldsymbol{\pi}}^i)} \left[ A^{\boldsymbol{\pi}} \right] \right|
$$

$$
\leq \frac{1}{1-\gamma} \left| \mathbb{E}_{(s,\boldsymbol{a}) \sim (d^{\tilde{\boldsymbol{\pi}}^i}, \hat{\boldsymbol{\pi}}^i)} \left[ A^{\hat{\boldsymbol{\pi}}^{i-1}} \right] - \mathbb{E}_{(s,\boldsymbol{a}) \sim (d^{\boldsymbol{\pi}}, \hat{\boldsymbol{\pi}}^i)} \left[ A^{\boldsymbol{\pi}} \right] \right|
$$

$$
\leq \frac{1}{1-\gamma} \left| \mathbb{E}_{(s,\boldsymbol{a}) \sim (d^{\tilde{\boldsymbol{\pi}}^i}, \hat{\boldsymbol{\pi}}^i)} \left[ A^{\hat{\boldsymbol{\pi}}^{i-1}} \right] - \mathbb{E}_{(s,\boldsymbol{a}) \sim (d^{\boldsymbol{\pi}}, \hat{\boldsymbol{\pi}}^i)} \left[ A^{\hat{\boldsymbol{\pi}}^{i-1}} \right] \right|
$$

$$
+ \frac{1}{1-\gamma} \left| \mathbb{E}_{(s,\boldsymbol{a}) \sim (d^{\boldsymbol{\pi}}, \hat{\boldsymbol{\pi}}^i)} \left[ A^{\hat{\boldsymbol{\pi}}^{i-1}} \right] - \mathbb{E}_{(s,\boldsymbol{a}) \sim (d^{\boldsymbol{\pi}}, \hat{\boldsymbol{\pi}}^i)} \left[ A^{\boldsymbol{\pi}} \right] \right|
$$

$$
\leq 4\epsilon^{\hat{\boldsymbol{\pi}}^{i-1}} \alpha^i \sum_{t=0}^{\infty} \gamma^t (1 - (1 - \sum_{j \in (e^i \cup \{i\})} \alpha^j)^t)
$$

$$
+ \frac{1}{1-\gamma} \mathbb{E}_{(s,\boldsymbol{a}) \sim (d^{\boldsymbol{\pi}}, \hat{\boldsymbol{\pi}}^i)} \left[ \left| A^{\hat{\boldsymbol{\pi}}^{i-1}} - A^{\boldsymbol{\pi}} \right| \right]
$$

$$
\leq 4\epsilon^{\hat{\boldsymbol{\pi}}^{i-1}} \alpha^i \Big( \frac{1}{1-\gamma} - \frac{1}{1 - \gamma(1 - \sum_{j \in (e^i \cup \{i\})} \alpha^j)} \Big) + \frac{1}{1-\gamma} \left[ 4\alpha^i \epsilon^{\hat{\boldsymbol{\pi}}^{i-1}} + 2 \sum_{j \in e^i} \alpha^j \epsilon^{\boldsymbol{\pi}} \right]
$$

### A.4 PROOFS OF MONOTONIC POLICY IMPROVEMENT OF A2PO

**Theorem 1 (Single Agent Monotonic Bound)** *For agent $i$, let $\epsilon^i = \max_{s,\boldsymbol{a}} |A^{\hat{\boldsymbol{\pi}}^{i-1}}(s,\boldsymbol{a})|$, $\xi^i = \max_{s,\boldsymbol{a}} |A^{\boldsymbol{\pi}, \hat{\boldsymbol{\pi}}^{i-1}}(s,\boldsymbol{a}) - A^{\hat{\boldsymbol{\pi}}^{i-1}}(s,\boldsymbol{a})|$, $\alpha^j = D_{TV}^{\max}(\pi^j \| \bar{\pi}^j) \ \forall j \in (e^i \cup \{i\})$, then we have:*

$$
\left| \mathcal{J}(\hat{\boldsymbol{\pi}}^i) - \mathcal{L}_{\hat{\boldsymbol{\pi}}^{i-1}}(\hat{\boldsymbol{\pi}}^i) \right| \leq 4\epsilon^i \alpha^i \Big( \frac{1}{1-\gamma} - \frac{1}{1 - \gamma(1 - \sum_{j \in (e^i \cup \{i\})} \alpha^j)} \Big) + \frac{\xi^i}{1-\gamma}
$$

$$
\leq \frac{4\gamma\epsilon^i}{(1-\gamma)^2} \Big( \alpha^i \sum_{j \in (e^i \cup \{i\})} \alpha^j \Big) + \frac{\xi^i}{1-\gamma} . \tag{4}
$$

*Proof.* Using Lemma 3 and Prop. 5, we get

$$
\left| \mathcal{J}(\hat{\boldsymbol{\pi}}^i) - \mathcal{J}(\hat{\boldsymbol{\pi}}^{i-1}) - \frac{1}{1-\gamma} \mathbb{E}_{(s,\boldsymbol{a}) \sim (d^{\boldsymbol{\pi}}, \hat{\boldsymbol{\pi}}^i)} \left[ A^{\boldsymbol{\pi}, \hat{\boldsymbol{\pi}}^{i-1}} \right] \right|
$$

$$
= \frac{1}{1-\gamma} \left| \mathbb{E}_{(s,\boldsymbol{a}) \sim (d^{\tilde{\boldsymbol{\pi}}^i}, \hat{\boldsymbol{\pi}}^i)} \left[ A^{\hat{\boldsymbol{\pi}}^{i-1}} \right] - \mathbb{E}_{(s,\boldsymbol{a}) \sim (d^{\boldsymbol{\pi}}, \hat{\boldsymbol{\pi}}^i)} \left[ A^{\boldsymbol{\pi}, \hat{\boldsymbol{\pi}}^{i-1}} \right] \right|
$$

$$
\leq \frac{1}{1-\gamma} \left| \mathbb{E}_{(s,\boldsymbol{a}) \sim (d^{\tilde{\boldsymbol{\pi}}^i}, \hat{\boldsymbol{\pi}}^i)} \left[ A^{\hat{\boldsymbol{\pi}}^{i-1}} \right] - \mathbb{E}_{(s,\boldsymbol{a}) \sim (d^{\boldsymbol{\pi}}, \hat{\boldsymbol{\pi}}^i)} \left[ A^{\hat{\boldsymbol{\pi}}^{i-1}} \right] \right|
$$

$$
+ \frac{1}{1-\gamma} \left| \mathbb{E}_{(s,\boldsymbol{a}) \sim (d^{\boldsymbol{\pi}}, \hat{\boldsymbol{\pi}}^i)} \left[ A^{\hat{\boldsymbol{\pi}}^{i-1}} \right] - \mathbb{E}_{(s,\boldsymbol{a}) \sim (d^{\boldsymbol{\pi}}, \hat{\boldsymbol{\pi}}^i)} \left[ A^{\boldsymbol{\pi}, \hat{\boldsymbol{\pi}}^{i-1}} \right] \right|
$$

$$
\leq 4\epsilon^{\hat{\boldsymbol{\pi}}^{i-1}} \alpha^i \sum_{t=0}^{\infty} \gamma^t (1 - (1 - \sum_{j \in (e^i \cup \{i\})} \alpha^j)^t) + \frac{1}{1-\gamma} \mathbb{E}_{(s,\boldsymbol{a}) \sim (d^{\boldsymbol{\pi}}, \hat{\boldsymbol{\pi}}^i)} \left[ \left| A^{\hat{\boldsymbol{\pi}}^{i-1}} - A^{\boldsymbol{\pi}, \hat{\boldsymbol{\pi}}^{i-1}} \right| \right]
$$

$$
\leq 4\epsilon^{\hat{\boldsymbol{\pi}}^{i-1}} \alpha^i \Big( \frac{1}{1-\gamma} - \frac{1}{1 - \gamma(1 - \sum_{j \in (e^i \cup \{i\})} \alpha^j)} \Big) + \frac{1}{1-\gamma} \xi^i
$$

$\square$

**Theorem 2 (Joint Monotonic Bound)** *For each agent* $i \in \mathcal{N}$, *let* $\epsilon^i = \max_{s,\boldsymbol{a}} |A^{\hat{\boldsymbol{\pi}}^{i-1}}(s,\boldsymbol{a})|$, $\alpha^i = D_{TV}^{\max}(\pi^i \| \bar{\pi}^i)$, $\xi^i = \max_{s,\boldsymbol{a}} |A^{\boldsymbol{\pi},\hat{\boldsymbol{\pi}}^{i-1}}(s,\boldsymbol{a}) - A^{\hat{\boldsymbol{\pi}}^{i-1}}(s,\boldsymbol{a})|$, *and* $\epsilon = \max_i \epsilon^i$, *then we have:*

$$|\mathcal{J}(\bar{\boldsymbol{\pi}}) - \mathcal{G}_{\boldsymbol{\pi}}(\bar{\boldsymbol{\pi}})| \leq 4\epsilon \sum_{i=1}^{n} \alpha^i \Big( \frac{1}{1-\gamma} - \frac{1}{1 - \gamma(1 - \sum_{j \in (e^i \cup \{i\})} \alpha^j)} \Big) + \frac{\sum_{i=1}^{n} \xi^i}{1-\gamma}$$

$$\leq \frac{4\gamma\epsilon}{(1-\gamma)^2} \sum_{i=1}^{n} \Big( \alpha^i \sum_{j \in (e^i \cup \{i\})} \alpha^j \Big) + \frac{\sum_{i=1}^{n} \xi^i}{1-\gamma} . \tag{5}$$

*Proof.*

$$|\mathcal{J}(\bar{\boldsymbol{\pi}}) - \mathcal{G}_{\boldsymbol{\pi}}(\bar{\boldsymbol{\pi}})|$$

$$= \left| \mathcal{J}(\bar{\boldsymbol{\pi}}) - \mathcal{J}(\boldsymbol{\pi}) - \sum_{i=1}^{n} \mathbb{E}_{(s,\boldsymbol{a}) \sim (d^{\boldsymbol{\pi}}, \hat{\boldsymbol{\pi}}^i)} \left[ A^{\boldsymbol{\pi}, \hat{\boldsymbol{\pi}}^{i-1}}(s,\boldsymbol{a}) \right] \right|$$

$$= \left| \mathcal{J}(\hat{\boldsymbol{\pi}}^n) - \mathcal{J}(\hat{\boldsymbol{\pi}}^{n-1}) + \cdots + \mathcal{J}(\hat{\boldsymbol{\pi}}^1) - \mathcal{J}(\hat{\boldsymbol{\pi}}^0) - \frac{1}{1-\gamma} \sum_{i=1}^{n} \mathbb{E}_{(s,\boldsymbol{a}) \sim (d^{\boldsymbol{\pi}}, \hat{\boldsymbol{\pi}}^i)} \left[ A^{\boldsymbol{\pi}, \hat{\boldsymbol{\pi}}^{i-1}}(s,\boldsymbol{a}) \right] \right|$$

$$\leq \sum_{i=1}^{n} \left| \mathcal{J}(\hat{\boldsymbol{\pi}}^i) - \mathcal{J}(\hat{\boldsymbol{\pi}}^{i-1}) - \frac{1}{1-\gamma} \mathbb{E}_{(s,\boldsymbol{a}) \sim (d^{\boldsymbol{\pi}}, \hat{\boldsymbol{\pi}}^i)} \left[ A^{\boldsymbol{\pi}, \hat{\boldsymbol{\pi}}^{i-1}}(s,\boldsymbol{a}) \right] \right|$$

$$\leq 4\epsilon \sum_{i=1}^{n} \alpha^i \left( \frac{1}{1-\gamma} - \frac{1}{1 - \gamma(1 - \sum_{j \in (e^i \cup \{i\})} \alpha^j)} \right) + \frac{\sum_{i=1}^{n} \xi^i}{1-\gamma}$$

$$\leq \frac{4\gamma\epsilon}{(1-\gamma)^2} \sum_{i=1}^{n} \left( \alpha^i \sum_{j \in (e^i \cup \{i\})} \alpha^j \right) + \frac{\sum_{i=1}^{n} \xi^i}{1-\gamma} .$$

$\square$

## A.5 PROOFS OF INCREMENTALLY TIGHTENED BOUND OF A2PO

Assume agent $k$ is updated with order $k$ in the sequence $1, \ldots, n$, since $\hat{\boldsymbol{\pi}}^{k-1}$ is known, we have

$$|\mathcal{J}(\bar{\boldsymbol{\pi}}) - \mathcal{G}_{\boldsymbol{\pi}}(\bar{\boldsymbol{\pi}})|$$

$$\leq \sum_{i=1}^{k-1} \left| \mathcal{J}(\hat{\boldsymbol{\pi}}^i) - \mathcal{L}_{\hat{\boldsymbol{\pi}}^{i-1}}(\hat{\boldsymbol{\pi}}^i) \right| + 4\epsilon \sum_{i=k}^{n} \alpha^i \left( \frac{1}{1-\gamma} - \frac{1}{1 - \gamma(1 - \sum_{j \in (e^i \cup \{i\})} \alpha^j)} \right) + \frac{\sum_{i=k}^{n} \xi^i}{1-\gamma}$$

$$\leq \sum_{i=1}^{k-2} \left| \mathcal{J}(\hat{\boldsymbol{\pi}}^i) - \mathcal{L}_{\hat{\boldsymbol{\pi}}^{i-1}}(\hat{\boldsymbol{\pi}}^i) \right| + 4\epsilon \sum_{i=k-1}^{n} \alpha^i \left( \frac{1}{1-\gamma} - \frac{1}{1 - \gamma(1 - \sum_{j \in (e^i \cup \{i\})} \alpha^j)} \right) + \frac{\sum_{i=k-1}^{n} \xi^i}{1-\gamma}$$

$$\vdots$$

$$\leq 4\epsilon \sum_{i=1}^{n} \alpha^i \left( \frac{1}{1-\gamma} - \frac{1}{1 - \gamma(1 - \sum_{j \in (e^i \cup \{i\})} \alpha^j)} \right) + \frac{\sum_{i=1}^{n} \xi^i}{1-\gamma}$$

Thus the condition for improving $\mathcal{J}(\bar{\boldsymbol{\pi}})$ is relaxed during updating agents at a stage.

## A.6 PROOFS OF MONOTONIC POLICY IMPROVEMENT OF MAPPO, COPPO AND HAPPO

In this section, we give proof of the monotonic policy improvement of MAPPO, and unify the formats of the monotonic bounds of CoPPO and HAPPO, without considering the parameter-sharing method.

**MAPPO**. For MAPPO, $\mathcal{L}_{\boldsymbol{\pi}}(\bar{\boldsymbol{\pi}}) = \sum_{i=1}^{n} \mathcal{J}(\boldsymbol{\pi}) + \frac{1}{1-\gamma} \left[ \mathbb{E}_{(s,\boldsymbol{a}) \sim (d^{\boldsymbol{\pi}}, \boldsymbol{\pi})} \left[ \frac{\bar{\pi}^i}{\pi^i} A^{\boldsymbol{\pi}} \right] \right]$. We first prove that for agent $i$, $\mathcal{J}(\bar{\boldsymbol{\pi}}) - \mathcal{J}(\boldsymbol{\pi}) - \frac{1}{1-\gamma} \left[ \mathbb{E}_{(s,\boldsymbol{a}) \sim (d^{\boldsymbol{\pi}}, \boldsymbol{\pi})} \left[ \frac{\bar{\pi}^i}{\pi^i} A^{\boldsymbol{\pi}} \right] \right]$ is bounded.

$$\left| \mathcal{J}(\bar{\boldsymbol{\pi}}) - \mathcal{J}(\boldsymbol{\pi}) - \frac{1}{1-\gamma} \left[ \mathbb{E}_{(s,\boldsymbol{a})\sim(d^{\boldsymbol{\pi}},\boldsymbol{\pi})} \left[ \frac{\bar{\pi}^i}{\pi^i} A^{\boldsymbol{\pi}} \right] \right] \right|$$

$$= \frac{1}{1-\gamma} \left| \mathbb{E}_{(s,\boldsymbol{a})\sim(d^{\bar{\boldsymbol{\pi}}},\bar{\boldsymbol{\pi}})} \left[ A^{\boldsymbol{\pi}} \right] - \mathbb{E}_{(s,\boldsymbol{a})\sim(d^{\boldsymbol{\pi}},\boldsymbol{\pi})} \left[ \frac{\bar{\pi}^i}{\pi^i} A^{\boldsymbol{\pi}} \right] \right|$$

$$= \sum_{t=0}^{\infty} \gamma^t \left| \mathbb{E}_{(s_t,\boldsymbol{a}_t)\sim(Pr^{\bar{\boldsymbol{\pi}}},\bar{\boldsymbol{\pi}})} A^{\boldsymbol{\pi}} - \mathbb{E}_{(s_t,\boldsymbol{a}_t)\sim(Pr^{\boldsymbol{\pi}},\boldsymbol{\pi})} \left[ \frac{\bar{\pi}^i}{\pi^i} A^{\boldsymbol{\pi}} \right] \right|$$

$$\leq \sum_{t=0}^{\infty} 2\gamma^t \left( \left( \sum_{j=1}^{n} \alpha^j \right) \cdot \epsilon^{\boldsymbol{\pi}} + \alpha^i \cdot \epsilon^{\boldsymbol{\pi}} \right)$$

$$= \frac{2\epsilon^{\boldsymbol{\pi}}}{1-\gamma} \left( \alpha^i + \sum_{j=1}^{n} \alpha^j \right)$$

Sum the bounds for all agents and take the average, we get

$$\left| \mathcal{J}(\bar{\boldsymbol{\pi}}) - \mathcal{J}(\boldsymbol{\pi}) - \frac{1}{n} \frac{1}{1-\gamma} \sum_{i=1}^{n} \left[ \mathbb{E}_{(s,\boldsymbol{a})\sim(d^{\boldsymbol{\pi}},\boldsymbol{\pi})} \left[ \frac{\bar{\pi}^i}{\pi^i} A^{\boldsymbol{\pi}} \right] \right] \right|$$

$$\leq \frac{2\epsilon^{\boldsymbol{\pi}}}{1-\gamma} \frac{n+1}{n} \sum_{j=1}^{n} \alpha^j$$

Finally, the monotonic bound for MAPPO is

$$\left| \mathcal{J}(\bar{\boldsymbol{\pi}}) - \mathcal{J}(\boldsymbol{\pi}) - \frac{1}{1-\gamma} \sum_{i=1}^{n} \left[ \mathbb{E}_{(s,\boldsymbol{a})\sim(d^{\boldsymbol{\pi}},\boldsymbol{\pi})} \left[ \frac{\bar{\pi}^i}{\pi^i} A^{\boldsymbol{\pi}} \right] \right] \right|$$

$$\leq \left| \mathcal{J}(\bar{\boldsymbol{\pi}}) - \mathcal{J}(\boldsymbol{\pi}) - \frac{1}{n} \frac{1}{1-\gamma} \sum_{i=1}^{n} \left[ \mathbb{E}_{(s,\boldsymbol{a})\sim(d^{\boldsymbol{\pi}},\boldsymbol{\pi})} \left[ \frac{\bar{\pi}^i}{\pi^i} A^{\boldsymbol{\pi}} \right] \right] \right|$$

$$+ \frac{n-1}{n} \frac{1}{1-\gamma} \left| \sum_{i=1}^{n} \left[ \mathbb{E}_{(s,\boldsymbol{a})\sim(d^{\boldsymbol{\pi}},\boldsymbol{\pi})} \left[ \frac{\bar{\pi}^i}{\pi^i} A^{\boldsymbol{\pi}} \right] \right] \right|$$

$$\leq \frac{2\epsilon^{\boldsymbol{\pi}}}{1-\gamma} \frac{n+1}{n} \sum_{j=1}^{n} \alpha^j + \frac{n-1}{n} \sum_{i=1}^{n} \frac{1}{1-\gamma} \alpha^i \cdot 2\epsilon^{\boldsymbol{\pi}}$$

$$= \frac{4\epsilon^{\boldsymbol{\pi}}}{1-\gamma} \sum_{i=1}^{n} \alpha^i$$

**CoPPO**. We prove the results of CoPPO in a unified and convenient form. For CoPPO, $\mathcal{L}_{\boldsymbol{\pi}}(\bar{\boldsymbol{\pi}}) = \mathcal{J}(\boldsymbol{\pi}) + \frac{1}{1-\gamma} \mathbb{E}_{(s,\boldsymbol{a})\sim(d^{\boldsymbol{\pi}},\bar{\boldsymbol{\pi}})} [A^{\boldsymbol{\pi}}(s,\boldsymbol{a})]$, we prove the bound using Lemma 3.

$$\left| \mathcal{J}(\bar{\boldsymbol{\pi}}) - \mathcal{J}(\boldsymbol{\pi}) - \frac{1}{1-\gamma} \mathbb{E}_{(s,\boldsymbol{a}) \sim (d^{\boldsymbol{\pi}}, \bar{\boldsymbol{\pi}})}[A^{\boldsymbol{\pi}}] \right|$$

$$\leq \frac{1}{1-\gamma} \left| \mathbb{E}_{(s,\boldsymbol{a}) \sim (d^{\bar{\boldsymbol{\pi}}}, \bar{\boldsymbol{\pi}})}[A^{\boldsymbol{\pi}}] - \mathbb{E}_{(s,\boldsymbol{a}) \sim (d^{\boldsymbol{\pi}}, \bar{\boldsymbol{\pi}})}[A^{\boldsymbol{\pi}}] \right|$$

$$\leq \sum_{t=0}^{\infty} \gamma^t \left| \mathbb{E}_{(s,\boldsymbol{a}) \sim (Pr^{\bar{\boldsymbol{\pi}}}, \bar{\boldsymbol{\pi}})}[A^{\boldsymbol{\pi}}] - \mathbb{E}_{(s,\boldsymbol{a}) \sim (Pr^{\boldsymbol{\pi}}, \bar{\boldsymbol{\pi}})}[A^{\boldsymbol{\pi}}] \right|$$

$$\leq 4\epsilon^{\boldsymbol{\pi}} \sum_{t=0}^{\infty} \gamma^t \sum_{i=1}^{n} \alpha^i \left( 1 - (1 - D_{TV}^{max}(\boldsymbol{\pi} \| \bar{\boldsymbol{\pi}}))^t \right)$$

$$\leq 4\epsilon^{\boldsymbol{\pi}} \sum_{i=1}^{n} \alpha^i \left( \frac{1}{1-\gamma} - \frac{1}{1-\gamma(1-\sum_{j=1}^{n} \alpha^j)} \right)$$

**HAPPO**. Following the proof of Lemma 2 in Kuba et al. (2022), we know that HAPPO has the same monotonic improvement bound as that of CoPPO. For the monotonic improvement of a single agent, we formulate the surrogate objective of agent $i$ using HAPPO as $\mathcal{J}(\hat{\boldsymbol{\pi}}^{i-1}) + \frac{1}{1-\gamma} \mathbb{E}_{(s,\boldsymbol{a}) \sim (d^{\boldsymbol{\pi}}, \hat{\boldsymbol{\pi}}^i)}[A^{\boldsymbol{\pi}}(s, \boldsymbol{a})] - \frac{1}{1-\gamma} \mathbb{E}_{(s,\boldsymbol{a}) \sim (d^{\boldsymbol{\pi}}, \hat{\boldsymbol{\pi}}^{i-1})}[A^{\boldsymbol{\pi}}(s, \boldsymbol{a})]$, as shown in Proposition 3 of Kuba et al. (2022). Following the proof of Thm. 1, we get the following inequality.

$$\left| \mathcal{J}(\hat{\boldsymbol{\pi}}^i) - \mathcal{J}(\hat{\boldsymbol{\pi}}^{i-1}) - \frac{1}{1-\gamma} \mathbb{E}_{(s,\boldsymbol{a}) \sim (d^{\boldsymbol{\pi}}, \hat{\boldsymbol{\pi}}^i)}[A^{\boldsymbol{\pi}}] + \frac{1}{1-\gamma} \mathbb{E}_{(s,\boldsymbol{a}) \sim (d^{\boldsymbol{\pi}}, \hat{\boldsymbol{\pi}}^{i-1})}[A^{\boldsymbol{\pi}}(s, \boldsymbol{a})] \right|$$

$$\leq \frac{1}{1-\gamma} \left| \mathbb{E}_{(s,\boldsymbol{a}) \sim (d^{\hat{\boldsymbol{\pi}}^i}, \hat{\boldsymbol{\pi}}^i)} \left[ A^{\hat{\boldsymbol{\pi}}^{i-1}} \right] - \mathbb{E}_{(s,\boldsymbol{a}) \sim (d^{\boldsymbol{\pi}}, \hat{\boldsymbol{\pi}}^i)}[A^{\boldsymbol{\pi}}] \right| + \frac{1}{1-\gamma} \left| \mathbb{E}_{(s,\boldsymbol{a}) \sim (d^{\boldsymbol{\pi}}, \hat{\boldsymbol{\pi}}^{i-1})}[A^{\boldsymbol{\pi}}(s, \boldsymbol{a})] \right|$$

$$\leq \frac{1}{1-\gamma} \left| \mathbb{E}_{(s,\boldsymbol{a}) \sim (d^{\hat{\boldsymbol{\pi}}^i}, \hat{\boldsymbol{\pi}}^i)} \left[ A^{\hat{\boldsymbol{\pi}}^{i-1}} \right] - \frac{1}{1-\gamma} \mathbb{E}_{(s,\boldsymbol{a}) \sim (d^{\boldsymbol{\pi}}, \hat{\boldsymbol{\pi}}^i)} \left[ A^{\hat{\boldsymbol{\pi}}^{i-1}} \right] \right|$$

$$+ \frac{1}{1-\gamma} \left| \mathbb{E}_{(s,\boldsymbol{a}) \sim (d^{\boldsymbol{\pi}}, \hat{\boldsymbol{\pi}}^i)} \left[ A^{\hat{\boldsymbol{\pi}}^{i-1}} \right] - \mathbb{E}_{(s,\boldsymbol{a}) \sim (d^{\boldsymbol{\pi}}, \hat{\boldsymbol{\pi}}^i)}[A^{\boldsymbol{\pi}}] \right| + 2\frac{1}{1-\gamma} \sum_{j \in e^i} \alpha^j e^{\boldsymbol{\pi}}$$

$$\leq 4\epsilon^{\hat{\boldsymbol{\pi}}^{i-1}} \alpha^i \sum_{t=0}^{\infty} \gamma^t (1 - (1 - \sum_{j \in (e^i \cup \{i\})} \alpha^j)^t)$$

$$+ \frac{1}{1-\gamma} \mathbb{E}_{(s,\boldsymbol{a}) \sim (d^{\boldsymbol{\pi}}, \hat{\boldsymbol{\pi}}^i)} \left[ \left| A^{\hat{\boldsymbol{\pi}}^{i-1}} - A^{\boldsymbol{\pi}} \right| \right] + 2\frac{1}{1-\gamma} \sum_{j \in e^i} \alpha^j e^{\boldsymbol{\pi}}$$

$$\leq 4\epsilon^{\hat{\boldsymbol{\pi}}^{i-1}} \alpha^i (\frac{1}{1-\gamma} - \frac{1}{1-\gamma(1-\sum_{j \in (e^i \cup \{i\})} \alpha^j)}) + \frac{1}{1-\gamma} \left[ 4\alpha^i \epsilon^{\hat{\boldsymbol{\pi}}^{i-1}} + 4 \sum_{j \in e^i} \alpha^j \epsilon^{\boldsymbol{\pi}} \right]$$

The right side of the last inequality is not a monotonic improvement bound, or it does not provide a guarantee for improving the expected performance $\mathcal{J}(\hat{\boldsymbol{\pi}}^i)$ since the term $\sum_{j \in e^i} \alpha^j \epsilon^{\boldsymbol{\pi}}$ is not controllable for agent $i$, whether through policy improvement or value learning. The uncontrollable term means the expected performance may not be improved even if the total variation distances of consecutive policies are well constrained.

## A.7    COMPARISONS ON MONOTONIC IMPROVEMENT BOUNDS

CoPPO and HAPPO have the same monotonic bound that is tighter than that of MAPPO. A2PO achieves the tightest monotonic bound given mild assumptions about the errors of preceding-agent off-policy correction, which is valid and easy to achieve since preceding-agent off-policy correction is a contraction operator. A sufficient condition that A2PO has the tightest bound is that $\xi^i <$ $\frac{\gamma(1-\gamma) \sum_{j \in \mathcal{N} - e^i - \{i\}} \alpha^j}{(1-\gamma(1-\sum_{j \in e^i \cup \{i\}} \alpha^j))(1-\gamma(1-\sum_{j=1}^{n} \alpha^j))}$, for all $i \in \mathcal{N}$.

### A.8 PRECEDING-AGENT OFF-POLICY CORRECTION

In Retrace($\lambda$) (Munos et al., 2016), consider the current policy as $\hat{\pi}^{i=1}$ and base policy as $\pi$, we have the following definition:

$$\mathcal{R}_t = r_t + \gamma Q_{t+1} + \sum_{k \geq 1} \gamma^k \Big( \prod_{j=1}^{k} \lambda \min \big( 1.0, \frac{\hat{\pi}^{i-1}(\boldsymbol{a}_{t+j}|s_{t+j})}{\boldsymbol{\pi}(\boldsymbol{a}_{t+j}|s_{t+j})} \big) \Big)(r_{t+k} + \gamma Q_{t+k+1} - Q_{t+k}) \,,$$

Following that same structure, we have:

$$\mathcal{R}_t = r_t + \gamma V_{t+1} + \sum_{k \geq 1} \gamma^k \Big( \prod_{j=1}^{k} \lambda \min \big( 1.0, \frac{\hat{\pi}^{i-1}(\boldsymbol{a}_{t+j}|s_{t+j})}{\boldsymbol{\pi}(\boldsymbol{a}_{t+j}|s_{t+j})} \big) \Big)(r_{t+k} + \gamma V_{t+k+1} - V_{t+k}) \,,$$

By subtracting $V_t$, we get the definition of PreOPC. Or one can get $\gamma A^{\boldsymbol{\pi}, \hat{\boldsymbol{\pi}}^{i-1}}$ by substituting $r_t + \gamma V_{t+1}$ for $Q_t$ and subtracting $r_t + \gamma V_{t+1}$.

### A.9 WHY OFF-POLICYNESS IS MORE SERIOUS IN SEQUENTIAL UPDATE SCHEME?

As shown in Fig. 13, the off policy correction in sequential update algorithms improves the performance significantly while similar performance gaps are not observed when used in simultaneous update algorithms. We attribute the difference to the influence of the clipping mechanism on the total variation distance.

From Corollary 2, $D_{TV}(\boldsymbol{\pi} \| \bar{\boldsymbol{\pi}}) < \sum_{i=1}^{n} D_{TV}(\pi^i \| \bar{\pi}^i)$. Although we can not prove exact relations, clipping the agents independently tends to larger total variation distances between the current and future policies of the agents, leading to more 'off-policyness' in sequential update algorithms.

## B  EXPERIMENTAL DETAILS

### B.1  IMPLEMENTATION

For a fair comparison, we (re)implement A2PO and the baselines based on the implementation of MAPPO. We keep the same structures for all the algorithms and tune all the algorithms following the same process, i.e., a grid search over a small collection of hyper-parameters, to avoid the influence of different implementation details on the results. The grid search is performed on three hyper-parameters: the learning rate, $\lambda$ and the agent block num in the tasks with numerous agents.

The algorithms, including A2PO and baselines, are implemented into both parameter sharing and parameter independent versions. A2PO in the parameter sharing version is implemented as in Alg. 2. The main modifications are colored in blue. We rearrange the loops of agents and ppo epochs. The number of ppo epochs is divided by $n$ for comparable updating times with the simultaneous algorithms. The approximated advantage is estimated by correcting the action probabilities of all the agents given such $e^i$.

---

**Algorithm 2:** Agent-by-agent Policy Optimization (Parameter Sharing)

1  Initialize the shared joint policy $\boldsymbol{\pi}_0 = \{\pi_0^1, \ldots, \pi_0^n\}$ with $\pi_0^1 = \cdots = \pi_0^n$, and the global value function $V$.
2  **for** *iteration* $m = 1, 2, \ldots$ **do**
3  $\quad$ Collect data using $\boldsymbol{\pi}_{m-1}$.
4  $\quad$ Policy $\boldsymbol{\pi}_m = \boldsymbol{\pi}_{m-1}$.
5  $\quad$ **for** $\lceil \frac{P}{n} \rceil$ *epochs* **do**
6  $\quad\quad$ **for** $k = 1, \ldots, n$ **do**
7  $\quad\quad\quad$ Agent $i = \mathcal{R}(k)$, preceding agents $e^i = \{\mathcal{R}(1), \ldots, \mathcal{R}(n-1)\}$.
8  $\quad\quad\quad$ Joint policy $\hat{\boldsymbol{\pi}}^i = \boldsymbol{\pi}_m$.
9  $\quad\quad\quad$ Compute the advantage approximation as $A^{\boldsymbol{\pi}, \hat{\boldsymbol{\pi}}^{i-1}}(s, \boldsymbol{a})$ via eq. (2).
10 $\quad\quad\quad$ Compute the value target $v(s_t) = A^{\boldsymbol{\pi}, \hat{\boldsymbol{\pi}}^{i-1}}(s, \boldsymbol{a}) + V(s)$.
11 $\quad\quad\quad$ $\pi_m^i = \arg\max_{\pi_m^i} \tilde{\mathcal{L}}_{\hat{\boldsymbol{\pi}}^{i-1}}(\hat{\boldsymbol{\pi}}^i)$ as in eq. (6).
12 $\quad\quad\quad$ $V = \arg\min_V \mathbb{E}_{s \sim d^{\boldsymbol{\pi}}} \|v(s) - V(s)\|^2$.

---

Practically, each agent is equipped with a value function, we generate the agent order at once to avoid estimating the advantage function $\frac{n(n-1)}{2}$ times. The order becomes $[1, \ldots, i, \ldots, j, \ldots, n]$ in which $\mathbb{E}|A^i| >= \mathbb{E}|A^j|$.

### B.2  EXPERIMENTAL SETUP AND ADDITIONAL RESULTS

#### B.2.1  STARCRAFTII MULTI-AGENT CHALLENGE

StarCraftII Multi-agent Challenge (SMAC) (Samvelyan et al., 2019) provides a wide range of multi-agent tasks in the battle scenarios of StarCraftII. Algorithms adopting parameter sharing have shown superior performance in SMAC, so all the algorithms are implemented as parameter sharing. As shown in Tab. 5, we evaluate the algorithms in 12 maps of SMAC with various difficulties, in which the baselines can not achieve 100% win rates easily. We use the results of Qmix in Yu et al. (2022). The learning curves for episode return are summarized in Fig. 8.

#### B.2.2  MULTI-AGENT MUJOCO

Multi-agent MuJoCo (MA MuJoCo) (Peng et al., 2021) contains a range of multi-agent robot continuous control tasks, in which an agent controls the composition of robot joints. MA MuJoCo extends the high-dimensional single-agent locomotion tasks in MuJoCo (Todorov et al., 2012), a widely adopted benchmark for SARL algorithms (Haarnoja et al., 2018; He & Hou, 2020), into the multi-agent case. Agents must cooperate in their actions for robot locomotion, and different agents control different compositions of the robot joints. We use the reward settings of the original paper

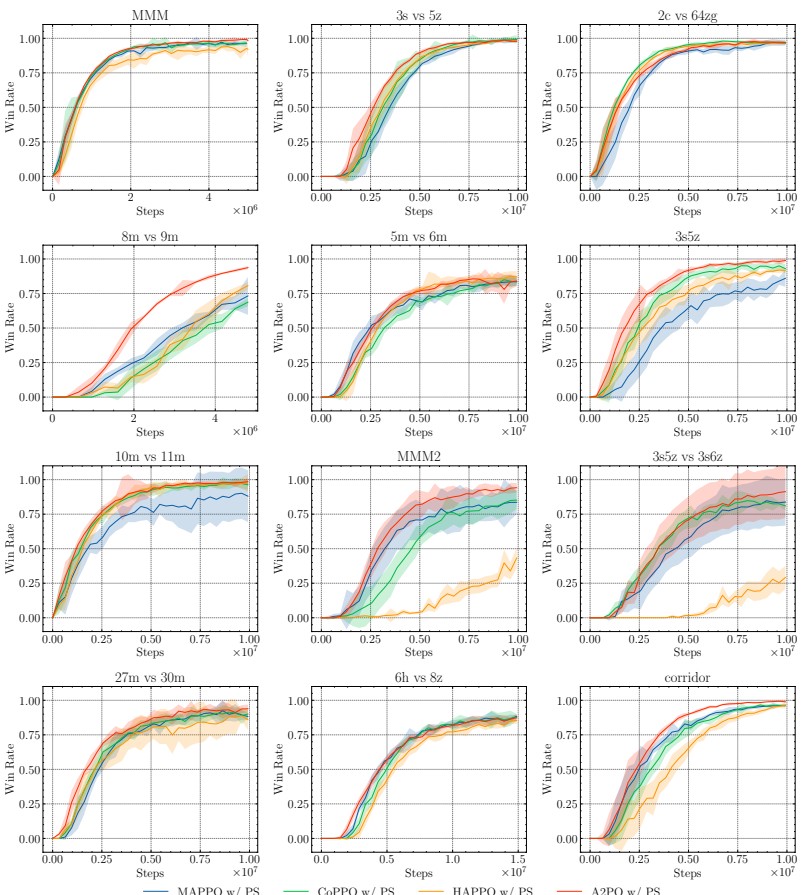

Figure 8: Comparisons of median win rate on SMAC.

Table 5: Median win rates and standard deviations on SMAC tasks. 'w/ PS' means the algorithm is implemented as parameter sharing

| Map | Difficulty | MAPPO w/ PS | CoPPO w/ PS | HAPPO w/ PS | A2PO w/ PS | Qmix w/ PS |
|---|---|---|---|---|---|---|
| MMM | Easy | 96.9(0.988) | 96.9(1.25) | 95.3(2.48) | **100(1.07)** | 95.3(2.5) |
| 3s_vs_5z | Hard | **100(1.17)** | **100(2.08)** | **100(0.659)** | **100(0.534)** | 98.4(2.4) |
| 2c_vs_64zg | Hard | **98.4(1.74)** | 96.9(0.521) | 96.9(0.521) | 96.9(0.659) | 92.2(4.0) |
| 3s5z | Hard | 84.4(4.39) | 92.2(2.35) | 92.2(1.74) | **98.4(1.04)** | 88.3(2.9) |
| 5m_vs_6m | Hard | 84.4(2.77) | 84.4(2.12) | 87.5(2.51) | **90.6(3.06)** | 75.8(3.7) |
| 8m_vs_9m | Hard | 84.4(2.39) | 84.4(2.04) | 96.9(3.78) | **100(1.04)** | 92.2(2.0) |
| 10m_vs_11m | Hard | 93.8(18.7) | 96.9(2.6) | 98.4(2.99) | **100(0.521)** | 95.3(1.0) |
| 6h_vs_8z | Super Hard | 87.5(1.53) | **90.6(0.765)** | 87.5(1.49) | **90.6(1.32)** | 9.4(2.0) |
| 3s5z_vs_3s6z | Super Hard | 82.8(19.2) | 84.4(2.9) | 37.5(13.2) | **93.8(19.8)** | 82.8(5.3) |
| MMM2 | Super Hard | 90.6(8.89) | 90.6(6.93) | 51.6(9.01) | **98.4(1.25)** | 87.5(2.6) |
| 27m_vs_30m | Super Hard | 93.8(3.75) | 93.8(2.2) | 90.6(4.77) | **100(1.55)** | 39.1(9.8) |
| corridor | Super Hard | 96.9(0) | **100(0.659)** | 96.9(0.96) | **100(0)** | 84.4(2.5) |
| overall | / | 91.1(5.46) | 92.6(2.2) | 85.9(3.68) | **97.4(2.65)** | 78.4(3.6) |

but set the environment to be fully observable[6]. The agents are heterogeneous and mostly asymmetric in MA-MuJoCo, so we implement the algorithms as parameter-independent. We test 14 tasks of 6 scenarios in MA MuJoCo, as illustrated in Fig. 9.

### B.2.3 MULTI-AGENT PARTICLE ENVIRONMENT

We consider the Navigation task of the Multi-agent Particle Environment (MPE) (Lowe et al., 2017) implemented in PettingZoo (Terry et al., 2021) which implements MPE with minor fixes and provides convenience for customizing the number of agents and landmarks, and customizing the global and local rewards., with 3 and 5 agents and corresponding numbers of landmarks. The agents are rewarded based on the minimum distance to the landmarks and penalized for colliding with each other, meaning that the reward is entirely up to the coordination behavior. We adopted two different reward settings: Fully Cooperative and General-sum. In the Fully Cooperative setting, the agents share the same reward, while in the General-sum setting, the agents are additionally rewarded based on the local collision detection. The results in Fig. 10 show that A2PO generally outperforms the baselines on the average return and the sample efficiency. Noted that A2PO is developed in fully cooperative games, the results in the General-sum setting reveal the potential of extending A2PO into general-sum games. Further, the performance gap between A2PO and the baselines enlarges with the increasing number of agents.

### B.2.4 GOOGLE RESEARCH FOOTBALL

In the above experiments, we have evaluated A2PO in tasks where agents can learn both their micro-operations and coordination behaviors (SMAC and MA-MuJoCo) and tasks where agents can only learn coordination behaviors (the Navigation task). However, the coordination behaviors in the above tasks are relatively easy to discover, e.g., agents learn to concentrate their fire to shoot the enemies and cover each other in SMAC. Recent works (Wen et al., 2022; Yu et al., 2022) have conducted experiments on Google Research Football academic scenarios with a small number of players and easily accessible targets, making the coordination behavior also easy to discover. In contrast, we evaluate A2PO in the full-game scenarios, where the players of the left team, except for the goalkeeper, are controlled to play a football match against the right team controlled by the built-in AI provided by GRF. The agents in the full-game scenarios have high-dimensional observations, complex action spaces, and a long-span timescale (3000 steps). We reconstruct the observation space and

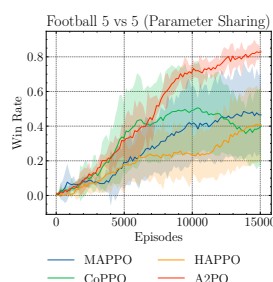

Figure 11: 5-vs-5 scenario with Parameter sharing.

---

[6]Empirically, we find the fully observable setting does not make the tasks easier because of the information redundancy.

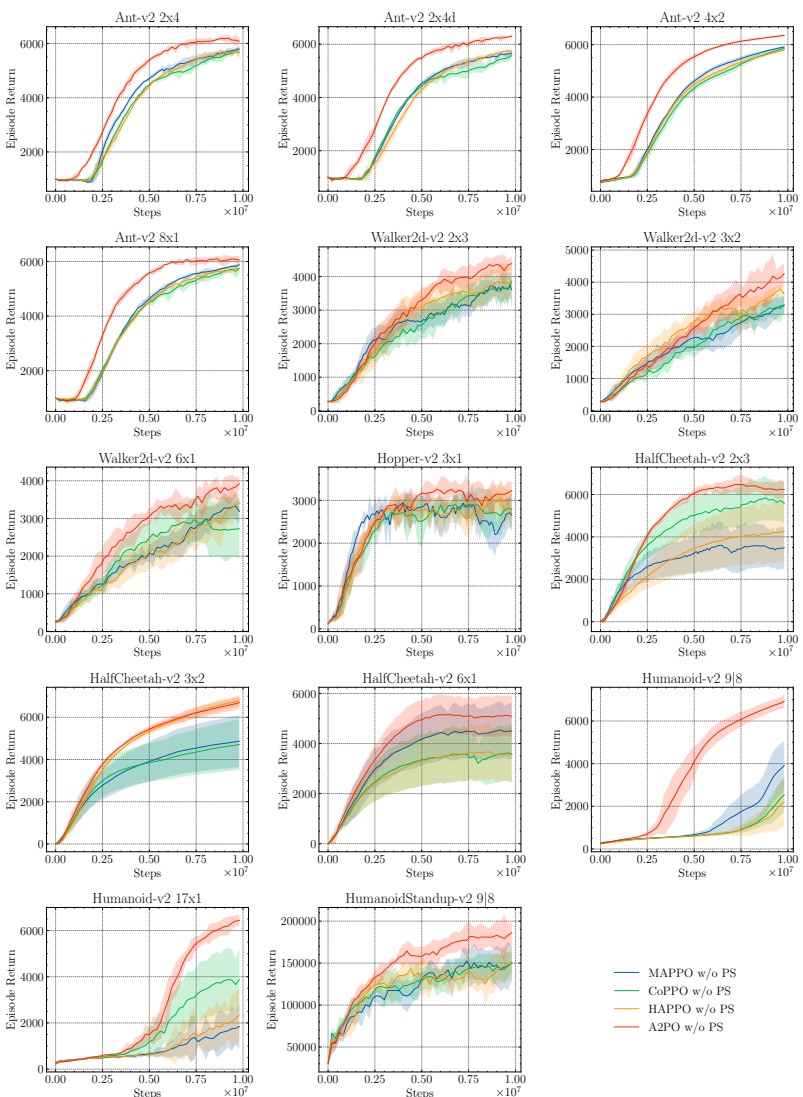

Figure 9: Comparisons of average episode return on MA-MuJoCo.

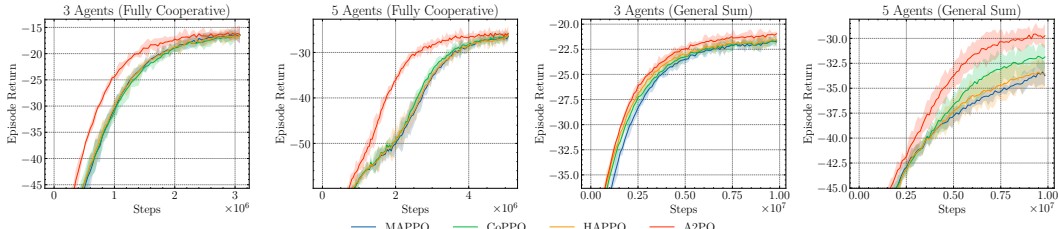

Figure 10: Comparisons of averaged return on the Multi-agent Particle Environment Navigation task. **Left**: The fully cooperative setting. **Right**: The general-sum setting.

design a dense reward to facilitate training in these scenarios based on Football-paris. The observation is formed to be agent-specific. The reward function estimates the behaviors of the entire team, including scoring, and carrying the ball to the opponent's restricted area et al., but not the individual behaviors such as ball-passing (Li et al., 2021). We implement all the algorithms for the 5-vs-5 scenario as both parameter sharing and parameter-independent. The additional results with algorithms implemented as parameter sharing are shown in Fig. 11, in which A2PO gets free from the trouble that the controlled agents have similar behavior and compete for the ball (Li et al., 2021).

We implement all the algorithms on the 11-vs-11 scenario as parameter sharing using MALib (Zhou et al., 2021) for acceleration and train the algorithms for 300M environment steps. We summarize the learned behaviors observed in the game videos:

- **Basic Skills**. The agents trained by MAPPO and CoPPO perform unsatisfactorily in basic skills such as dribbling, shooting, and the agents even run out of bounds frequently. In contrast, the agents trained by HAPPO and A2PO perform better in the basic skills. We attribute the problems to the non-stationarity issue that seriously influences the simultaneous updating algorithms. We also note that the agents trained by all the algorithms fail to understand the off-side mechanism and occasionally gather together on the opponent's bottom line.

- **Passing and Receiving Coordination**. We analyze the direct way for coordination: passing and receiving the ball. As illustrated in Tab. 3, the agents trained by MAPPO have the lowest number of successful passes and the lowest successful pass rate, and we can hardly observe the agents passing the ball. Agents trained by CoPPO perform better on passing the ball but suffer from poor basic skills, and get tackled after receiving the ball. Agents trained by HAPPO prefer passing the ball without considering the teammates' situations, e.g., the receiver is marked by several opponents. Agents trained by A2PO can pass the ball to their teammates in a way that leads to a score. We attribute the performance gain to the preceding-agent off-policy correction, which means that agents estimate the teammates' situations and intentions better.

We further visualize the learned behaviors of A2PO in Fig. 12. In the top of Fig. 12, two players cooperatively break through the opponent's defense and complete a passing and receiving coordination for scoring. In the bottom of Fig. 12, three players make a fast thrust by two long passes: the goalkeeper passes the ball to the player at the edge, and the player at the edge passes the ball to the player behind the opponents. The complex coordination strategies are hardly observed in other baselines.

### B.2.5 ABLATION

**Preceding agent off-policy correction**. More ablations on preceding-agent off-policy correction are shown in Fig. 13. The baselines are:

- MAPPO w/ V-trace, CoPPO w/ V-trace: Simultaneous update methods with advantage estimation as V-trace.

- HAPPO w/ PreOPC: HAPPO with advantage estimation as PreOPC.

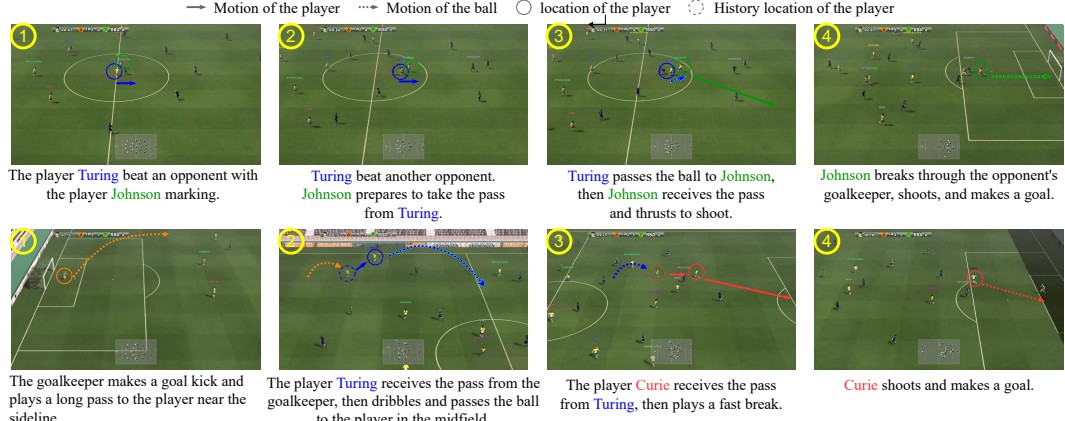

Figure 12: Visualization of trained A2PO policies on the Google Research Football 11-vs-11 scenario, which shows that A2PO encourages complex cooperation behaviors to make a goal. **Top**: Player Turing and Johnson cooperate to beat multiple opponents to break through the defense and make a goal. **Bottom**: The goalkeeper, player Turing, and Curie achieve the pass and receive cooperation twice. A fast thrust is made by consecutively passing the ball.

In this ablation study, the baselines are equipped with off-policy correction methods. The experiment yields the following three conclusions:

- The results firstly support the conclusion in Sec. 3.3 that applying PreOPC to sequential update methods results in a greater performance improvement than applying V-trace to simultaneous update methods.

- Secondly, the primary distinction between A2PO and HAPPO with PreOPC is the clipping objective. The results demonstrate that the clipping objective derived from the single-agent improvement bound contributes to the performance improvement.

- And thirdly, although we were unable to assess the error of PreOPC, we compare A2PO with RPISA-PPO, which can be viewed as A2PO algorithms with error-free off-policy correction methods (the advantage estimation is error-free) at the expense of sample inefficiency. A2PO reaches or outperforms the asymptotic performance of RPISA-PPO. A2PO outperforms RPISA-PPO since RPISA-PPO suffers from performance degradation as a result of agents updating policies with separated data (Taheri & Thrampoulidis, 2022).

We further analyze the sensitivity to the hyper-parameter $\lambda$. Results in Fig. 14 illustrate that preceding-agent off-policy correction does not introduce more sensitivity.

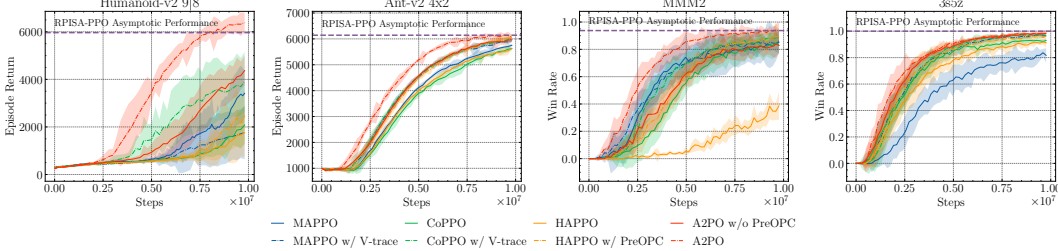

Figure 13: Ablation experiments on preceding-agent off-policy correction.

**Agent Selection Rule**. More ablations on the agent selection rules are shown in Fig. 13. We compare two additional rules: 'Reverse-greedy' and 'Reverse-semi-greedy'. 'Reverse' means selecting the agent with the minimal advantage first. While we observe that the effect of the selection rule becomes less significant in tasks with homogeneous or symmetric agents.

Going deeper into the effects of agent selection rules, we show that the agents with implicit guidance from the advantage estimation benefit from greedily selecting agents in Fig. 16 and 17. More

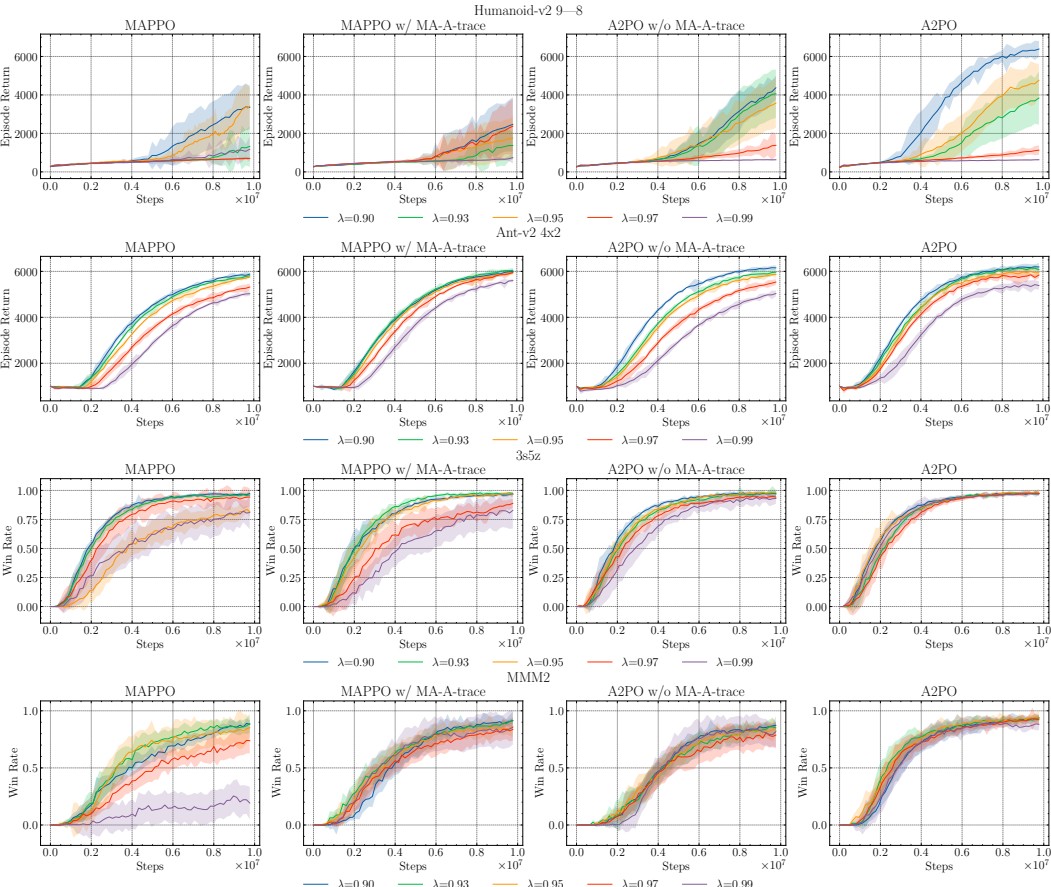

Figure 14: Sensitivity analysis of $\lambda$.

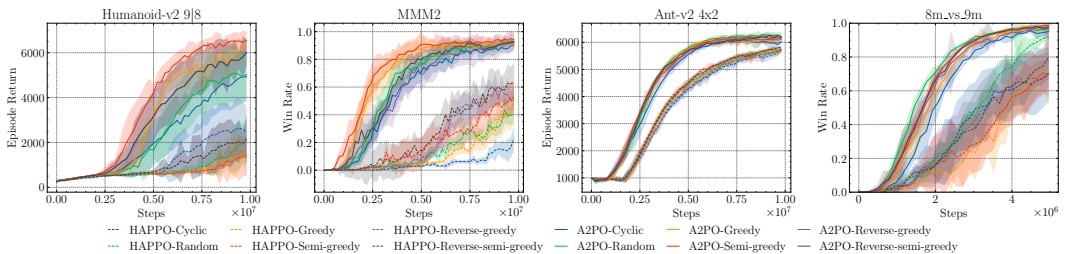

Figure 15: Ablation experiments on the agent selection rules. **Left**: Heterogeneous or asymmetric agents. **Right**: Homogeneous or symmetric agents.

even bars appear in one fig means the agents are more balanced in terms of the guidance from the advantage estimation. Take the agent 10 in Fig. 16 for example, under 'Cyclic' and 'Random' rules, agent 10 perform the worst with high proportions, while it has higher proportions in prior ranks under 'Greedy' rule.

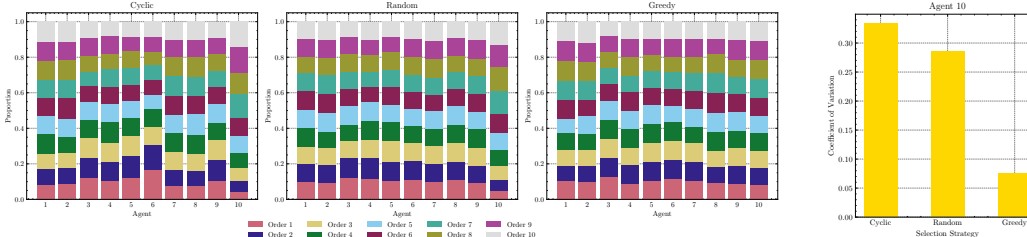

(a) The imbalance of the agents. The bar of agent $i$ illustrates the proportion of its ranks in terms of $\mathbb{E}_{s,a^i}[|A^{\pi,\hat{\pi}^i}|]$. Especially, agent 10 has implicit guidance, i.e., a small absolute value of advantage function when using Cyclic and Random selection rule, but is comparable with other agents with Greedy selection rule.

(b) The coefficient of variance of agent 10's order proportions.

Figure 16: Agents' imbalance in terms of the estimated advantage. The experiment is conducted on the MMM2 task of SMAC.

**Adaptive Clipping Parameter**. More ablations on the adaptive clipping parameter are shown in Fig. 18. Similarly, we observe that the effect of the adaptive clipping parameter becomes less significant in tasks with homogeneous or symmetric agents.

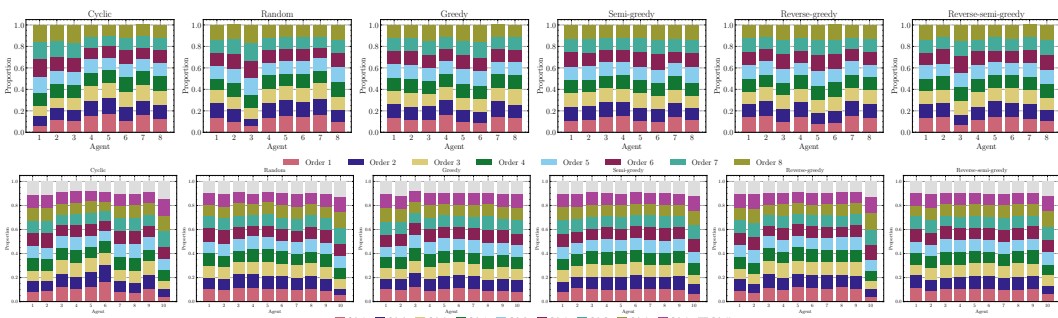

Figure 17: More experiments on the agents' imbalance in terms of the estimated advantage. **Top**: 3s5z task. **Bottom**: MMM2 task.

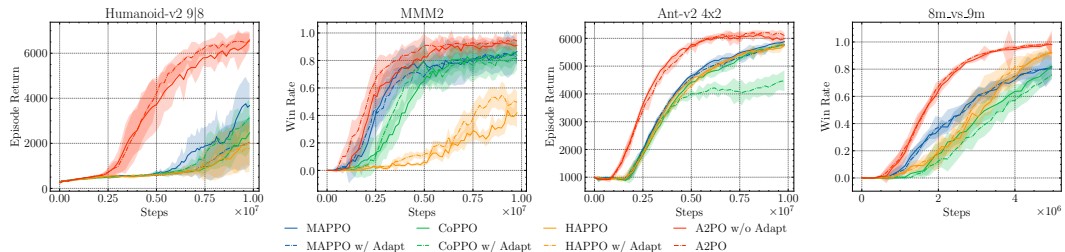

Figure 18: More ablation experiments on the adaptive clipping parameter. **Left**: Heterogeneous or asymmetric agents. **Right**: Homogeneous or symmetric agents.

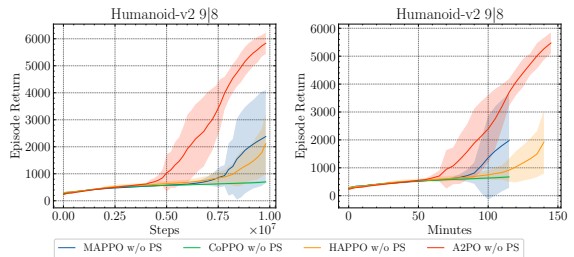

(a) Comparison on Humanoid 9|8 over both environment steps and training time.

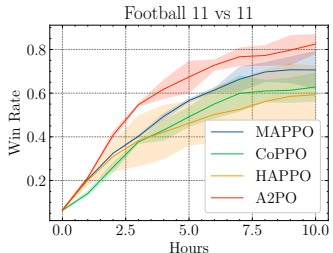

(b) Comparison on GRF 11-vs-11 scenario.

Figure 19: Wall time Analysis.

## B.3 WALL TIME ANALYSIS

Multiple updates in a stage may increase training time, and the need for more training time may impact the scalability of A2PO, which is a common concern regarding the sequential update scheme. Nevertheless, a sequential update scheme will increase training time less than might be expected. Before proceeding, we note that the majority of experiments in our work are synchronously implemented, and the training time consists of the time spent updating policies and collecting samples.

We have proposed a simple yet effective method for controlling training time in order to reduce training time. As a trade-off between performance and training time, we divide the agents into blocks to reduce the number of update iterations. For example, the tasks with 10 agents can be divided into 3 blocks, with sizes 3, 3, 4, respectively, and only 3 updates will be performed in a policy update iteration. From the implementation perspective, since the number of samples used in a single update decreases, the sequential update scheme requires less memory and less updating time when update policies. Therefore, it is possible to control the training time as less than 1.5 times the training time of the simultaneous update methods. In addition, assuming a good implementation, fewer update iterations will be performed if mini-batches are used in a single policy update, as the size of a mini-batch can be greater in sequential update methods under limited memory resources. In such a case, fewer mini-batches will be used, further decreasing the training time. Moreover, sampling consumes the majority of the training time, and the increased updating time appears less significant when analysing the wall time for on-policy algorithms with synchronized implementations.

The training time is depicted in Tab. 6. A2PO achieves significantly greater performance with only marginally more training time. In addition, we illustrate the Humanoid 9|8 comparisons regarding environment steps and training time in Fig. 19a, and the comparisons on the GRF 11-vs-11 scenario in Fig. 19b. A2PO maintains an advantage in terms of training time.

## B.4 HYPER-PARAMETERS

We tune several hyper-parameters in all the benchmarks, other hyper-parameters refer to the settings used in MAPPO. $c_\epsilon$ are selected to be $0.5$ in all the tasks.

Table 6: The comparison of training duration. The format of the first line in a cell is: Training time(Sampling time+Updating Time). The second line of a cell represents the time normalized.

| Task | MAPPO | CoPPO | HAPPO | A2PO |
|---|---|---|---|---|
| 3s5z | 3h29m(3h3m+0h26m) 1.00(0.87 + 0.13) | 3h33m(3h6m+0h27m) 1.02(0.89 + 0.13) | 3h49m(3h7m+0h42m) 1.10(0.89 + 0.20) | 4h32m(3h41m+0h51m) 1.30(1.06 + 0.25) |
| 27m vs 30m | 13h23m(8h31m + 4h52m) 1.00(0.64 + 0.36) | 13h19m(8h24m + 4h55m) 1.00(0.63 + 0.37) | 16h2m(8h20m + 7h42m) 1.20(0.62 + 0.58) | 15h53m(8h7m + 7h46m) 1.19(0.61 + 0.58) |
| Humanoid 9\|8 | 2h0m(1h45m + 0h15m) 1.00(0.87 + 0.13) | 1h58m(1h43m + 0h15m) 0.99(0.86 + 0.13) | 2h15m(1h45m + 0h30m) 1.12(0.87 + 0.25) | 2h31m(2h0m + 0h31m) 1.26(1.00 + 0.26) |
| Ant 4x2 | 6h42m(6h16m + 0h26m) 1.00(0.93 + 0.07) | 6h45m(6h19m + 0h26m) 1.01(0.94 + 0.07) | 7h29m(6h5m + 1h24m) 1.12(0.91 + 0.21) | 7h2m(5h34m + 1h28m) 1.05(0.83 + 0.22) |
| Humanoid 17x1 | 12h9m(10h6m + 2h3m) 1.00(0.83 + 0.17) | 17h7m(15h5m + 2h2m) 1.41(1.24 + 0.17) | 16h55m(11h2m + 5h53m) 1.39(0.91 + 0.48) | 19h25m(11h59m + 7h26m) 1.60(0.99 + 0.61) |
| Football 5vs5 | 34h46m(32h47m + 1h59m) 1.00(0.94 + 0.06) | 32h46m(30h49m + 1h57m) 0.94(0.89 + 0.06) | 39h26m(31h54m + 7h32m) 1.13(0.92 + 0.22) | 37h26m(30h2m + 7h24m) 1.08(0.86 + 0.21) |

### B.4.1 STARCRAFTII MULTI-AGENT CHALLENGE

We list the hyper-parameters used for each task of SMAC in Tab. 7.

Table 7: Hyper-parameters in SMAC.

| Hyperparameters | agent block | ppo epoch | $\lambda$ | $\epsilon$ |
|---|---|---|---|---|
| MMM | 3 | 12 | 0.95 | 0.2 |
| 3s_vs_5z | 3 | 15 | 0.95 | 0.05 |
| 2c_vs_64zg | 2 | 5 | 0.95 | 0.2 |
| 3s5z | 3 | 8 | 0.95 | 0.2 |
| 5m_vs_6m | 2 | 10 | 0.93 | 0.05 |
| 8m_vs_9m | 5 | 15 | 0.95 | 0.05 |
| 10m_vs_11m | 2 | 10 | 0.97 | 0.2 |
| 6h_vs_8z | 2 | 8 | 0.99 | 0.2 |
| 3s5z_vs_3s6z | 2 | 5 | 0.90 | 0.2 |
| MMM2 | 2 | 5 | 0.95 | 0.2 |
| 27m_vs_30m | 3 | 5 | 0.95 | 0.2 |
| corridor | 2 | 5 | 0.95 | 0.2 |

### B.4.2 MULTI-AGENT MUJOCO

For the model structure in MA MuJoCo, the output from the last layer is processed by a Tanh layer and the action distribution is modeled as a Gaussian distribution initialized with mean as 0 and log std as -0.5. The probability output of different actions are averaged when computing the policy ratio. The common hyper-parameters used in MA MuJoCo are listed in Tab. 8.

Table 8: Common hypermeters in MA MuJoCo.

| Hyperparameters | Values |
|---|---|
| entropy | 0 |
| gain | 0.01 |
| batch size | 4000 |

### B.4.3 MULTI-AGENT PARTICLE ENVIRONMENT

We list the hyper-parameters used in MPE in Tab. 10.

### B.4.4 GOOGLE RESEARCH FOOTBALL

We list the hyper-parameters used in the GRF 5-vs-5 scenario in Tab. 11.

Table 9: Hypermeters for the scenarios in MA MuJoCo.

| Hyperparameters | Ant | HalfCheetah | Hopper | Humanoid | HumanoidStandup | Walker2d |
|---|---|---|---|---|---|---|
| agent block | 8x1:4 | / | / | 17x1:5 | 17x1:4 | / |
| ppo epoch | 8 | 5 | 8 | 5 | 5 | 5 |
| actor lr | 3e-4 | 3e-4 | 1e-4 | 3e-4 | 3e-4 | 3e-4 |
| critic lr | 3e-4 | 3e-4 | 1e-4 | 3e-4 | 3e-4 | 3e-4 |
| $\lambda$ | 0.93 | 0.93 | 0.95 | 0.9 | 0.93 | 0.93 |
| $\epsilon$ | 0.2 | 0.2 | 0.1 | 0.2 | 0.2 | 0.2 |

Table 10: Hypermeters for the scenarios in MPE.

| Hyperparameters | Values |
|---|---|
| ppo epoch | 8 |
| chunk length | 5 |
| entropy | 0.05 |
| actor lr | 2e-4 |
| critic lr | 2e-4 |
| $\lambda$ | 0.97 |
| $\epsilon$ | 0.2 |

## C  THE RELATED WORK OF OTHER MARL METHODS

**Value decomposition methods**. The value decomposition methods such as VDN (Sunehag et al., 2017) and Qmix (Rashid et al., 2018), factorize the joint value function and adopt the centralized training and decentralized execution paradigm. The Individual-Global-MAX (IGM) principle is proposed to ensure consistency between the joint and local greedy action selections in the joint $Q$-value function $Q_{tot}(\boldsymbol{\tau}, \boldsymbol{a})$ and the individual $Q$-value function $\{Q^i(\tau^i, a^i)\}_{i=1}^n$: $\forall \boldsymbol{\tau} \in \mathcal{T}, \arg\max_{\boldsymbol{a} \in \mathcal{A}} Q_{tot}(\boldsymbol{\tau}, \boldsymbol{a}) = (\arg\max_{a^1 \in \mathcal{A}^1} Q^1(\tau^1, a^1), \ldots, \arg\max_{a^n \in \mathcal{A}^n} Q^n(\tau^n, a^n))$. Two sufficient conditions, the additivity and the monotonicity, to satisfy IGM are proposed in Sunehag et al. (2017) and Rashid et al. (2018) respectively. In addition to the $V$ function and $Q$ function decomposition, QPLEX (Wang et al., 2021) considers implementing IGM in the dueling structure where $Q = V + A$. QPLEX only constrains the advantage functions to satisfy the IGM principle. The global advantage function is decomposed as $A_{tot}(\boldsymbol{\tau}, \boldsymbol{a}) = \sum_{i=1}^n \lambda_i(\boldsymbol{\tau}, \boldsymbol{a}) A_i(\boldsymbol{\tau}, a_i)$, where $\lambda_i(\boldsymbol{\tau}, \boldsymbol{a}) > 0$. We evaluate the performance of Qmix in Tab. 2 and Tab. 5. Integrating the IGM principle into A2PO without compromising the monotonic improvement guarantee is a desirable extension. Specifically, the advantage-based IGM establishes a connection between the global advantage function and the local advantage functions, and the advantage decomposition $A_{tot}(\boldsymbol{\tau}, \boldsymbol{a}) = \sum_{i=1}^n \lambda_i(\boldsymbol{\tau}, \boldsymbol{a}) A_i(\boldsymbol{\tau}, a_i)$ will not jeopardize the derivation of the monotonic improvement guarantee.

**Convergence and optimality of MARL**. T-PPO (Ye et al., 2022) firstly introduce a framework called Generalized Multi-Agent Actor-Critic with Policy Factorization (GPF-MAC), which consists of methods with factorized local policies and may become stuck in sub-optimality. To address this problems, T-PPO transforms a multi-agent MDP into a special "single-agent" MDP with a sequential structure. T-PPO transforms a multi-agent MDP into a "single-agent" MDP with a sequential structure to address this issue. T-PPO has been shown to produce an optimal policy if implemented properly. Theoretically, sequential update methods, such as A2PO and HAPPO, are also instances of GPF-MAC and may be stuck into sub-optimal policies. The main differences between A2PO and T-PPO include that A2PO updates the factorized policies sequentially and makes decisions simultaneously, while T-PPO makes decisions sequentially, and that A2PO does not introduce the virtual state and the sequential transformation framework network. And theoretically, T-PPO may compromise the monotonic improvement guarantee. In Tab. 12, we compare A2PO, MAPPO and T-PPO on SMAC tasks empirically. A2PO is superior to T-PPO in the majority of tasks.

Table 11: Hypermeters for the scenarios in MPE.

| Hyperparameters | Values |
|---|---|
| ppo epoch | 10 |
| chunk length | 10 |
| entropy | 0.001 |
| actor lr | 5e-4 |
| critic lr | 5e-4 |
| $\lambda$ | 0.95 |
| $\epsilon$ | 0.25 |
| $\gamma$ | 0.995 |

Table 12: Comparisons of A2PO, MAPPO and T-PPO.

| Map | Difficulty | MAPPO w/ PS | T-PPO w/ PS | A2PO w/ PS |
|---|---|---|---|---|
| 1c3s5z | Easy | **100(0.0)** | 99.8((0.0) | **100(0.0)** |
| MMM2 | Super Hard | 90.6(8.9) | 81.6(7.7) | **98.4(1.3)** |
| 3s5z_vs_3s6z | Super Hard | 82.8(19.2) | 85.5(5.2) | **93.8(19.8)** |
| 6h_vs_8z | Super Hard | 87.5(1.5) | **91.8(1.1)** | 90.6(1.3) |
| corridor | Super Hard | 99.1(0.3) | 96.9(0.0) | **100(0.0)** |

## D  THE RELATED WORK OF COORDINATE DESCENT

Realizing the similarity between the sequential policy update scheme and the block coordinate descent algorithms, we borrow the optimization techniques in the coordinate descent algorithms to accelerate the optimization and amplify the convergence advantage over the simultaneous update scheme (Gordon & Tibshirani, 2015; Shi et al., 2017). One of the critical questions in the coordinate descent algorithms is selecting the coordinate for the next-step optimization. Glasmachers & Dogan (2013); Lu et al. (2018) provided analyses of the convergence rate advantage of the Gauss-Southwell rule, i.e., greedily selecting the coordinate with the maximal gradient, over the random selection rule. We recognize the optimization of our surrogate objective (Schulman et al., 2017) agent-by-agent as a block coordinate descent problem. Therefore the agent selection rule plays a crucial role in accelerating the optimization. Inspired by the coordinate selection rules, we propose greedy and semi-greedy agent selection rules and empirically show that the underperforming agents benefit from the greedily selecting agents.

