# OpenReview forum: "Order Matters: Agent-by-agent Policy Optimization"
_ICLR.cc/2023/Conference — ICLR 2023 poster_

### Official Review · Reviewer_N8aU · 2022-10-24

**Confidence:** 3
**Correctness:** 3
**Technical Novelty And Significance:** 3
**Empirical Novelty And Significance:** 3
**Recommendation:** 8

**Clarity, Quality, Novelty And Reproducibility:**

Although the proposed algorithm and the experimental results are interesting, the overall presentation of the paper is unsatisfactory.



**Strength And Weaknesses:**

This paper addresses an important problem in MARL. In prior work, the non-stationarity due to simultaneous policy update of agents in the multi-agent setting gives rise to high variance in the gradients, requiring more samples for convergence of each agent. Instead, the proposed A2PO algorithm uses importance sampling to weight off-policy samples used in sequential update of agent policies for the different agents in the team.

In general, the paper is not easy to follow due to the use of several notations that are not properly defined before being referenced and therefore, forces the reader to go back and forth looking for the appropriate definitions. It would help to perhaps create a consolidated table for all the notations and symbols used in the paper.

Specific comments about notation:

- Page 3, Section 3.2 - what does the superscript $\mathcal{I}$ indicate in $\mathcal{L}^{\mathcal{I}}_{\hat{\pi}^{i-1}}$?

- Page 6, first paragraph, second line after eq 6: $\mathcal{C}(\cdot,\cdot)$ has been used without first defining it. The definition is given in Page 7.

Sec 4, Semi-greedy Agent Selection Rule: “purely greedy selection may lead to early convergence which harms the performance.” - Could you please elaborate this point?

Sec 4, Adaptive Clipping Parameter: Could you please explain what is precisely meant by “balanced and sufficient” clipping ranges?

Sec 5.2: The results in the ablation study are difficult to follow. The paragraph PreOPC compares to MAPPO and CoPPO with V-trace, also to HAPPO and RPISA-PPO : but these methods are not described in the main paper which makes it difficult to understand the results.



**Summary Of The Paper:**

This paper proposes a sequential (agent by agent) policy gradient update algorithm (A2PO) in the MARL setting that guarantees monotonic policy improvement for each agent and also their joint policy. Theoretical guarantees are provided for monotonic improvement and experimental results also support the claims made in the paper.


**Summary Of The Review:**

I would recommend rejecting the paper in its current form, primarily because the presentation of the material is difficult to follow and the paper does not seem to be polished overall due to several typos, even in the appendix, that make it cumbersome to read.

===========

After the rebuttal period, the authors' response to all the reviews has helped me understand the work more clearly. The authors have added a Table for consolidating all the notations used in the appendix. Although my initial reading of the paper gave me the impression that the paper is hard to follow and from a quick comparison of the latest version of the paper, it does not look like the authors have included any additional explanations in the text - I do not think the other reviewers had the same problem, so it might just be me. But I will agree that the authors have sufficiently addressed concerns regarding the main contributions of the paper in their response here to the reviewers and their experimental results are thorough and extensive. So, I am updating my score accordingly.

---

> ### Author Response · Authors · 2022-11-19
> **Response to Reviewer N8aU [1/2]**
>
> Thank you for suggesting potential improvements to the presentation. We are glad that the reviewer considers the proposed method and the experiments here interesting. We apologize for the fact that we use some notations and concepts that are implicitly defined by context, and we have elaborated on them in the paper.
>
> > It would help to perhaps create a consolidated table for all the notations and symbols used in the paper.
>
> At the beginning of appendix A, we have included a table detailing the notations and symbols used.
>
> > Page 3, Section 3.2 - what does the superscript $I$ indicate in $\mathcal{L}^{I}_{\hat{\boldsymbol{\pi}}^{i-1}}(\hat{\boldsymbol{\pi}}^{i})$?
>
> The full sentence is "Since agent $i$ updates its policy from $\hat{\boldsymbol{\pi}}^{i-1}$, an **intuitive surrogate objective** used by agent $i$ could be formulated as $\mathcal{L}^{I}_{\hat{\boldsymbol{\pi}}^{i-1}}(\hat{\boldsymbol{\pi}}^{i})$ ...". Thus the notation denotes the mentioned 'intuitive surrogate objective' for agent $i$ and the subscript $I$ means 'Intuitive' here. We also further explain it in the paper.
>
> > Page 6, first paragraph, second line after eq 6: $\mathcal{C}(\cdot, \cdot)$ has been used without first defining it. The definition is given in Page 7.
>
> $\mathcal{C}(\cdot, \cdot)$ firstly appears as "The clipping parameter $\epsilon^{i}$ is selected as $\epsilon^{i}=\mathcal{C}(\epsilon, i)$, where $\epsilon$ is the base clipping parameter.", which means that $\mathcal{C}(\cdot, \cdot)$ is a function taking a real number  (the base clipping parameter) and an integer (the agent index) as input, and outputs the clipping parameter for the specific agent. Briefly, $\mathcal{C}(\cdot, \cdot): \mathbb{R}\times \mathbb{N} \mapsto \mathbb{R} $ is the function determining the clipping parameters. Moreover, "The definition is given in Page 7" in the question refers to "We propose a simple yet effective method, named adaptive clipping parameter, to adjust the clipping parameters according to the updating order: $\mathcal{C}(\epsilon, k) = \epsilon \cdot c_{\epsilon}+\epsilon \cdot (1-c_{\epsilon}) \cdot k / n$, where $c_{\epsilon}$ is a hyper-parameter.",  which is the **implementation** adopted in the paper. We additionally explain that "$\mathcal{C}(\cdot,\cdot)$ is the clipping parameter adapting function" after $\mathcal{C}(\cdot,\cdot)$ first appears.
>
> > Sec 4, Semi-greedy Agent Selection Rule: “purely greedy selection may lead to early convergence which harms the performance.” - Could you please elaborate this point?
>
> - **Deficiency of the Greedy Selection Rule in Coordinate Ascent**. In the paragraph titled "Semi-greedy Agent Selection Rule," we concentrate on optimizing the surrogate objective of all agents more efficiently by selecting suitable agent selection rules. After realizing that optimizing the surrogate objective is a block coordinate ascent problem, selecting agents is a particular case of selecting coordinate blocks. In the sentence next to the quoted sentence in the question, i.e., "Therefore, we ... to avoid converging too early (Lu et al., 2018).", we cite the paper that elaborates on the deficiency of greedy coordinate selection. Lu et al. (2018) investigated the random, greedy, and semi-greedy rules in coordinate descent. They demonstrated that greedy coordinate descent lacks guarantees of theoretical convergence and convergence rate. They discovered empirically that greedy coordinate descent tends to converge too quickly and has inferior performance in comparison to semi-greedy coordinate descent. Moreover, they demonstrated that the semi-greedy coordinate has the highest rate of convergence.
> - **Semi-greedy Agent Selection Rule**. Inspired by the insights mentioned above, in our paper, we consequently choose the semi-greedy agent selection rule to reduce the possibility of early convergence brought by the greedy agent selection rule and accelerate the optimization of the surrogate objective of all agents. In our ablations on the agent selection rule shown in Sec. 5.2 and appendix B.2.5, we also empirically found that the greedy agent selection rule performs worse than the semi-greedy agent selection, which supports our proposed semi-greedy agent selection rule.
>
> To sum up, the statement "purely greedy selection may lead to early convergence which harms the performance." **represents our empirical finding that greedily selecting agents based on the absolute value of the estimated advantage function tends to converge too early and has inferior performance compared to other agent selection rules, such as the random selection rule and the cyclic selection rule**. Moreover, similar findings can also be found in (Lu et al., 2018). Based on this observation, we introduce randomness into the greedy agent selection rule and propose the semi-greedy agent selection.

---

> > ### Author Response · Authors · 2022-11-19
> > **Response to Reviewer N8aU [2/2]**
> >
> >
> > > Sec 4, Adaptive Clipping Parameter: Could you please explain what is precisely meant by “balanced and sufficient” clipping ranges?
> >
> > The phrase "balanced and sufficient" is used to describe the Figure 2 on page 7.  "balanced and sufficient" means that the clipping ranges of the 3 agents are neither excessively small nor excessively large, and an illustration of the opposite, i.e., imbalanced or insufficient clipping ranges, is shown on the left side of Figure 2: agent 1 has an extremely small clipping range due to the large variations of agents 2 and 3, and the extremely small clipping range may prevent agent 1 from finding a policy that improves expected performance.
> >
> > > Sec 5.2: The results in the ablation study are difficult to follow. The paragraph PreOPC compares to MAPPO and CoPPO with V-trace, also to HAPPO and RPISA-PPO : but these methods are not described in the main paper which makes it difficult to understand the results.
> >
> >
> > 1. The Related Works section describes and appropriately cites the baselines MAPPO, CoPPO, and HAPPO. In addition, since surrogate objectives are the primary difference between baselines and A2PO, we describe the baselines' surrogate objectives in the appendix. RPISA is also described in the Related Works section, and RPISA-PPO is an implementation of RPISA that selects PPO as its base algorithm, as described in the caption of Table 1.
> > 2. We include the citation of V-trace in this paragraph. V-trace is a well-known off-policy correction method based on multi-step importance sampling. Our proposed PreOPC is also based on the multi-step importance sampling but only corrects the preceding agents' policies. According to the definitions of these off-policy correction methods, the flags "w/ V-trace" and "w/ PreOPC" indicate using the corresponding off-policy correction in advantage estimation, as described in Section 3.3.
> > 3. We explain further why this ablation study is necessary. Since A2PO employs PreOPC in advantage estimation, we compare A2PO with baselines equipped with appropriate off-policy correction techniques: applying V-trace to the simultaneous methods, i.e., MAPPO and CoPPO, and PreOPC to the sequential method HAPPO. In addition, according to Table 1, RPISA-PPO achieves the tightest bound, and its asymptotic performance is comparable to that of A2PO algorithms with perfect off-policy correction techniques (the advantage approximation has no errors). The followings are the objectives of this ablation.
> >     - To verify that PreOPC provides a more significant improvement to sequential update methods than V-trace does to simultaneous update methods, as predicted by the theoretical results presented in Section 3.
> >     - To verify that the clipping objective used in A2PO, derived under PreOPC, contributes to the performance gain by comparing the performance of A2PO and that of HAPPO w/ PreOPC.
> >     - To demonstrate the efficacy of PreOPC, that is, to demonstrate whether A2PO can reach the asymptotic performance of RPISA-PPO.
> >
> > 	The preceding description of the ablation study is added to appendix B.2.5.
> >
> >
> > \[Lu et al., 2018\] Lu, Haihao, Robert Freund, and Vahab Mirrokni. "Accelerating greedy coordinate descent methods." International Conference on Machine Learning. PMLR, 2018.

---

> > > ### Comment · Reviewer_N8aU · 2022-11-26
> > > **Post-rebuttal reply**
> > >
> > > Thank you for the thorough explanations. After reading the authors' response, I have a better understanding of the paper. I will change my recommendation accordingly.

---

### Official Review · Reviewer_7L45 · 2022-10-25

**Confidence:** 4
**Clarity, Quality, Novelty And Reproducibility:** A sound work with a clear presentatio…
**Correctness:** 3
**Technical Novelty And Significance:** 2
**Empirical Novelty And Significance:** 3
**Recommendation:** 6

**Strength And Weaknesses:**

**STRENGTHS**

This paper does a sufficient literature survey. It is well-structured and easy to follow. Mathematical statements and proofs look sound. Experimental results show improvements over baselines.

WEAKNESSES

1) The motivation is a little confusing. Why should monotonic improvement for each agent be guaranteed? Is it possible that a better joint bound is obtained by giving up a monotonic improvement guarantee for each agent?

2) The sample-inefficient claim for RPISA may not be right. It is not necessary to drop $(n-1)/n$ of the collected sample. For example, [1] is an implementation of RPISA, which is proven to be sample efficient. It will be helpful to have further discussions on these algorithms both theoretically and empirically.

3) A2PO seems to take a lot of time in training. It is highly desired to elaborate on this issue.

[1] Jianing Ye, Chenghao Li, Jianhao Wang, and Chongjie Zhang. Towards Global Optimality in Cooperative MARL with Sequential Transformation. ArXiv, abs/2207.11143, 2022.



**Summary Of The Paper:**

This paper briefly reviews previous works of sequential updating schemes in multiagent trust region algorithms and points out the limitations of these works, which are either sample inefficient or lack of monotonic improvement guarantee for each agent. To achieve both sample efficiency and monotonic improvement guarantee for each agent, it proposes a new algorithm Agent-by-Agent Policy Optimization (A2PO) and analyzes its monotonic improvement bound.

**Summary Of The Review:**

This work discusses the limitations of previous multi-agent trust-region algorithms and proposes an interesting method, which empirically shows outperformance. However, some further discussions are needed to justify the significance of this work. A final recommendation will depend on how questions in WEAKNESS are clarified.

---

> ### Author Response · Authors · 2022-11-19
> **Response to Reviewer 7L45**
>
> We thank the reviewer for the valuable comments and are glad to discuss the suggested interesting work.
>
> > Why should monotonic improvement for each agent be guaranteed? Is it possible that a better joint bound is obtained by giving up a monotonic improvement guarantee for each agent?
>
> In the common response, we detail the importance of the single-agent policy improvement guarantee in 3 points.
> The answer to whether a tighter joint bound exists without the single-agent improvement guarantee is contingent on the analysis tools used.
> - Without considering the single agent improvement, the joint bound in CoPPO/HAPPO may be the tightest given the analysis tools used in this paper.
> - Using the new analysis tools proposed by Li et al. (2022), which derives a closed-form solution to trust-region optimization using calculus of variation and tightens existing bounds, it is possible to obtain a tighter bound than those in CoPPO/HAPPO without the single-agent improvement guarantee.
>
> > The sample-inefficient claim for RPISA may not be right. It is not necessary to drop (n−1)/n of the collected sample. For example, (Ye et al., 2022) is an implementation of RPISA, which is proven to be sample efficient. It will be helpful to have further discussions on these algorithms both theoretically and empirically.
>
> We first compare RPISA and Ye et al. (2022). **The implementation of RPISA mentioned in our work retains the monotonic improvement guarantees of a single agent and the joint policy at the cost of low sample efficiency**. We include the RPISA algorithm since RPISA alternates between updating one agent and sampling data, which means that the original implementation of RPISA updates the agents sequentially. In addition, the state distribution perceived by an agent is consistent with the joint policy of the time, as new samples are collected after the previous agent updates its policy and before the agent updates its policy. RPISA is free of the non-stationarity problem and can be considered a "Single-agent" problem, and thus retains the theoretical properties of the selected base algorithm. The implementation of RPISA, RPISA-PPO, therefore, retains the monotonic improvement guarantees of a single agent and the joint policy. Low sample efficiency is the price paid to preserve monotonic improvement properties. We are happy to see another implementation of RPISA in Ye et al. (2022). Ye et al. (2022) proposed a sequential transformation for global optimality. However, the proposed method may still suffer from the non-stationarity problem that the state distribution changes because of the policy variation of the preceding agents. Moreover, whether the proposed method retains the monotonic improvement properties remains to be discovered.
>
> Though the work (Ye et al. 2022) was recently submitted to Arxiv in July, we are glad to add this remarkable and concurrent work to related works. We discuss this work in detail in appendix C and summarize the discussion as follows:
> - Theoretically, the proposed method in (Ye et al., 2022), T-PPO, converges to the global optimality under mild assumptions. The proposed framework Generalized Multi-Agent Actor-Critic with Policy Factorization (GPF-MAC) unifies the methods with factorized policies, including A2PO and HAPPO, and the instances of GPF-MAC may be stuck in the sub-optimality. We have not discussed convergence and optimality in our work, and the discussion about (Ye et al., 2022) will help investigate the whole picture of multi-agent policy optimization.
> - Empirically, we compare A2PO, MAPPO, and T-PPO on SMAC tasks in the following table[^1]. The results show that A2PO outperforms other baselines in the majority of the tasks.
>
>     | Task         | A2PO           | T-PPO         | MAPPO        |
>     | ------------ | -------------- | ------------- | ------------ |
>     | MMM2         | **98.4(1.3)**  | 81.6(7.7)     | 90.6(8.9)    |
>     | 3s5z vs 3s6z | **93.8(19.8)** | 85.5(5.2)     | 82.8(19.2)   |
>     | 6h vs 8z     | 90.6(1.3)      | **91.8(1.1)** | 87.5(1.5)    |
>     | corridor     | **100(0.0)**   | 96.9(0.0)     | 99.1(0.3)    |
>     | 1c3s5z       | **100(0.0)**   | 99.8(0.0)     | **100(0.0)** |
>
> > A2PO seems to take a lot of time in training. It is highly desired to elaborate on this issue.
>
> The detailed discussion can be found in the common response, and we add a wall-time analysis in appendix B.3. The conclusion is that A2PO does not require evident increased training time compared with other baselines due to the fact that A2PO requires less updating time than intuitively expected and the majority of training time is spent sampling data.
>
> [^1]: T-PPO results are the original results since A2PO and T-PPO are both implemented based on MAPPO.
>
> [Li et al., 2022] Li, H., Clavette, N. &amp; He, H.. (2022). An Analytical Update Rule for General Policy Optimization. ICML 2022.

---

> ### Author Response · Authors · 2022-11-29
> **Looking Forward to Your Response**
>
> Dear reviewer 7L45,
>
> We thank you again for your valuable comments and suggestions. We have made great efforts to address the concerns.
> - The common response clarifies the significance of the single-agent policy improvement guarantee, and we discuss how to obtain a tighter bound without the single-agent guarantee.
> - We firstly distinguish the implementation of RPISA in our paper and T-PPO. Then we compare A2PO and T-PPO both theoretically and empirically.
> - Based on an analysis of wall-time usage in the common response, we conclude that A2PO does not require significantly more training time than other baselines.
>
> We look forward to your response and are happy to continue our discussion.
>
> Sincerely, Authors

---

### Official Review · Reviewer_GdLP · 2022-10-31

**Confidence:** 5
**Correctness:** 1
**Technical Novelty And Significance:** 2
**Empirical Novelty And Significance:** 2
**Recommendation:** 5

**Clarity, Quality, Novelty And Reproducibility:**

- Quality: Good.
- Clarity: Good.
- Originality: Good.

**Strength And Weaknesses:**

Strength:
1. I admit this paper has good writing and analysis.

Weakness:
1. The wall time may be increased due to the multiple updates. The authors should compare wall time usage of different methods.

2. The authors try to improve the sample efficiency problem of current MARL algorithms, but the performance improvements are not significant. Reasons below:

- a. The benchmarks are somewhat toy. For instance, SMAC is a well-studied benchmark, many methods have shown performance not worse than A2PO (such as MAT[1]).

- b. The author should consider MARL tasks that are really in the predicament of the sample efficiency, such as multi-agent competitive environments (such as Neural MMO[2]) and multi-agent visual navigation tasks.


 [1] Wen M, Kuba J G, Lin R, et al. Multi-Agent Reinforcement Learning is a Sequence Modeling Problem[J]. arXiv preprint arXiv:2205.14953, 2022.

 [2] Suarez, Joseph, et al. "The neural mmo platform for massively multiagent research." arXiv preprint arXiv:2110.07594 (2021).


**Summary Of The Paper:**

This paper proposed a new MARL training objective to improve training efficiency. In the experiment, the proposed method, A2PO, outperforms previous SOTA methods.

**Summary Of The Review:**

This paper proposed a new MARL method and evaluated on some MARL benchmarks. But the improvements are not that satisfactory,

---

> ### Author Response · Authors · 2022-11-19
> **Response to Reviewer GdLP [1/3]**
>
> We appreciate the reviewer's comments and suggestions, while he or she may have misunderstood the paper's motivation and main contributions.
>
> ### Possible misunderstood points
>
> >  This paper proposed a new MARL training objective to improve training efficiency.
> >  The authors try to improve the sample efficiency problem of current MARL algorithms, but the performance improvements are not significant.
>
> As stated in the first sentences of the second and third paragraphs, **the purpose of our research is to investigate sequential update methods and examine the role of agent update orders**. Sequential update methods mitigate the problem of non-stationarity that is inherent to simultaneous update methods. The non-stationarity issue causes a high variance in policy optimization and necessitates the requirement for more samples. **A2PO alleviates the non-stationarity problem by sequentially updating the agents and adapting the clipping parameters for each agent. The improved sample efficiency is one of the results of alleviating the non-stationarity problem.**
> The phrase "improve the sample efficiency" in the abstract has two meanings: a) the sequential update method, RPISA, drops $(n-1)/n$ samples whereas A2PO does not; b) we empirically observe the advantage of A2PO in sample efficiency in the experiments due to the fact that A2PO alleviates the non-stationarity problem.
> Moreover, in the experiments, we mainly focus on the expected return and the learned cooperative behaviors instead of the sample complexity.
>
> > a. The benchmarks are somewhat toy. For instance, SMAC is a well-studied benchmark, many methods have shown performance not worse than A2PO (such as MAT[1]).
>
> 1. We admit that SMAC is a well-studied benchmark, which is why fair comparisons between A2PO and other recent SOTA methods on SMAC are necessary.
>     - SMAC is well-studied means that we can conveniently compare different algorithms already evaluated in SMAC, and the re-produced results are reliable.
>     - **Although many methods have demonstrated good performance in SMAC, the experiments conducted on SMAC are still convincing when considering fairness.** We note that a comparison with the methods that do not focus on the optimization objective or do not follow the CTDE paradigm could not support or refute our claims. Consider the method mentioned in the question, MAT, as an illustration. MAT takes into account a fully centralized paradigm. A2PO, on the other hand, considers centralized training with decentralized execution. Moreover, MAT focuses on the action and feature processing using a transformer rather than the optimization objective. As a more fair comparison, the following table demonstrates that A2PO still has an advantage compared to the decentralized MAT (MAT-dec), which follows the decentralized execution paradigm and uses a transformer to process the feature input[^1].
>
>         | Task         | A2PO      | MAT-dec   | MAPPO     |
>         | ------------ | --------- | --------- | --------- |
>         | MMM          | **100(1.1)**  | 98.1(2.1) | 96.9(1.0) |
>         | 2c vs 64zg   | 96.9(0.7) | 95.9(2.3) | **98.4(1.7)** |
>         | 5m vs 6m     | **90.6(3.1)** | 83.1(4.6) | 84.4(2.8) |
>         | 8m vs 9m     | **100(1.0)**  | 95.0(4.6) | 84.4(2.4) |
>         | 3s5z_vs_3s6z | **93.8(20)**  | 85.3(7.5) | 82.8(19)  |
>         | MMM2         | **98.4(1.3)** | 91.2(5.3) | 90.6(8.9) |
>         | 27m_vs_30m   | **100(1.55)** | 95.3(2.2) | 93.8(3.8) |
>
>         The results show that the experiments conducted in SMAC can still demonstrate the performance improvement of A2PO when considering fairness in comparison.
>
> [^1]: We use the original results of MAT-dec since MAT-dec and A2PO are both implemented based on the same implementation of MAPPO.

---

> > ### Author Response · Authors · 2022-11-19
> > **Response to Reviewer GdLP [2/3]**
> >
> > > a. The benchmarks are somewhat toy. For instance, SMAC is a well-studied benchmark, many methods have shown performance not worse than A2PO (such as MAT[1]).
> >
> > **[continued]**
> >
> > 2. **Two Benchmarks that are more complicated and not well-studied**. Except for SMAC, we have evaluated our method on two additional benchmarks: Multi-agent MuJoCo (MA MuJoCo) (Peng et al., 2021) and Google Research Football (GRF) full-game scenarios (Kurach et al., 2020). Both benchmarks, particularly GRF full-game scenarios, are complex and require further investigation.
> >     - MA MuJoCo is a benchmark for continuous multi-agent robotic control. MA MuJoCo consists of various robotic tasks in which multiple agents within a single robot cooperate to solve a locomotion task. MA MuJoCo has become gradually popular in recent multi-agent works (Wen et al., 2022; Pan et al., 2022).
> >     - GRF is a benchmark for training agents to play football in a physics-based 3D simulator. Agents in GRF are required to complete difficult tasks (beating built-in football bots), utilize high-dimensional (1280 x 720 visual input or vectors with hundreds of floats) and partially observable features, make complex decisions (nearly 20 actions), receive sparse rewards, and operate in stochastic environments. GRF is a challenging benchmark and is in no way a toy.
> >         Several recent works adopt academic tasks of GRF to evaluate the methods (wen et al., 2022, Fu et al., 2022; Wang et al., 2022; Yu et al., 2021), while we have further evaluated A2PO on the full-game scenarios, the most challenging tasks of GRF. In the full-game scenarios, agents meet extra difficulties, including a long episode horizon (3000), an increased number of agents (10 and 22), and the numerous requirements of samples ($10^8\sim 10^9$). A2PO outperforms the baselines in the GRF full-game scenarios. Especially, A2PO obtain a nearly 100% win rate on the 5-vs-5 scenario. **The results on GRF full-game scenarios demonstrate that A2PO significantly improves performance on the not-toy and challenging tasks.**
> >
> > 3. **Analyzable behaviors in toy benchmarks**. It is more practical to conduct experiments with well-studied or toy tasks to analyze learned behaviors. We take Multi-agent Particle Environment (MPE) (Lowe et al., 2017), the fundamental benchmark in MARL as an example. As pointed out at the beginning of Sec. 5, we conduct experiments in MPE with two factors in mind: a) The (near) optimal behaviors in MPE are simple to recognize, and the learned behaviors are easy to visualize and analyze; b) MPE is simple to modify, and the (near) optimal behaviors in the modified tasks are also simple to recognize. We modify the navigation task as a general-sum task and evaluate A2PO's potential for general-sum task extension. We can determine if the A2PO agents acquire the expected coordination behaviors in the fundamental benchmark MPE.
> >     Experiments conducted in well-studied or toy tasks serve a specific purpose and are necessary for facilitating the analysis of learned behaviors.
> >
> >
> > > b. The author should consider MARL tasks that are really in the predicament of the sample efficiency, such as multi-agent competitive environments (such as Neural MMO[2]) and multi-agent visual navigation tasks.
> >
> > We have clarified that the purpose of our research is to investigate sequential update methods and examine the role of agent update orders and that the improvement in the sample efficiency is one of the results of alleviating the non-stationarity problem. Thus benchmarks used in experiments do not need to be in the predicament of the sample efficiency. We thank the reviewer for suggesting benchmarks, though they may not be appropriate.
> > - We aim to investigate the role of agent update orders and the sequential update scheme, which requires the centralized training and decentralized execution (CTDE) paradigm. Because adversaries in competitive scenarios often refuse to share information during training or "cooperate" by acting as the way in training during the execution stage, the CTDE paradigm requires fully cooperative multi-agent scenarios. Neural MMO is a competitive environment, and thus, we can not evaluate A2PO on it.
> > - Regarding multi-agent visual navigation tasks, the question of how to process the visual input may be one of the most important problems (Liu et al., 2022), but it is beyond the scope of this paper.
> >
> > > Correctness: 1: The main claims of the paper are incorrect or not at all supported by theory or empirical results.
> >
> > We guess you selected this choice because you might misunderstand that our motivation and contribution is to improve the sample efficiency. If not, we would like to know why you believe our main claims to be false or unsupported, and we are happy to discuss them with you further.

---

> > > ### Author Response · Authors · 2022-11-19
> > > **Response to Reviewer GdLP [3/3]**
> > >
> > > ### The suggestions
> > >
> > > > The wall time may be increased due to the multiple updates. The authors should compare wall time usage of different methods.
> > >
> > > Thank you for your suggestions. We put the wall-time comparison and detailed discussion in the common response, and appendix B.3. The results indicate that the increased time required for updating policies is not evident and that A2PO still shows an advantage on the expected return in terms of the training time.
> > >
> > > **References**
> > >
> > > [Lowe et al., 2017] Lowe, R., Wu, Y. I., Tamar, A., Harb, J., Pieter Abbeel, O., & Mordatch, I. (2017). Multi-agent actor-critic for mixed cooperative-competitive environments. Advances in neural information processing systems, 30.
> > >
> > > [Peng et al., 2021] Peng, B., Rashid, T., Schroeder de Witt, C., Kamienny, P. A., Torr, P., Böhmer, W., & Whiteson, S. (2021). Facmac: Factored multi-agent centralised policy gradients. Advances in Neural Information Processing Systems, 34, 12208-12221.
> > >
> > > [Pan et al., 2022] Pan, L., Huang, L., Ma, T., & Xu, H. (2022, June). Plan better amid conservatism: Offline multi-agent reinforcement learning with actor rectification. In International Conference on Machine Learning (pp. 17221-17237). PMLR.
> > >
> > > [Wen et al., 2022] Wen, M., Kuba, J. G., Lin, R., Zhang, W., Wen, Y., Wang, J., & Yang, Y. (2022). Multi-Agent Reinforcement Learning is a Sequence Modeling Problem. NeurIPS 2022.
> > >
> > > [Kurach et al., 2020] Kurach, Karol, et al. "Google research football: A novel reinforcement learning environment." Proceedings of the AAAI Conference on Artificial Intelligence. Vol. 34. No. 04. 2020.
> > >
> > > [Fu et al., 2022] Fu, W., Yu, C., Xu, Z., Yang, J., & Wu, Y.M. (2022). Revisiting Some Common Practices in Cooperative Multi-Agent Reinforcement Learning. ICML.
> > >
> > > [Wang et al., 2022] Wang, L., Zhang, Y., Hu, Y., Wang, W., Zhang, C., Gao, Y., ... & Fan, C. (2022, June). Individual Reward Assisted Multi-Agent Reinforcement Learning. In International Conference on Machine Learning (pp. 23417-23432). PMLR.
> > >
> > > [Yu et al., 2021] Yu, C., Velu, A., Vinitsky, E., Wang, Y., Bayen, A., & Wu, Y. (2021). The surprising effectiveness of PPO in cooperative, multi-agent games. NeurIPS 2022.
> > >
> > > [Liu, et al., 2022] Liu, X., Guo, D., Liu, H., & Sun, F. (2022). Multi-Agent Embodied Visual Semantic Navigation With Scene Prior Knowledge. IEEE Robotics and Automation Letters, 7(2), 3154-3161.

---

> ### Author Response · Authors · 2022-11-29
> **Looking Forward to Your Response**
>
> Dear reviewer GdLP,
> Thanks again for your valuable comments and suggestions, and we have done our best to answer your questions.
> - We guess you might misunderstand our motivations and contributions. We have further explained that the purpose of our research is to investigate the sequential update methods which have advantages in mitigating the non-stationarity problem, as well as examine the role of agent update orders.
> - For the experimental results, we firstly argue that experiments on well-studied benchmarks are necessary since the results on those benchmarks are reliable and that fair comparisons are convenient to obtain. Then, we clarify that we conducted experiments on two poorly-investigated benchmarks, i.e., Multi-agent MuJoCo and Google Research Football (GRF) full-scenario tasks. In particular, GRF full-scenario tasks are not toy and are challenges for current MARL methods. Lastly, we explain that the agents in toy games, such as Multi-agent Particle Environment (MPE), have analyzable behaviors and that the experiments on toy games serve a specific purpose besides demonstrating performance improvement.
> - The suggested benchmarks may not be applicable because this paper focuses on cooperative tasks and does not address the processing of visual input.
> - We analyze the wall-time usage in the common response, which indicates that the increased time required for updating policies is not evident.
>
> We look forward to your response and are eager to continue our discussion.
>
> Sincerely, Authors

---

### Official Review · Reviewer_GN3i · 2022-11-03

**Confidence:** 4
**Correctness:** 3
**Technical Novelty And Significance:** 3
**Empirical Novelty And Significance:** 3
**Recommendation:** 6

**Clarity, Quality, Novelty And Reproducibility:**

Overall, I think this paper is good in clarity, quality, novelty, and reproducibility. The theoretical analysis and experimental results are solid and quite convincing.

**Strength And Weaknesses:**

Strength:
1.	The paper is well written and well organized.

2.	The paper provides the single-agent monotonic bound in the setting with single-rollout and sequential policy update.

3.	Experimental results are solid.

Weaknesses:
Some questions:
1.	It seems that the theoretical results are established on the condition that all agents can observe the global state and take this global state as input. I am somewhat unsure whether the theoretical results are fully satisfied by the experiments because in the environments such as SMAC, agents typically have access to local observation.

2.	Regarding the sequential update scheme, one concern is scalability, though the experiments in this work show that the sequential update scheme has better performance. This is because the update of an agent's policy can be done only when all preceding agents have updated their policies. When facing the settings with a large number of agents, this could be slow compared to simultaneous update schemes such as MAPPO. So, in this sense, I would like to know how long (wall clock time) it takes for the A2PO and other simultaneous update schemes such as MAPPO and CoPPO.

3.	Can the authors provide more explanations about the assumption on $\xi^i$ in Theorem 1 and 2? Compared to previous monotonic bounds, A2PO can achieve a tighter bound only given that $\xi^i$ is small enough. How to ensure this in experiments (please correct me if I have misunderstood or missed something)?

4.	Though this paper focuses on the policy-based methods, I think some discussion on the value-based methods such as QMIX and its variants is necessary (maybe in the appendix).


**Summary Of The Paper:**

This paper considers the sample efficiency and single-agent monotonic improvement guarantees in sequential (agent-by-agent) policy updates in cooperative multi-agent tasks. To retain the guarantees of monotonic improvement for single agent, the authors propose the PreOPC, which approximate the true advantage of an agent. With this single-agent monotonic bound, a tighter joint monotonic bound can be achieved. Then, by optimizing the surrogate objective with the approximated advantage, the authors propose the A2PO algorithm, which achieves better performance with two techniques: semi-greedy agent selection rule and adaptive parameter clipping. Experiments have shown the effectiveness of the proposed algorithm.

**Summary Of The Review:**

After reading the paper carefully, I think my comments and recommendation are accurate.

---

> ### Author Response · Authors · 2022-11-19
> **Response to Reviewer GN3i**
>
> We thank you for the comments and suggestions, and we are glad to learn that you consider our work solid and convincing.
>
> > It seems that the theoretical results are established on the condition that all agents can observe the global state and take this global state as input. I am somewhat unsure whether the theoretical results are fully satisfied by the experiments because in the environments such as SMAC, agents typically have access to local observation.
>
> Yes, we establish our analysis within simplified scenarios, i.e., assuming full observability, because introducing other factors, such as partial observability and communication, complicates the problem formulation and makes it more difficult to discover significant theoretical findings. Thus we develop A2PO in simplified scenarios and evaluate its performance using more realistic tasks. In addition, for the partial observability issue in SMAC, we employ RNN in SMAC tasks.
>
> > When facing the settings with a large number of agents, this could be slow compared to simultaneous update schemes such as MAPPO. So, in this sense, I would like to know how long (wall clock time) it takes for the A2PO and other simultaneous update schemes such as MAPPO and CoPPO.
>
> The wall-time analysis is discussed in detail in the common response and appendix. B.3. A sequential update scheme will increase training time by a less amount than would be expected intuitively. For the scalability issue, we particularly present the wall-time in large-scale applications with numerous agents in the preceding in-depth discussion, and A2PO continues to demonstrate an advantage in terms of training time.
>
> > Can the authors provide more explanations about the assumption on $\xi^i$ in Theorem 1 and 2? A2PO can achieve a tighter bound only given that $\xi^i$ is small enough. How to ensure this in experiments?
>
> **The validity of the assumption about $\xi^i$**. We claim that PreOPC is a contraction operator at the end of Sec. 3 as *\`since preceding-agent off-policy correction is a contraction operator, which is a corollary of Theorem 1 in Munos et al. (2016).\'*. The following will elaborate on this statement:
>
> Based on Retrace (Munos et al., 2016), we propose our off-policy correction method, PreOPC. Recall that using PreOPC, the estimation of the advantage under joint policy $\hat{\boldsymbol{\pi}}^{i-1}$ can be rewritten as $A^{\boldsymbol{\pi},\hat{\boldsymbol{\pi}}^{i-1}}(s,\boldsymbol{a})=\mathcal{R}^{\hat{\boldsymbol{\pi}}^{i-1}}A^{\boldsymbol{\pi}}\; \forall s,\boldsymbol{a}$, where $\mathcal{R}^{\hat{\boldsymbol{\pi}}^{i-1}}$ is an operator. The following statements are corollaries of Theorem 1 in Munos et al. (2016).
> 1. $\mathcal{R}^{\hat{\boldsymbol{\pi}}^{i-1}}$ has a unique fixed point $A^{\hat{\boldsymbol{\pi}}^{i-1}}$.
> 2. For any advantage estimation function $A$, $\|\mathcal{R}^{\hat{\boldsymbol{\pi}}^{i-1}}A-A^{\hat{\boldsymbol{\pi}}^{i-1}}\|\leq \gamma\|A-A^{\hat{\boldsymbol{\pi}}^{i-1}}\|$.
>
> The propositions suggest that the operator $\mathcal{R}^{\hat{\boldsymbol{\pi}}^{i-1}}$ is a $\gamma$-contraction mapping around $A^{\hat{\boldsymbol{\pi}}^{i-1}}$. We define $\xi^{i}=\max_{s,\boldsymbol{a}}\|\mathcal{R}^{\hat{\boldsymbol{\pi}}^{i-1}}A-A^{\hat{\boldsymbol{\pi}}^{i-1}}\|$. Therefore $\xi^{i}$ converges to $0$ theoretically when the value function $V$ is updated iteratively using PreOPC. The assumption that $\xi^i$ becomes sufficiently small is valid.
>
> **To ensure that $\xi$ is sufficiently small in experiments**. Empirically, we firstly update $V(s)$ with the target as $V(s)+A^{\boldsymbol{\pi},\hat{\boldsymbol{\pi}}^{i-1}}(s,\boldsymbol{a})$ to ensure that $\xi^{i}$ will be small enough, which is ensured by the fact that the operator $\mathcal{R}^{\hat{\boldsymbol{\pi}}^{i-1}}$ is a $\gamma$-contraction mapping. However, we can not directly evaluate whether $\xi^{i}$ becomes sufficiently small through experimentation. Fortunately, we can validate the efficacy of PreOPC by comparing the performance of A2PO and RPISA-PPO since RPISA-PPO can be considered to have a perfect off-policy correction, i.e., $\xi^i=0$ at the cost of low sample efficiency. The experiments mentioned above have been presented in the first ablation study in Sec. 5.2. The results show that A2PO reaches or outperforms the asymptotic performance of RPISA-PPO using PreOPC, validating the efficacy of PreOPC.
>
> > Though this paper focuses on the policy-based methods, I think some discussion on the value-based methods such as QMIX and its variants is necessary (maybe in the appendix).
>
> Thanks for your suggestions, and we have updated the paper accordingly as follows:
> 1. We additionally compare the performance of Qmix in SMAC in Tables 2 and 5.
> 2. We discuss the value decomposition methods in appendix C.
>
> [Munos et al., 2016] Munos, R., Stepleton, T., Harutyunyan, A., & Bellemare, M. (2016). Safe and efficient off-policy reinforcement learning. Advances in neural information processing systems, 29.

---

> > ### Comment · Reviewer_GN3i · 2022-11-22
> > **Thanks for The Rebuttal**
> >
> > Dear Authors,
> >
> > Thanks for the rebuttal. I think the rebuttal addresses my most concerns. I would keep my score.

---

### Official Review · Reviewer_ZzqH · 2022-11-04

**Confidence:** 3
**Correctness:** 4
**Technical Novelty And Significance:** 3
**Empirical Novelty And Significance:** 4
**Recommendation:** 8

**Clarity, Quality, Novelty And Reproducibility:**

#### $\textbf{Clarity and Quality}$

The paper is well-written and not hard to follow. Although some definitions might be improved, this does not affect the overall clarity of the paper. Further, the problem is well-motivated and well-explained.

#### $\textbf{Novelty}$

The novelty of the paper stands in the key insight of using the off-policy correction method into a sequential update scheme. This combination results in higher performance, both theoretically and empirically. Theoretical techniques are standard.

#### $\textbf{Reproducibility}$

I did not check the code.

**Strength And Weaknesses:**

#### $\textbf{Strength}$

1. The problem considered in the paper is well-motivated and the challenges are incrementally introduced, together with the relevant related work.
2. The proposed off-policy correction method is theoretically guaranteed to satisfy monotonic single-agent improvement and, as a consequence, overall monotonic improvement.
3. The joint policy monotonic improvement is tighter than previous state of the art.
4. There are extensive experimental evaluations on well-known MARL benchmarks which clearly demonstrate the advantage of using A2PO.
5. The authors further provide experimental justification for using semi-greedy agent selection and adaptive clipping in the estimated probabilities.

#### $\textbf{Weaknesses/Questions}$

1. In the abstract it is claimed that A2PO improves sample efficiency, and in Table 1 it is again stated that the sample efficiency is high. What is meant by sample efficiency? Is it sample complexity? In any case, to my understanding, there is no subsequent result that shows an improvement of the sample efficiency in the paper. Is the statement based on a similar efficiency result imported from HAPPO since the rationale (at lease with respect to single roll-out usage) is similar?
2. I would have liked to see some convergence guarantees of A2PO for single agent and/or multiple agents scenarios. Does A2PO converge to individual optimal policies for every agent (jointly), and if so, how fast compared to HAPPO or similar methods, in terms of agents number, episode length and epoch length?
3. Can you elaborate on the importance of single-agent policy improvement in this setting? The reward function $r$ is the same for all agents, so there is no conflict of interest or notion of equilibrium, all agents are striving towards the same goal. If it wasn't for the improvement of joint monotonic bounds with respect to HAPPO, why should one care about single-agent policy improvement in this scenario?

#### $\textbf{Suggestions}$

1. In some places such as first paragraph of Section 4 or Table 1, in-line equations may be improved to increase readability.
2. In Section 3.1, either use the usual $\mathbb{P}$ notation for probability, or you may want to define what $Pr$ means.
3. I suggest you define the sample trajectory right after the definition of the value function, since that is where you first use it.
4. In the sequential update scheme illustration (pipeline) in Section 3.2, you might want to define the surrogate objective $\mathcal{L}$.

**Summary Of The Paper:**

This paper proposes a new surrogate objective in joint policy optimization for coordination tasks in MARL, following single roll-out and sequential update schemes. The new surrogate depends on a new off-policy correction method similar to the one in (Munos et al., 2016), and retains the monotonic improvement guarantees on individual agent and overall performance, while also maintaining high sample efficiency. Furthermore, they empirically show that the order of the agents' update and an adaptive clipping on the importance sampling probabilities has a substantial effect on the overall performance. Finally, they provide extensive experimental results on four MARL benchmarks, where they outperform previous trust-region MARL methods.


**Summary Of The Review:**

#### $\textbf{Summary}$

Overall, I find the contribution of the paper important to the MARL community. The authors provide an elegant solution to the single-agent policy improvement problem in trust region cooperative MARL using sequential update schemes. Furthermore, the extensive experimental results offer a complete picture of the benefits of A2PO, and also of the benefits of agent selection order.

---

> ### Author Response · Authors · 2022-11-19
> **Response to Reviewer ZzqH**
>
> We appreciate the reviewer's insightful comments and suggestions, and we have made several modifications to the paper in response to the suggestions.
>
> > In the abstract it is claimed that A2PO improves sample efficiency, and in Table 1 it is again stated that the sample efficiency is high. What is meant by sample efficiency? Is the statement based on a similar efficiency result imported from HAPPO since the rationale (at least with respect to single roll-out usage) is similar?
>
> When we state "Improvement in sample efficiency" in our paper, we mean that the method reduces sample complexity. In this work, we examine two sequential update baselines, HAPPO and RPISA, based on the single rollout and multiple rollout schemes, respectively. The phrase "improve the sample efficiency" mentioned in the abstract has two meanings: a) the sequential update method, RPISA, drops $(n-1)/n$ samples, whereas A2PO does not. The word 'Low' in Table 1 refers to the multiple rollout scheme, and the word 'High' in Table 1 refers to the single rollout scheme, and b) we empirically observe the sample efficiency advantage of A2PO over other baselines in the experiments.
>
> > I would have liked to see some convergence guarantees of A2PO for single agent and/or multiple agents scenarios. Does A2PO converge to individual optimal policies for every agent (jointly), and if so, how fast compared to HAPPO or similar methods, in terms of agents number, episode length and epoch length?
>
> To our knowledge, convergence guarantees of general MARL methods, even in simplified scenarios, remain open problems (Leonardos et al., 2021). In addition, there is no theoretical convergence rate analysis for the baseline methods, including HAPPO. We can only observe that A2PO converges faster than the baselines regarding environment steps.
>
> > Can you elaborate on the importance of single-agent policy improvement in this setting? If it wasn't for the improvement of joint monotonic bounds with respect to HAPPO, why should one care about single-agent policy improvement in this scenario?
>
> In the common response, we detail the importance of single-agent policy improvement in 3 points. The single-agent policy improvement guarantee not only results in a tighter joint bound but also derives a clipping objective that is verified to be beneficial and provides a perspective that profiles the contributions of each agent to the overall optimization. Inspired by the profiled contributions of each agent, we decompose the non-stationarity encountered by an agent into the non-stationarity introduced by the policy variation of preceding agents and propose the clipping parameter adaptation method.
>
> [Leonardos et al., 2021] Leonardos, S., Overman, W., Panageas, I., & Piliouras, G. (2021). Global convergence of multi-agent policy gradient in Markov potential games. arXiv preprint arXiv:2106.01969.

---

> > ### Comment · Reviewer_ZzqH · 2022-11-25
> > **Response**
> >
> > The reviewers have addressed my questions and thus I keep my score.

---

### Author Response · Authors · 2022-11-19
**Common Response [1/3]**

We gratefully acknowledge all reviewers for their insightful comments and suggestions. Two critical issues summarised from the comments and suggestions in the common response will be discussed as follows. The remaining issues will be addressed separately.

## Scalability and Wall Time
Multiple updates in a stage may increase training time, and the need for more training time may impact the scalability of A2PO, a common concern regarding the sequential update scheme. We argue that a sequential update scheme will increase training time by a less amount than would be expected intuitively. Before proceeding, we note that the algorithms in the majority of experiments in our work are implemented to be serial (alternating between sampling and updating), and the training time consists of the time spent updating policies and collecting samples. A detailed discussion about the wall time usage is provided in appendix B.2.3.

1. **Agent blocks division in order to control the training time**. We first acknowledge that sequentially updating agents increases the training time in the experiments. We have proposed a simple yet effective method to control the training time, as stated in the second paragraph of Sec.5: *\`We divide the agents into blocks for tasks with numerous agents to control the training time of A2PO comparable to other algorithms.\'*. We divide the agents into blocks to reduce the number of update iterations, which is a tradeoff between performance and training time. For example, the tasks with 10 agents can be divided into 3 blocks, with sizes 3, 3, 4, respectively, and only 3 updates will be performed in a policy update iteration. This will greatly reduce the training time while sacrificing relatively tiny performance compared to updating one agent at a time.
2. **sampling consumes the majority of training time.** The majority of experiments are conducted with serial implementations of on-policy algorithms (iterating between sampling and updating). Analysis of wall time reveals that sampling consumes the majority of training time, making the increased updating time appear less significant.
3. **Less computation requirement per update**. From the implementation perspective, since the number of samples used in a single update decreases, the sequential update scheme requires less memory and computation time per update. Therefore, the training time can be controlled to be less than 1.5 times that of the simultaneous update methods. In addition, assuming a suitable implementation, fewer mini-batches will be utilized if mini-batches are used in a single policy update, as the size of a mini-batch can be bigger in sequential update methods under limited memory resources. In such a case, fewer updates will be performed, further decreasing the training duration.

    The following table provides a comparison of different agent block numbers. Experiments are conducted using the Humanoid 17x1 task, which consists of seventeen agents. The results indicate that the update time will not increase linearly with the number of agent blocks.

    | Block Num | Update Time | Update Time Ratio | Train Time | Train Time Ratio |
    | --------- | ----------- | ----------------- | ---------- | ---------------- |
    | 2         | 7h26m       | 1.0               | 19h25m     | 1.0              |
    | 3         | 9h33m       | 1.3               | 20h12m     | 1.04             |
    | 5         | 14h14m      | 1.9               | 25h59m     | 1.34             |

### Results of the wall time usage


- **Wall time usage comparison**. We compare the training time of different methods in the table below. The total training time comprises the time used for sampling and updating.

    | Task            | MAPPO                | CoPPO                | HAPPO                | A2PO                 |
    | --------------- | -------------------- | -------------------- | -------------------- | -------------------- |
    | 3s5z            | 3h29m(3h3m+0h26m)    | 3h33m(3h6m+0h27m)    | 3h49m(3h7m+0h42m)    | 4h32m(3h41m+0h51m)   |
    | 27m_vs_30m      | 13h23m(8h31m+4h52m)  | 13h19m(8h24m+4h55m)  | 16h2m(8h20m+7h42m)   | 15h53m(8h7m+7h46m)   |
    | Humanoid 9\|8   | 2h0m(1h45m+0h15m)    | 1h58m(1h43m+0h15m)   | 2h15m(1h45m+0h30m)   | 2h31m(2h0m+0h31m)    |
    | Ant 4x2         | 6h42m(6h16m+0h26m)   | 6h45m(6h19m+0h26m)   | 7h29m(6h5m+1h24m)    | 7h2m(5h34m+1h28m)    |
    | Humanoid 17x1   | 12h9m(10h6m+2h3m)    | 17h7m(15h5m+2h2m)    | 16h55m(11h2m+5h53m)  | 19h25m(11h59m+7h26m) |
    | Football 5-vs-5 | 34h46m(32h47m+1h59m) | 32h46m(30h49m+1h57m) | 39h26m(31h54m+7h32m) | 37h26m(30h2m+7h24m)  |

---

> ### Author Response · Authors · 2022-11-19
> **Common Response [2/3]**
>
> - **Wall time usage comparison[Continued]**. We normalize the training time by dividing the training time of MAPPO in each task and show the results in the following table.
>
>     | Task            | MAPPO           | CoPPO           | HAPPO           | A2PO            |
>     | --------------- | --------------- | --------------- | --------------- | --------------- |
>     | 3s5z            | 1.00(0.87+0.13) | 1.02(0.89+0.13) | 1.10(0.89+0.20) | 1.30(1.06+0.25) |
>     | 27m_vs_30m      | 1.00(0.64+0.36) | 1.00(0.63+0.37) | 1.20(0.62+0.58) | 1.19(0.61+0.58) |
>     | Humanoid 9\|8   | 1.00(0.87+0.13) | 0.99(0.86+0.13) | 1.12(0.87+0.25) | 1.26(1.00+0.26) |
>     | Ant 4x2         | 1.00(0.93+0.07) | 1.01(0.94+0.07) | 1.12(0.91+0.21) | 1.05(0.83+0.22) |
>     | Humanoid 17x1   | 1.00(0.83+0.17) | 1.41(1.24+0.17) | 1.39(0.91+0.48) | 1.60(0.99+0.61) |
>     | Football 5-vs-5 | 1.00(0.94+0.06) | 0.94(0.89+0.06) | 1.13(0.92+0.22) | 1.08(0.86+0.21) |
>
>
> - **Performance in terms of environment steps and training time**. The following tables also present the experiments conducted on Humanoid 9|8. A2PO is advantageous in terms of both environment steps and wall time. A2PO's performance has improved significantly with minimal additional training time.
>
>     | Algo  | 0.5H                   | 1H                      | 1.5H                      | 2H                       | End - Time                       |
>     | ----- | ---------------------- | ----------------------- | ------------------------- | ------------------------ | -------------------------------- |
>     | MAPPO | $445.65(\pm11.43)$     | $555.34(\pm10.05)$      | $811.70(\pm384.30)$       | -                        | $2199.93(\pm1700.22)$ - 2h0m     |
>     | CoPPO | $463.18(\pm10.42)$     | $551.50(\pm12.35)$      | $613.08(\pm25.57)$        | $676.57(\pm42.25)$       | $676.57(\pm42.25)$ - 1h58m       |
>     | HAPPO | $\boldsymbol{468.13(\pm62.80)}$ | $573.01(\pm77.86)$      | $689.67(\pm118.64)$       | $1038.89(\pm560.46)$     | $1610.38(\pm826.31)$ - 2h15m     |
>     | A2PO  | $436.54(\pm10.71)$     | $\boldsymbol{636.14(\pm115.58)}$ | $\boldsymbol{1884.11(\pm1234.30)}$ | $\boldsymbol{4081.04(\pm477.27)}$ | $\boldsymbol{5595.27(\pm478.50)}$ - 2h31m |
>
>
>     | Algo  | $1e^6$                 | $2e^6$                  | $4e^6$                  | $8e^6$                   | $1e^7$                   |
>     | ----- | ---------------------- | ----------------------- | ----------------------- | ------------------------ | ------------------------ |
>     | MAPPO | $494.36(\pm12.96)$     | $544.92(\pm12.58)$      | $561.24(\pm14.82)$      | $928.36(\pm390.02)$      | $2199.93(\pm1700.22)$    |
>     | CoPPO | $522.24(\pm13.13)$     | $554.46(\pm12.23)$      | $562.80(\pm12.49)$      | $632.66(\pm25.75)$       | $676.57(\pm42.25)$       |
>     | HAPPO | $\boldsymbol{550.75(\pm69.06)}$ | $593.41(\pm77.86)$      | $616.11(\pm93.12)$      | $881.40(\pm258.87)$      | $1610.38(\pm826.31)$     |
>     | A2PO  | $538.77(\pm39.45)$     | $\boldsymbol{668.72(\pm115.58)}$ | $\boldsymbol{761.37(\pm197.91)}$ | $\boldsymbol{3976.87(\pm609.98)}$ | $\boldsymbol{5595.27(\pm478.50)}$ |
>
> - **Large-scale Applications**. One may worry that the increased training time requirement prevents applying A2PO in large-scale tasks with numerous agents. We demonstrate that the additional training time cost of A2PO is even less significant in light of the significant performance improvement in large-scale applications. For instance, experiments in Google Research Football 11-vs-11 full-game scenarios require a large number of training samples ($\sim 10^9$) and asynchronous training (collecting samples and training policies parallelly at the same time). The following table compares the mean win rates and standard errors of different methods over the training time. The results show that A2PO has an advantage over the wall-time in large-scale asynchronous training, though A2PO requires more time per policy update iteration.
>
>     | Algo        | 1 Hour              | 2 Hour              | 4 Hour              | 6 Hour              | 8 Hour              | 10 Hour             |
>     | ----------- | ------------------- | ------------------- | ------------------- | ------------------- | ------------------- | ------------------- |
>     | MAPPO       | $\boldsymbol{12.5(\pm0.3)}$  | $28.6(\pm 0.2)$     | $44.1(\pm 1.6)$     | $61.4(\pm 1.1)$     | $68.6(\pm 4.5)$     | $70.6(\pm 8.8)$     |
>     | CoPPO       | $11.0(\pm1.3)$      | $17.1(\pm2.5)$      | $40.3(\pm4.9)$      | $55.1(\pm 6.8)$     | $62.2(\pm5.2)$      | $62.8(\pm 6.7)$     |
>     | HAPPO       | $\boldsymbol{12.5(\pm 2.3)}$ | $27.5(\pm6.9)$      | $41.9(\pm12.5)$     | $50.3(\pm 12.9)$    | $54.6(\pm 14.7)$    | $59.3(\pm 13.2)$    |
>     | A2PO (ours) | $11.8(\pm 0.5)$     | $\boldsymbol{30.1(\pm 1.9)}$ | $\boldsymbol{58.7(\pm 4.5)}$ | $\boldsymbol{72.8(\pm 2.6)}$ | $\boldsymbol{77.6(\pm 4.8)}$ | $\boldsymbol{82.5(\pm 4.6)}$ |

---

> > ### Author Response · Authors · 2022-11-19
> > **Common Response [3/3]**
> >
> > ## Importance of the Monotonic Improvement Bound of A Single Agent
> >
> > One of the contributions of our work is retaining the monotonic improvement of a single agent. Here we clarify the importance in the following 3 points:
> >
> > 1. **Tighter Joint Monotonic Improvement Bound**. Given the monotonic improvement of a single agent, as shown in Table 1 in the paper, we obtain the tightest monotonic improvement bound among the single-rollout baselines for the joint policy. In addition, as described in appendix A.4, the monotonic improvement bounds encountered by various agents are gradually tightened. The properties relax the conditions to satisfy policy improvement for the agents, making the policy optimization more effective.
> > 2. **Derivation of the effective clipping objective**. The form of the single-agent clipping objective, shown in Equation 6, differs from that in HAPPO and other baselines. As mentioned at the beginning of Section 4, it is the monotonic improvement bound for a single agent that enables us to derive the proposed clipping objective. Detailedly, the term in the single-agent bound, i.e., $\alpha^{i}\sum_{j \in (e^{i} \cup \{i\})}\alpha^{j}$, indicates that when updating agent $i$, both the variation of the preceding agents' policies and the variation of agent $i$'s policy should be constrained. The results in the first ablation in Sec. 5.2 shows that clipping objective contributes to the performance improvement .
> > 3. **Profiling the contribution of each agent**. In contrast to HAPPO, where the joint objective is first proposed and then decomposed into individual objectives, in A2PO, the individual surrogate objectives are first proposed given the monotonic single-agent bound, and then constitute the joint surrogate objective in A2PO. The constitution of the joint objective can profile the influence of each agent updating policies on the overall optimization. We provide an illustration of how the influence is profiled in our work:
> >     - The clipping parameter adaptation. Knowing the constitution of the joint objective, we recognize that the non-stationarity encountered by an agent can be decomposed into the non-stationarity encountered by the preceding agents, as demonstrated by Proposition 2. The policy variation of a former agent contributes more to the non-stationarity problem than that of a latter agent, according to our findings. Given these observations, we propose the clipping parameter adaptation method to alleviate the non-stationarity problem. The ablation study depicted in Figure 7 demonstrates the clipping parameter adaptation method contributes to the performance improvement of A2PO while has no distinct influence on the performance of HAPPO.
> >
> >
> > ## Summary of Paper Updates
> > All minor issues have been addressed, including typographical errors, the absence of relevant references, tables, and figures, and the improper use of notations. The corresponding corrections and updates are summarized as follows for convenience.
> >
> > **Main Paper**
> > 1. We correct the typos in figure 2.
> > 2. We add a reference to V-trace in Sec. 5.2
> > 3. The results of Qmix are compared in Tables 2 and 5.
> > 4. The definitions of certain notations in Sec. 3.1 have been re-arranged.
> >
> > **Supplementary**
> > 1. At the beginning of appendix A, we include a table for the symbols and notations used in this paper.
> > 2. In appendix B.3, we analyze the wall-time usage.
> > 3. In appendix C, we discuss the value-based methods and T-PPO.
> > 4. The ablations are further explained in appendix B.2.5.

---

### Decision · Program_Chairs · 2023-01-20

**Decision:**

Accept: poster

**Justification For Why Not Higher Score:**

The paper could possibly be given a score of "Accept (spotlight)" based on score calibration with other accepted papers.

**Justification For Why Not Lower Score:**

"Accept (poster)" is justified as the reviewers have an overall positive assessment, and there is a consensus for acceptance.

**Metareview: Summary, Strengths And Weaknesses:**

The reviewers agreed that the paper addresses an important problem in multi-agent settings and proposes a novel policy gradient update algorithm for multi-agent reinforcement learning. However, the reviewers also raised several concerns and questions in their initial reviews. We want to thank the authors for their responses and active engagement during the discussion phase. The reviewers appreciated the responses, which helped in answering their key questions. The reviewers have an overall positive assessment of the paper, and there is a consensus for acceptance. The reviewers have provided detailed feedback, and we strongly encourage the authors to incorporate this feedback when preparing the final version of the paper.

**Note From Pc:**

if the above contains the word "oral" or "spotlight" please see: "oral" presentation means -> notable-top-5% and "spotlight" means -> notable-top-25%. As stated in our emails, we are disassociating presentation type from AC recommendations